# On the Local Minima of the Empirical Risk

**Chi Jin**[*]
University of California, Berkeley
chijin@cs.berkeley.edu

**Lydia T. Liu**[*]
University of California, Berkeley
lydiatliu@cs.berkeley.edu

**Rong Ge**
Duke University
rongge@cs.duke.edu

**Michael I. Jordan**
University of California, Berkeley
jordan@cs.berkeley.edu

## Abstract

Population risk is always of primary interest in machine learning; however, learning algorithms only have access to the empirical risk. Even for applications with nonconvex nonsmooth losses (such as modern deep networks), the population risk is generally significantly more well-behaved from an optimization point of view than the empirical risk. In particular, sampling can create many spurious local minima. We consider a general framework which aims to optimize a smooth nonconvex function $F$ (population risk) given only access to an approximation $f$ (empirical risk) that is pointwise close to $F$ (i.e., $\|F - f\|_\infty \leq \nu$). Our objective is to find the $\epsilon$-approximate local minima of the underlying function $F$ while avoiding the shallow local minima—arising because of the tolerance $\nu$—which exist only in $f$. We propose a simple algorithm based on stochastic gradient descent (SGD) on a smoothed version of $f$ that is guaranteed to achieve our goal as long as $\nu \leq O(\epsilon^{1.5}/d)$. We also provide an almost matching lower bound showing that our algorithm achieves optimal error tolerance $\nu$ among all algorithms making a polynomial number of queries of $f$. As a concrete example, we show that our results can be directly used to give sample complexities for learning a ReLU unit.

## 1 Introduction

The optimization of nonconvex loss functions has been key to the success of modern machine learning. While classical research in optimization focused on convex functions having a unique critical point that is both locally and globally minimal, a nonconvex function can have many local maxima, local minima and saddle points, all of which pose significant challenges for optimization. A recent line of research has yielded significant progress on one aspect of this problem—it has been established that favorable rates of convergence can be obtained even in the presence of saddle points, using simple variants of stochastic gradient descent [e.g., Ge et al., 2015, Carmon et al., 2016, Agarwal et al., 2017, Jin et al., 2017a]. These research results have introduced new analysis tools for nonconvex optimization, and it is of significant interest to begin to use these tools to attack the problems associated with undesirable local minima.

It is NP-hard to avoid all of the local minima of a general nonconvex function. But there are some classes of local minima where we might expect that simple procedures—such as stochastic gradient descent—may continue to prove effective. In particular, in this paper we consider local minima that are created by small perturbations to an underlying smooth objective function. Such a setting is natural in statistical machine learning problems, where data arise from an underlying population, and the population risk, $F$, is obtained as an expectation over a continuous loss function and is hence

---

[*]The first two authors contributed equally.

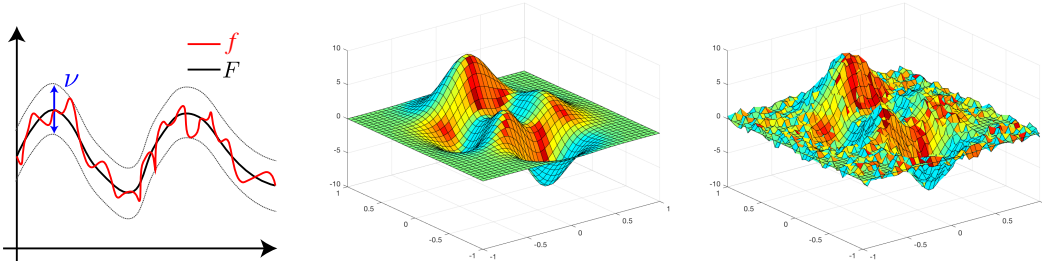

Figure 1: a) Function error $\nu$; b) Population risk vs empirical risk

smooth; i.e., we have $F(\boldsymbol{\theta}) = \mathbb{E}_{\mathbf{z} \sim \mathcal{D}}[L(\boldsymbol{\theta}; \mathbf{z})]$, for a loss function $L$ and population distribution $\mathcal{D}$. The sampling process turns this smooth risk into an empirical risk, $f(\boldsymbol{\theta}) = \sum_{i=1}^{n} L(\boldsymbol{\theta}; \mathbf{z}_i)/n$, which may be nonsmooth and which generally may have many shallow local minima. From an optimization point of view $f$ can be quite poorly behaved; indeed, it has been observed in deep learning that the empirical risk may have exponentially many shallow local minima, even when the underlying population risk is well-behaved and smooth almost everywhere [Brutzkus and Globerson, 2017, Auer et al., 1996]. From a statistical point of view, however, we can make use of classical results in empirical process theory [see, e.g., Boucheron et al., 2013, Bartlett and Mendelson, 2003] to show that, under certain assumptions on the sampling process, $f$ and $F$ are uniformly close:

$$\|F - f\|_\infty \le \nu, \tag{1}$$

where the error $\nu$ typically decreases with the number of samples $n$. See Figure 1(a) for a depiction of this result, and Figure 1(b) for an illustration of the effect of sampling on the optimization landscape. We wish to exploit this nearness of $F$ and $f$ to design and analyze optimization procedures that find approximate local minima (see Definition 1) of the smooth function $F$, while avoiding the local minima that exist only in the sampled function $f$.

Although the relationship between population risk and empirical risk is our major motivation, we note that other applications of our framework include two-stage robust optimization and private learning (see Section 5.2). In these settings, the error $\nu$ can be viewed as the amount of adversarial perturbation or noise due to sources other than data sampling. As in the sampling setting, we hope to show that simple algorithms such as stochastic gradient descent are able to escape the local minima that arise as a function of $\nu$.

Much of the previous work on this problem studies relatively small values of $\nu$, leading to "shallow" local minima, and applies relatively large amounts of noise, through algorithms such as simulated annealing [Belloni et al., 2015] and stochastic gradient Langevin dynamics (SGLD) [Zhang et al., 2017]. While such "large-noise algorithms" may be justified if the goal is to approach a stationary distribution, it is not clear that such large levels of noise is necessary in the optimization setting in order to escape shallow local minima. The best existing result for the setting of nonconvex $F$ requires the error $\nu$ to be smaller than $O(\epsilon^2/d^8)$, where $\epsilon$ is the precision of the optimization guarantee (see Definition 1) and $d$ is the problem dimension [Zhang et al., 2017] (see Figure 2). A fundamental question is whether algorithms exist that can tolerate a larger value of $\nu$, which would imply that they can escape "deeper" local minima. In the context of empirical risk minimization, such a result would allow fewer samples to be taken while still providing a strong guarantee on avoiding local minima.

We thus focus on the two central questions: **(1) Can a simple, optimization-based algorithm avoid shallow local minima despite the lack of "large noise"? (2) Can we tolerate larger error $\nu$ in the optimization setting, thus escaping "deeper" local minima? What is the largest error that the best algorithm can tolerate?**

In this paper, we answer both questions in the affirmative, establishing optimal dependencies between the error $\nu$ and the precision of a solution $\epsilon$. We propose a simple algorithm based on SGD (Algorithm 1) that is guaranteed to find an approximate local minimum of $F$ efficiently if $\nu \le O(\epsilon^{1.5}/d)$, thus escaping all saddle points of $F$ and all additional local minima introduced by $f$. Moreover, we provide a matching lower bound (up to logarithmic factors) for all algorithms making a polynomial number of queries of $f$. The lower bound shows that our algorithm achieves the optimal

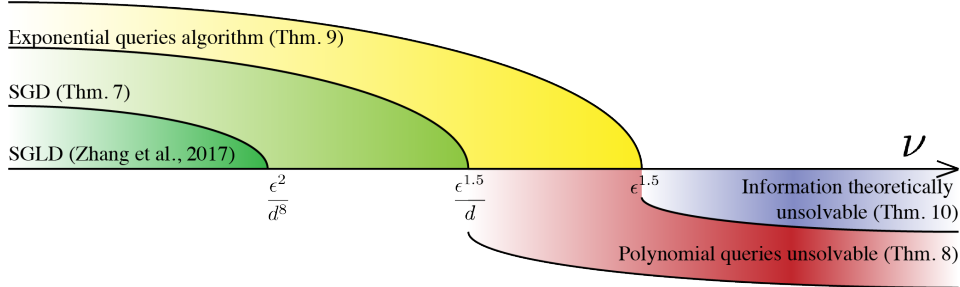

Figure 2: Complete characterization of error $\nu$ vs accuracy $\epsilon$ and dimension $d$.

tradeoff between $\nu$ and $\epsilon$, as well as the optimal dependence on dimension $d$. We also consider the information-theoretic limit for identifying an approximate local minimum of $F$ regardless of the number of queries. We give a sharp information-theoretic threshold: $\nu = \Theta(\epsilon^{1.5})$ (see Figure 2).

As a concrete example of the application to minimizing population risk, we show that our results can be directly used to give sample complexities for learning a ReLU unit, whose empirical risk is nonsmooth while the population risk is smooth almost everywhere.

## 1.1 Related Work

A number of other papers have examined the problem of optimizing a target function $F$ given only function evaluations of a function $f$ that is pointwise close to $F$. Belloni et al. [2015] proposed an algorithm based on simulated annealing. The work of Risteski and Li [2016] and Singer and Vondrak [2015] discussed lower bounds, though only for the setting in which the target function $F$ is convex. For nonconvex target functions $F$, Zhang et al. [2017] studied the problem of finding approximate local minima of $F$, and proposed an algorithm based on Stochastic Gradient Langevin Dynamics (SGLD) [Welling and Teh, 2011], with maximum tolerance for function error $\nu$ scaling as $O(\epsilon^2/d^8)^2$. Other than difference in algorithm style and $\nu$ tolerance as shown in Figure 2, we also note that we do not require regularity assumptions on top of smoothness, which are inherently required by the MCMC algorithm proposed in Zhang et al. [2017]. Finally, we note that in parallel, Kleinberg et al. [2018] solved a similar problem using SGD under the assumption that $F$ is one-point convex.

Previous work has also studied the relation between the landscape of empirical risks and the landscape of population risks for nonconvex functions. Mei et al. [2016] examined a special case where the individual loss functions $L$ are also smooth, which under some assumptions implies uniform convergence of the gradient and Hessian of the empirical risk to their population versions. Loh and Wainwright [2013] showed for a restricted class of nonconvex losses that even though many local minima of the empirical risk exist, they are all close to the global minimum of population risk.

Our work builds on recent work in nonconvex optimization, in particular, results on escaping saddle points and finding approximate local minima. Beyond the classical result by Nesterov [2004] for finding first-order stationary points by gradient descent, recent work has given guarantees for escaping saddle points by gradient descent [Jin et al., 2017a] and stochastic gradient descent [Ge et al., 2015]. Agarwal et al. [2017] and Carmon et al. [2016] established faster rates using algorithms that make use of Nesterov's accelerated gradient descent in a nested-loop procedure [Nesterov, 1983], and Jin et al. [2017b] have established such rates even without the nested loop. There have also been empirical studies on various types of local minima [e.g. Keskar et al., 2016, Dinh et al., 2017].

Finally, our work is also related to the literature on zero-th order optimization or more generally, bandit convex optimization. Our algorithm uses function evaluations to construct a gradient estimate and perform SGD, which is similar to standard methods in this community [e.g., Flaxman et al., 2005, Agarwal et al., 2010, Duchi et al., 2015]. Compared to first-order optimization, however, the convergence of zero-th order methods is typically much slower, depending polynomially on the underlying dimension even in the convex setting [Shamir, 2013]. Other derivative-free optimization methods

include simulated annealing [Kirkpatrick et al., 1983] and evolutionary algorithms [Rechenberg and Eigen, 1973], whose convergence guarantees are less clear.

## 2 Preliminaries

**Notation** We use bold lower-case letters to denote vectors, as in $\mathbf{x}, \mathbf{y}, \mathbf{z}$. We use $\|\cdot\|$ to denote the $\ell_2$ norm of vectors and spectral norm of matrices. For a matrix, $\lambda_{\min}$ denotes its smallest eigenvalue. For a function $f : \mathbb{R}^d \to \mathbb{R}$, $\nabla f$ and $\nabla^2 f$ denote its gradient vector and Hessian matrix respectively. We also use $\|\cdot\|_\infty$ on a function $f$ to denote the supremum of its absolute function value over entire domain, $\sup_{\mathbf{x} \in \mathbb{R}^d} |f|$. We use $\mathbb{B}_0(r)$ to denote the $\ell_2$ ball of radius $r$ centered at $0$ in $\mathbb{R}^d$. We use notation $\tilde{O}(\cdot), \tilde{\Theta}(\cdot), \tilde{\Omega}(\cdot)$ to hide only absolute constants and poly-logarithmic factors. A multivariate Gaussian distribution with mean $\mathbf{0}$ and covariance $\sigma^2$ in every direction is denoted as $\mathcal{N}(\mathbf{0}, \sigma^2 \boldsymbol{I})$. Throughout the paper, we say "polynomial number of queries" to mean that the number of queries depends polynomially on all problem-dependent parameters.

**Objectives in nonconvex optimization** Our goal is to find a point that has zero gradient and positive semi-definite Hessian, thus escaping saddle points. We formalize this idea as follows.

**Definition 1.** $\mathbf{x}$ is called a **second-order stationary point** (SOSP) or **approximate local minimum** of a function $F$ if
$$\|\nabla F(\mathbf{x})\| = 0 \text{ and } \lambda_{\min}(\nabla^2 F(\mathbf{x})) \geq 0.$$

We note that there is a slight difference between SOSP and local minima—an SOSP as defined here does not preclude higher-order saddle points, which themselves can be NP-hard to escape from [Anandkumar and Ge, 2016].

Since an SOSP is characterized by its gradient and Hessian, and since convergence of algorithms to an SOSP will depend on these derivatives in a neighborhood of an SOSP, it is necessary to impose smoothness conditions on the gradient and Hessian. A minimal set of conditions that have become standard in the literature are the following.

**Definition 2.** A function $F$ is $\ell$-**gradient Lipschitz** if $\forall \mathbf{x}, \mathbf{y} \ \|\nabla F(\mathbf{x}) - \nabla F(\mathbf{y})\| \leq \ell \|\mathbf{x} - \mathbf{y}\|$.

**Definition 3.** A function $F$ is $\rho$-**Hessian Lipschitz** if $\forall \mathbf{x}, \mathbf{y} \ \|\nabla^2 F(\mathbf{x}) - \nabla^2 F(\mathbf{y})\| \leq \rho \|\mathbf{x} - \mathbf{y}\|$.

Another common assumption is that the function is bounded.

**Definition 4.** A function $F$ is $B$-**bounded** if for any $\mathbf{x}$ that $|F(\mathbf{x})| \leq B$.

For any finite-time algorithm, we cannot hope to find an *exact* SOSP. Instead, we can define $\epsilon$-approximate SOSP that satisfy relaxations of the first- and second-order optimality conditions. Letting $\epsilon$ vary allows us to obtain rates of convergence.

**Definition 5.** $\mathbf{x}$ is an $\epsilon$-**second-order stationary point** ($\epsilon$-SOSP) of a $\rho$-Hessian Lipschitz function $F$ if
$$\|\nabla F(\mathbf{x})\| \leq \epsilon \text{ and } \lambda_{\min}(\nabla^2 F(\mathbf{x})) \geq -\sqrt{\rho\epsilon}.$$

Given these definitions, we can ask whether it is possible to find an $\epsilon$-SOSP in polynomial time under the Lipchitz properties. Various authors have answered this question in the affirmative.

**Theorem 6.** *[e.g. Carmon et al., 2016, Agarwal et al., 2017, Jin et al., 2017a] If the function $F : \mathbb{R}^d \to \mathbb{R}$ is B-bounded, l-gradient Lipschitz and $\rho$ Hessian Lipschitz, given access to the gradient (and sometimes Hessian) of F, it is possible to find an $\epsilon$-SOSP in poly($d, B, l, \rho, 1/\epsilon$) time.*

## 3 Main Results

In the setting we consider, there is an unknown function $F$ (the population risk) that has regularity properties (bounded, gradient and Hessian Lipschitz). However, we only have access to a function $f$ (the empirical risk) that may not even be everywhere differentiable. The only information we use is that $f$ is pointwise close to $F$. More precisely, we assume

**Assumption A1.** We assume that the function pair $(F : \mathbb{R}^d \to \mathbb{R}, f : \mathbb{R}^d \to \mathbb{R})$ satisfies the following properties:

1. $F$ is $B$-bounded, $\ell$-gradient Lipschitz, $\rho$-Hessian Lipschitz.

---

**Algorithm 1** Zero-th order Perturbed Stochastic Gradient Descent (ZPSGD)

---

**Input:** $\mathbf{x}_0$, learning rate $\eta$, noise radius $r$, mini-batch size $m$.
    **for** $t = 0, 1, \ldots,$ **do**
        sample $(\mathbf{z}_t^{(1)}, \cdots, \mathbf{z}_t^{(m)}) \sim \mathcal{N}(0, \sigma^2 \mathbf{I})$
        $\mathbf{g}_t(\mathbf{x}_t) \leftarrow \sum_{i=1}^m \mathbf{z}_t^{(i)}[f(\mathbf{x}_t + \mathbf{z}_t^{(i)}) - f(\mathbf{x}_t)]/(m\sigma^2)$
        $\mathbf{x}_{t+1} \leftarrow \mathbf{x}_t - \eta(\mathbf{g}_t(\mathbf{x}_t) + \xi_t), \qquad \xi_t \text{ uniformly } \sim \mathbb{B}_0(r)$
    **return** $\mathbf{x}_T$

---

    2. $f, F$ are $\nu$-pointwise close; i.e., $\|F - f\|_\infty \leq \nu$.

As we explained in Section 2, our goal is to find second-order stationary points of $F$ given only function value access to $f$. More precisely:

**Problem 1.** Given a function pair $(F, f)$ that satisfies Assumption A1, find an $\epsilon$-second-order stationary point of $F$ with only access to values of $f$.

The only way our algorithms are allowed to interact with $f$ is to query a point $\mathbf{x}$, and obtain a function value $f(\mathbf{x})$. This is usually called a *zero-th order* oracle in the optimization literature. In this paper we give tight upper and lower bounds for the dependencies between $\nu$, $\epsilon$ and $d$, both for algorithms with polynomially many queries and in the information-theoretic limit.

### 3.1 Optimal algorithm with polynomial number of queries

There are three main difficulties in applying stochastic gradient descent to Problem 1: (1) in order to converge to a second-order stationary point of $F$, the algorithm must avoid being stuck in saddle points; (2) the algorithm does not have access to the gradient of $f$; (3) there is a gap between the observed $f$ and the target $F$, which might introduce non-smoothness or additional local minima. The first difficulty was addressed in Jin et al. [2017a] by perturbing the iterates in a small ball; this pushes the iterates away from any potential saddle points. For the latter two difficulties, we apply Gaussian smoothing to $f$ and use $\mathbf{z}[f(\mathbf{x} + \mathbf{z}) - f(\mathbf{x})]/\sigma^2$ ($\mathbf{z} \sim \mathcal{N}(0, \sigma^2 \mathbf{I})$) as a stochastic gradient estimate. This estimate, which only requires function values of $f$, is well known in the zero-th order optimization literature [e.g. Duchi et al., 2015]. For more details, see Section 4.1.

In short, our algorithm (Algorithm 1) is a variant of SGD, which uses $\mathbf{z}[f(\mathbf{x} + \mathbf{z}) - f(\mathbf{x})]/\sigma^2$ as the gradient estimate (computed over mini-batches), and adds isotropic perturbations. Using this algorithm, we can achieve the following trade-off between $\nu$ and $\epsilon$.

**Theorem 7** (Upper Bound (ZPSGD)). *Given that the function pair $(F, f)$ satisfies Assumption A1 with $\nu \leq O(\sqrt{\epsilon^3/\rho} \cdot (1/d))$, then for any $\delta > 0$, with smoothing parameter $\sigma = \Theta(\sqrt{\epsilon/(\rho d)})$, learning rate $\eta = 1/\ell$, perturbation $r = \tilde{\Theta}(\epsilon)$, and mini-batch size $m = poly(d, B, \ell, \rho, 1/\epsilon, \log(1/\delta))$, ZPSGD will find an $\epsilon$-second-order stationary point of $F$ with probability $1 - \delta$, in $poly(d, B, \ell, \rho, 1/\epsilon, \log(1/\delta))$ number of queries.*

Theorem 7 shows that assuming a small enough function error $\nu$, ZPSGD will solve Problem 1 within a number of queries that is polynomial in all the problem-dependent parameters. The tolerance on function error $\nu$ varies inversely with the number of dimensions, $d$. This rate is in fact optimal for all polynomial queries algorithms. In the following result, we show that the $\epsilon, \rho$, and $d$ dependencies in function difference $\nu$ are tight up to a logarithmic factors in $d$.

**Theorem 8** (Polynomial Queries Lower Bound). *For any $B > 0, \ell > 0, \rho > 0$ there exists $\epsilon_0 = \Theta(\min\{\ell^2/\rho, (B^2\rho/d^2)^{1/3}\})$ such that for any $\epsilon \in (0, \epsilon_0]$, there exists a function pair $(F, f)$ satisfying Assumption A1 with $\nu = \tilde{\Theta}(\sqrt{\epsilon^3/\rho} \cdot (1/d))$, so that any algorithm that only queries a polynomial number of function values of $f$ will fail, with high probability, to find an $\epsilon$-SOSP of $F$.*

This theorem establishes that for any $\rho, \ell, B$ and any $\epsilon$ small enough, we can construct a randomized 'hard' instance $(F, f)$ such that any (possibly randomized) algorithm with a polynomial number of queries will fail to find an $\epsilon$-SOSP of $F$ with high probability. Note that the error $\nu$ here is only a poly-logarithmic factor larger than the requirement for our algorithm. In other words, the guarantee of our Algorithm 1 in Theorem 7 is optimal up to a logarithmic factor.

## 3.2 Information-theoretic guarantees

If we allow an unlimited number of queries, we can show that the upper and lower bounds on the function error tolerance $\nu$ no longer depends on the problem dimension $d$. That is, Problem 1 exhibits a statistical-computational gap—polynomial-queries algorithms are unable to achieve the information-theoretic limit. We first state that an algorithm (with exponential queries) is able to find an $\epsilon$-SOSP of $F$ despite a much larger value of error $\nu$. The basic algorithmic idea is that an $\epsilon$-SOSP must exist within some compact space, such that once we have a subroutine that approximately computes the gradient and Hessian of $F$ at an arbitrary point, we can perform a grid search over this compact space (see Section D for more details):

**Theorem 9.** *There exists an algorithm so that if the function pair $(F, f)$ satisfies Assumption A1 with $\nu \leq O(\sqrt{\epsilon^3/\rho})$ and $\ell > \sqrt{\rho\epsilon}$, then the algorithm will find an $\epsilon$-second-order stationary point of $F$ with an exponential number of queries.*

We also show a corresponding information-theoretic lower bound that prevents any algorithm from even identifying a second-order stationary point of $F$. This completes the characterization of function error tolerance $\nu$ in terms of required accuracy $\epsilon$.

**Theorem 10.** *For any $B > 0, \ell > 0, \rho > 0$, there exists $\epsilon_0 = \Theta(\min\{\ell^2/\rho, (B^2\rho/d)^{1/3}\})$ such that for any $\epsilon \in (0, \epsilon_0]$ there exists a function pair $(F, f)$ satisfying Assumption A1 with $\nu = O(\sqrt{\epsilon^3/\rho})$, so that any algorithm will fail, with high probability, to find an $\epsilon$-SOSP of $F$.*

## 3.3 Extension: Gradients pointwise close

We may extend our algorithmic ideas to solve the problem of optimizing an unknown smooth function $F$ when given only a gradient vector field $\mathbf{g} : \mathbb{R}^d \to \mathbb{R}^d$ that is pointwise close to the gradient $\nabla F$. Specifically, we answer the question: what is the error in the gradient oracle that we can tolerate to obtain optimization guarantees for the true function $F$? We observe that our algorithm's tolerance on gradient error is much better compared to Theorem 7. Details can be found in Appendix E and F.

# 4 Overview of Analysis

In this section we present the key ideas underlying our theoretical results. We will focus on the results for algorithms that make a polynomial number of queries (Theorems 7 and 8).

## 4.1 Efficient algorithm for Problem 1

We first argue the correctness of Theorem 7. As discussed earlier, there are two key ideas in the algorithm: Gaussian smoothing and perturbed stochastic gradient descent. Gaussian smoothing allows us to transform the (possibly non-smooth) function $f$ into a smooth function $\tilde{f}_\sigma$ that has similar second-order stationary points as $F$; at the same time, it can also convert function evaluations of $f$ into a stochastic gradient of $\tilde{f}_\sigma$. We can use this stochastic gradient information to find a second-order stationary point of $\tilde{f}_\sigma$, which by the choice of the smoothing radius is guaranteed to be an approximate second-order stationary point of $F$.

First, we introduce Gaussian smoothing, which perturbs the current point $\mathbf{x}$ using a multivariate Gaussian and then takes an expectation over the function value.

**Definition 11** (Gaussian smoothing). Given $f$ satisfying assumption A1, define its Gaussian smoothing as $\tilde{f}_\sigma(\mathbf{x}) = \mathbb{E}_{\mathbf{z}\sim\mathcal{N}(0,\sigma^2\mathbf{I})}[f(\mathbf{x} + \mathbf{z})]$. The parameter $\sigma$ is henceforth called the smoothing radius.

In general $f$ need not be smooth or even differentiable, but its Gaussian smoothing $\tilde{f}_\sigma$ will be a differentiable function. Although it is in general difficult to calculate the exact smoothed function $\tilde{f}_\sigma$, it is not hard to give an unbiased estimate of function value and gradient of $\tilde{f}_\sigma$:

**Lemma 12.** *[e.g. Duchi et al., 2015] Let $\tilde{f}_\sigma$ be the Gaussian smoothing of $f$ (as in Definition 11), the gradient of $\tilde{f}_\sigma$ can be computed as $\nabla\tilde{f}_\sigma = \frac{1}{\sigma^2}\mathbb{E}_{\mathbf{z}\sim\mathcal{N}(0,\sigma^2\mathbf{I})}[(f(\mathbf{x} + \mathbf{z}) - f(\mathbf{x}))\mathbf{z}]$.*

Lemma 12 allows us to query the function value of $f$ to get an unbiased estimate of the gradient of $\tilde{f}_\sigma$. This stochastic gradient is used in Algorithm 1 to find a second-order stationary point of $\tilde{f}_\sigma$.

To make sure the optimizer is effective on $\tilde{f}_\sigma$ and that guarantees on $\tilde{f}_\sigma$ carry over to the target function $F$, we need two sets of properties: the smoothed function $\tilde{f}_\sigma$ should be gradient and Hessian

Lipschitz, and at the same time should have gradients and Hessians close to those of the true function $F$. These properties are summarized in the following lemma:

**Lemma 13** (Property of smoothing). *Assume that the function pair $(F, f)$ satisfies Assumption A1, and let $\tilde{f}_\sigma(\mathbf{x})$ be as given in definition 11. Then, the following holds*

1. *$\tilde{f}_\sigma(\mathbf{x})$ is $O(\ell + \frac{\nu}{\sigma^2})$-gradient Lipschitz and $O(\rho + \frac{\nu}{\sigma^3})$-Hessian Lipschitz.*

2. *$\|\nabla \tilde{f}_\sigma(\mathbf{x}) - \nabla F(\mathbf{x})\| \leq O(\rho d \sigma^2 + \frac{\nu}{\sigma})$ and $\|\nabla^2 \tilde{f}_\sigma(\mathbf{x}) - \nabla^2 F(\mathbf{x})\| \leq O(\rho \sqrt{d} \sigma + \frac{\nu}{\sigma^2})$.*

The proof is deferred to Appendix A. Part (1) of the lemma says that the gradient (and Hessian) Lipschitz constants of $\tilde{f}_\sigma$ are similar to the gradient (and Hessian) Lipschitz constants of $F$ up to a term involving the function difference $\nu$ and the smoothing parameter $\sigma$. This means as $f$ is allowed to deviate further from $F$, we must smooth over a larger radius—choose a larger $\sigma$—to guarantee the same smoothness as before. On the other hand, part (2) implies that choosing a large $\sigma$ increases the upper bound on the gradient and Hessian *difference* between $\tilde{f}_\sigma$ and $F$. Smoothing is a form of local averaging, so choosing a too-large radius will erase information about local geometry. The choice of $\sigma$ must strike the right balance between making $\tilde{f}_\sigma$ smooth (to guarantee ZPSGD finds a $\epsilon$-SOSP of $\tilde{f}_\sigma$ ) and keeping the derivatives of $\tilde{f}_\sigma$ close to those of $F$ (to guarantee any $\epsilon$-SOSP of $\tilde{f}_\sigma$ is also an $O(\epsilon)$-SOSP of $F$). In Appendix A.3, we show that this can be satisfied by choosing $\sigma = \sqrt{\epsilon/(\rho d)}$.

**Perturbed stochastic gradient descent** In ZPSGD, we use the stochastic gradients suggested by Lemma 12. Perturbed Gradient Descent (PGD) [Jin et al., 2017a] was shown to converge to a second-order stationary point. Here we use a simple modification of PGD that relies on batch stochastic gradient. In order for PSGD to converge, we require that the stochastic gradients are well-behaved; that is, they are unbiased and have good concentration properties, as asserted in the following lemma. It is straightforward to verify given that we sample $\mathbf{z}$ from a zero-mean Gaussian (proof in Appendix A.2).

**Lemma 14** (Property of stochastic gradient). *Let $\mathbf{g}(\mathbf{x}; \mathbf{z}) = \mathbf{z}[f(\mathbf{x} + \mathbf{z}) - f(\mathbf{x})]/\sigma^2$, where $\mathbf{z} \sim \mathcal{N}(0, \sigma^2 \mathbf{I})$. Then $\mathbb{E}_{\mathbf{z}} \mathbf{g}(\mathbf{x}; \mathbf{z}) = \nabla \tilde{f}_\sigma(\mathbf{x})$, and $\mathbf{g}(\mathbf{x}; \mathbf{z})$ is sub-Gaussian with parameter $\frac{B}{\sigma}$.*

As it turns out, these assumptions suffice to guarantee that perturbed SGD (PSGD), a simple adaptation of PGD in Jin et al. [2017a] with stochastic gradient and large mini-batch size, converges to the second-order stationary point of the objective function.

**Theorem 15** (PSGD efficiently escapes saddle points [Jin et al., 2018], informal). *Suppose $f(\cdot)$ is $\ell$-gradient Lipschitz and $\rho$-Hessian Lipschitz, and stochastic gradient $\mathbf{g}(\mathbf{x}, \theta)$ with $\mathbb{E}\mathbf{g}(\mathbf{x}; \theta) = \nabla f(\mathbf{x})$ has a sub-Gaussian tail with parameter $\sigma/\sqrt{d}$, then for any $\delta > 0$, with proper choice of hyperparameters, PSGD (Algorithm 3) will find an $\epsilon$-SOSP of $f$ with probability $1 - \delta$, in $\text{poly}(d, B, \ell, \rho, \sigma, 1/\epsilon, \log(1/\delta))$ number of queries.*

For completeness, we include the formal version of the theorem and its proof in Appendix H. Combining this theorem and the second part of Lemma 13, we see that by choosing an appropriate smoothing radius $\sigma$, our algorithm ZPSGD finds an $C\epsilon/\sqrt{d}$-SOSP for $\tilde{f}_\sigma$ which is also an $\epsilon$-SOSP for $F$ for some universal constant $C$.

### 4.2 Polynomial queries lower bound

The proof of Theorem 8 depends on the construction of a 'hard' function pair. The argument crucially depends on the concentration of measure in high dimensions. We provide a proof sketch in Appendix B and the full proof in Appendix C.

## 5 Applications

In this section, we present several applications of our algorithm. We first show a simple example of learning one rectified linear unit (ReLU), where the empirical risk is nonconvex and nonsmooth. We also briefly survey other potential applications for our model as stated in Problem 1.

### 5.1 Statistical Learning Example: Learning ReLU

Consider the simple example of learning a ReLU unit. Let $\text{ReLU}(z) = \max\{z, 0\}$ for $z \in \mathbb{R}$. Let $\mathbf{w}^\star(\|\mathbf{w}^\star\| = 1)$ be the desired solution. We assume data $(\mathbf{x}_i, \mathbf{y}_i)$ is generated as $y_i = \text{ReLU}(\mathbf{x}_i^\top \mathbf{w}^\star) +$

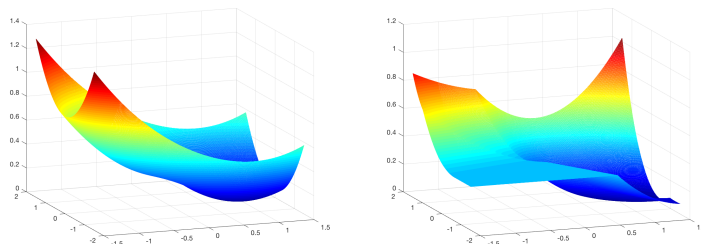

Figure 3: Population (left) and Empirical (right) risk for learning ReLU Unit , $d = 2$. Sharp corners present in the empirical risk are not found in the population version.

$\zeta_i$ where noise $\zeta_i \sim \mathcal{N}(0, 1)$. We further assume the features $\mathbf{x}_i \sim \mathcal{N}(0, \mathbf{I})$ are also generated from a standard Gaussian distribution. The empirical risk with a squared loss function is:

$$\hat{R}_n(\mathbf{w}) = \frac{1}{n} \sum_{i=1}^{n} (y_i - \text{ReLU}(\mathbf{x}_i^\top \mathbf{w}))^2.$$

Its population version is $R(\mathbf{w}) = \mathbb{E}[\hat{R}_n(\mathbf{w})]$. In this case, the empirical risk is highly nonsmooth—in fact, not differentiable in all subspaces perpendicular to each $\mathbf{x}_i$. The population risk turns out to be smooth in the entire space $\mathbb{R}^d$ except at $\mathbf{0}$. This is illustrated in Figure 3, where the empirical risk displays many sharp corners.

Due to nonsmoothness at $\mathbf{0}$ even for population risk, we focus on a compact region $\mathfrak{B} = \{\mathbf{w}|\mathbf{w}^\top \mathbf{w}^\star \geq \frac{1}{\sqrt{d}}\} \cap \{\mathbf{w}|\|\mathbf{w}\| \leq 2\}$ which excludes $\mathbf{0}$. This region is large enough so that a random initialization has at least constant probability of being inside it. We also show the following properties that allow us to apply Algorithm 1 directly:

**Lemma 16.** *The population and empirical risk $R$, $\hat{R}_n$ of learning a ReLU unit problem satisfies:*
*1. If $\mathbf{w}_0 \in \mathfrak{B}$, then runing ZPSGD (Algorithm 1) gives $\mathbf{w}_t \in \mathfrak{B}$ for all $t$ with high probability.*
*2. Inside $\mathfrak{B}$, $R$ is $O(1)$-bounded, $O(\sqrt{d})$-gradient Lipschitz, and $O(d)$-Hessian Lipschitz.*
*3. $\sup_{\mathbf{w} \in \mathfrak{B}} |\hat{R}_n(\mathbf{w}) - R(\mathbf{w})| \leq \tilde{O}(\sqrt{d/n})$ w.h.p.*
*4. Inside $\mathfrak{B}$, $R$ is nonconvex function, $\mathbf{w}^\star$ is the only SOSP of $R(\mathbf{w})$.*

These properties show that the population loss has a well-behaved landscape, while the empirical risk is pointwise close. This is exactly what we need for Algorithm 1. Using Theorem 7 we immediately get the following sample complexity, which guarantees an approximate population risk minimizer. We defer all proofs to Appendix G.

**Theorem 17.** *For learning a ReLU unit problem, suppose the sample size is $n \geq \tilde{O}(d^4/\epsilon^3)$, and the initialization is $\mathbf{w}_0 \sim \mathcal{N}(0, \frac{1}{d}\mathbf{I})$, then with at least constant probability, Algorithm 1 will output an estimator $\hat{\mathbf{w}}$ so that $\|\hat{\mathbf{w}} - \mathbf{w}^\star\| \leq \epsilon$.*

### 5.2 Other applications

**Private machine learning**    Data privacy is a significant concern in machine learning as it creates a trade-off between privacy preservation and successful learning. Previous work on differentially private machine learning [e.g. Chaudhuri et al., 2011] have studied *objective perturbation*, that is, adding noise to the original (convex) objective and optimizing this perturbed objective, as a way to simultaneously guarantee differential privacy and learning generalization: $f = F + p(\varepsilon)$. Our results may be used to extend such guarantees to nonconvex objectives, characterizing when it is possible to optimize $F$ even if the data owner does not want to reveal the true value of $F(\mathbf{x})$ and instead only reveals $f(\mathbf{x})$ after adding a perturbation $p(\varepsilon)$, which depends on the privacy guarantee $\varepsilon$.

**Two stage robust optimization**    Motivated by the problem of adversarial examples in machine learning, there has been a lot of recent interest [e.g. Steinhardt et al., 2017, Sinha et al., 2018] in a form of robust optimization that involves a minimax problem formulation: $\min_{\mathbf{x}} \max_{\mathbf{u}} G(\mathbf{x}, \mathbf{u})$. The function $F(\mathbf{x}) = \max_{\mathbf{u}} G(\mathbf{x}, \mathbf{u})$ tends to be nonconvex in such problems, since $G$ can be very complicated. It can be intractable or costly to compute the solution to the inner maximization exactly, but it is often possible to get a good enough approximation $f$, such that $\sup_{\mathbf{x}} |F(\mathbf{x}) - f(\mathbf{x})| = \nu$. It is then possible to solve $\min_{\mathbf{x}} f(\mathbf{x})$ by ZPSGD, with guarantees for the original optimization problem.

**Acknowledgments**

We thank Aditya Guntuboyina, Yuanzhi Li, Yi-An Ma, Jacob Steinhardt, and Yang Yuan for valuable discussions.

## Footnotes

[2]The difference between the scaling for $\nu$ asserted here and the $\nu = O(\epsilon^2)$ claimed in [Zhang et al., 2017] is due to difference in assumptions. In our paper we assume that the Hessian is Lipschitz with respect to the standard spectral norm; Zhang et al. [2017] make such an assumption with respect to nuclear norm.

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
