[Supplementary Material]

# Appendix for *On the Local Minima of the Empirical Risk*

## A  Efficient algorithm for optimizing the population risk

As we described in Section 4, in order to find a second-order stationary point of the population loss $F$, we apply perturbed stochastic gradient on a smoothed version of the empirical loss $f$. Recall that the smoothed function is defined as

$$\tilde{f}_\sigma(\mathbf{x}) = \mathbb{E}_{\mathbf{z}} f(\mathbf{x} + \mathbf{z}).$$

In this section we will also consider a smoothed version of the population loss $F$, as follows:

$$\tilde{F}_\sigma(\mathbf{x}) = \mathbb{E}_{\mathbf{z}} F(\mathbf{x} + \mathbf{z}).$$

This function is of course not accessible by the algorithm and we only use it in the proof of convergence rates.

This section is organized as follows. In section A.1, we present and prove the key lemma on the properties of the smoothed function $\tilde{f}_\sigma(\mathbf{x})$. Next, in section A.2, we prove the properties of the stochastic gradient $\mathbf{g}$. Combining the lemmas in these two subsections, in section A.3 we prove a main theorem about the guarantees of ZPSGD (Theorem 7). For clarity, we defer all technical lemmas and their proofs to section A.4.

### A.1  Properties of the Gaussian smoothing

In this section, we show the properties of smoothed function $\tilde{f}_\sigma(\mathbf{x})$. We first restate Lemma 13.

**Lemma 1** (Property of smoothing). *Assume that the function pair $(F, f)$ satisfies Assumption A1, and let $\tilde{f}_\sigma(\mathbf{x})$ be as given in definition 11. Then, the following holds*

1. *$\tilde{f}_\sigma(\mathbf{x})$ is $O(\ell + \frac{\nu}{\sigma^2})$-gradient Lipschitz and $O(\rho + \frac{\nu}{\sigma^3})$-Hessian Lipschitz.*

2. *$\|\nabla \tilde{f}_\sigma(\mathbf{x}) - \nabla F(\mathbf{x})\| \leq O(\rho d \sigma^2 + \frac{\nu}{\sigma})$ and $\|\nabla^2 \tilde{f}_\sigma(\mathbf{x}) - \nabla^2 F(\mathbf{x})\| \leq O(\rho \sqrt{d} \sigma + \frac{\nu}{\sigma^2})$.*

Intuitively, the first property states that if the original function $F$ is gradient and Hessian Lipschitz, the smoothed version of the perturbed function $f$ is also gradient and Hessian Lipschitz (note that this is of course not true for the perturbed function $f$); the second property shows that the gradient and Hessian of $\tilde{f}_\sigma$ is point-wise close to the gradient and Hessian of the original function $F$. We will prove the four points (1 and 2, gradient and Hessian) of the lemma one by one, in Sections A.1.1 to A.1.4.

In the proof, we frequently require the following lemma (see e.g. Zhang et al. [2017]) that gives alternative expressions for the gradient and Hessian of a smoothed function.

**Lemma 2** (Gaussian smoothing identities [Zhang et al., 2017]). *$\tilde{f}_\sigma$ has gradient and Hessian:*

$$\nabla \tilde{f}_\sigma(\mathbf{x}) = \mathbb{E}_{\mathbf{z}}\Big[\frac{\mathbf{z}}{\sigma^2} f(\mathbf{x} + \mathbf{z})\Big], \quad \nabla^2 \tilde{f}_\sigma(\mathbf{x}) = \mathbb{E}_{\mathbf{z}}\Big[\frac{\mathbf{z}\mathbf{z}^\top - \sigma^2 \mathbf{I}}{\sigma^4} f(\mathbf{x} + \mathbf{z})\Big].$$

*Proof.* Using the density function of a multivariate Gaussian, we may compute the gradient of the smoothed function as follows:

$$\nabla \tilde{f}_\sigma(\mathbf{x}) = \frac{\partial}{\partial \mathbf{x}} \frac{1}{(2\pi\sigma^2)^{d/2}} \int f(\mathbf{x}+\mathbf{z}) e^{-\|\mathbf{z}\|^2/2\sigma^2} d\mathbf{z} = \frac{1}{(2\pi\sigma^2)^{d/2}} \int \frac{\partial}{\partial \mathbf{x}} f(\mathbf{x}+\mathbf{z}) e^{-\|\mathbf{z}\|^2/2\sigma^2} d\mathbf{z}$$

$$= \frac{1}{(2\pi\sigma^2)^{d/2}} \int \frac{\partial}{\partial \mathbf{x}} f(\mathbf{z}) e^{-\|\mathbf{z}-\mathbf{x}\|^2/2\sigma^2} d\mathbf{z} = \frac{1}{(2\pi\sigma^2)^{d/2}} \int f(\mathbf{z}') \frac{\mathbf{x}-\mathbf{z}}{\sigma^2} e^{-\|\mathbf{z}-\mathbf{x}\|^2/2\sigma^2} d\mathbf{z}$$

$$= \frac{1}{(2\pi\sigma^2)^{d/2}} \int f(\mathbf{z}+\mathbf{x}) \frac{\mathbf{z}}{\sigma^2} e^{-\|-\mathbf{z}\|^2/2\sigma^2} d\mathbf{z} = \mathbb{E}_{\mathbf{z}}[\frac{\mathbf{z}}{\sigma^2} f(\mathbf{x}+\mathbf{z})],$$

and similarly, we may compute the Hessian of the smoothed function:

$$\nabla^2 \tilde{f}_\sigma(\mathbf{x}) = \frac{\partial}{\partial \mathbf{x}} \frac{1}{(2\pi\sigma^2)^{d/2}} \int \frac{\mathbf{z}}{\sigma^2} f(\mathbf{x}+\mathbf{z}) e^{-\|\mathbf{z}\|^2/2\sigma^2} d\mathbf{z}$$

$$= \frac{1}{(2\pi\sigma^2)^{d/2}} \int \frac{\partial}{\partial \mathbf{x}} \frac{\mathbf{z}-\mathbf{x}}{\sigma^2} f(\mathbf{z}) e^{-\|\mathbf{z}-\mathbf{x}\|^2/2\sigma^2} d\mathbf{z}$$

$$= \frac{1}{(2\pi\sigma^2)^{d/2}} \int f(\mathbf{z})(\frac{(\mathbf{z}-\mathbf{x})(\mathbf{z}-\mathbf{x})^\top}{\sigma^4} e^{-\|\mathbf{z}-\mathbf{x}\|^2/2\sigma^2} - \frac{\mathbf{I}}{\sigma^2} e^{-\|\mathbf{z}-\mathbf{x}\|^2/2\sigma^2}) d\mathbf{z}$$

$$= \frac{1}{(2\pi\sigma^2)^{d/2}} \int f(\mathbf{z}+\mathbf{x})(\frac{\mathbf{z}\mathbf{z}^\top - \sigma^2 \mathbf{I}}{\sigma^4}) e^{-\|\mathbf{z}\|^2/2\sigma^2} d\mathbf{z} = \mathbb{E}_{\mathbf{z}}[\frac{\mathbf{z}\mathbf{z}^\top - \sigma^2 \mathbf{I}}{\sigma^4} f(\mathbf{x}+\mathbf{z})].$$

$\square$

### A.1.1 Gradient Lipschitz

We bound the gradient Lipschitz constant of $\tilde{f}_\sigma$ in the following lemma.

**Lemma 3** (Gradient Lipschitz of $\tilde{f}_\sigma$). $\|\nabla^2 \tilde{f}_\sigma(\mathbf{x})\| \leq O(\ell + \frac{\nu}{\sigma^2})$.

*Proof.* For a twice-differentiable function, its gradient Lipschitz constant is also the upper bound on the spectral norm of its Hessian.

$$\|\nabla^2 \tilde{f}_\sigma(\mathbf{x})\| = \|\nabla^2 \tilde{F}_\sigma(\mathbf{x}) + \nabla^2 \tilde{f}_\sigma(\mathbf{x}) - \nabla^2 \tilde{F}_\sigma(\mathbf{x})\|$$

$$\leq \|\nabla^2 \tilde{F}_\sigma(\mathbf{x})\| + \|\nabla^2 \tilde{f}_\sigma(\mathbf{x}) - \nabla^2 \tilde{F}_\sigma(\mathbf{x})\|$$

$$= \|\nabla^2 \mathbb{E}_{\mathbf{z}}[F(\mathbf{x}+\mathbf{z})]\| + \|\mathbb{E}_z[\frac{\mathbf{z}\mathbf{z}^\top - \sigma^2 \mathbf{I}}{\sigma^4}(f-F)(\mathbf{x}+\mathbf{z})]\|$$

$$\leq \mathbb{E}_{\mathbf{z}}\|\nabla^2[F(\mathbf{x}+\mathbf{z})]\| + \frac{1}{\sigma^4}\|\mathbb{E}_z[\mathbf{z}\mathbf{z}^\top(f-F)(\mathbf{x}+\mathbf{z})]\| + \frac{1}{\sigma^2}\|\mathbb{E}_z[(f-F)(\mathbf{x}+\mathbf{z})\mathbf{I}]\|$$

$$\leq \ell + \frac{1}{\sigma^4}\|\mathbb{E}_z[\mathbf{z}\mathbf{z}^\top|(f-F)(\mathbf{x}+\mathbf{z})|]\| + \frac{1}{\sigma^2}\|\mathbb{E}_z[|(f-F)(\mathbf{x}+\mathbf{z})|\mathbf{I}]\|$$

$$= \ell + \frac{\nu}{\sigma^4}\|\mathbb{E}_z[\mathbf{z}\mathbf{z}^\top\| + \frac{\nu}{\sigma^2} = \ell + \frac{2\nu}{\sigma^2}$$

The last inequality follows from Lemma 9. $\square$

### A.1.2 Hessian Lipschitz

We bound the Hessian Lipschitz constant of $\tilde{f}_\sigma$ in the following lemma.

**Lemma 4** (Hessian Lipschitz of $\tilde{f}_\sigma$). $\|\nabla^2 \tilde{f}_\sigma(\mathbf{x}) - \nabla^2 \tilde{f}_\sigma(\mathbf{y})\| \leq O(\rho + \frac{\nu}{\sigma^3})\|\mathbf{x}-\mathbf{y}\|$.

*Proof.* By triangle inequality:

$$\|\nabla^2 \tilde{f}_\sigma(\mathbf{x}) - \nabla^2 \tilde{f}_\sigma(\mathbf{y})\|$$

$$= \|\nabla^2 \tilde{f}_\sigma(\mathbf{x}) - \nabla^2 \tilde{F}_\sigma(\mathbf{x}) - \nabla^2 \tilde{f}_\sigma(\mathbf{y}) + \nabla^2 \tilde{F}_\sigma(\mathbf{y}) + \nabla^2 \tilde{F}_\sigma(\mathbf{x}) - \nabla^2 \tilde{F}_\sigma(\mathbf{y})\|$$

$$\leq \|\nabla^2 \tilde{f}_\sigma(\mathbf{x}) - \nabla^2 \tilde{F}_\sigma(\mathbf{x}) - (\nabla^2 \tilde{f}_\sigma(\mathbf{y}) - \nabla^2 \tilde{F}_\sigma(\mathbf{y}))\| + \|\nabla^2 \tilde{F}_\sigma(\mathbf{x}) - \nabla^2 \tilde{F}_\sigma(\mathbf{y})\|$$

$$\leq O(\frac{\nu}{\sigma^3})\|\mathbf{x}-\mathbf{y}\| + O(\rho)\|\mathbf{x}-\mathbf{y}\| + O(\|\mathbf{x}-\mathbf{y}\|^2)$$

The last inequality follows from Lemma 10 and 11. $\square$

### A.1.3 Gradient Difference

We bound the difference between the gradients of smoothed function $\tilde{f}_\sigma(\mathbf{x})$ and those of the true objective $F$.

**Lemma 5** (Gradient Difference). $\|\nabla \tilde{f}_\sigma(\mathbf{x}) - \nabla F(\mathbf{x})\| \leq O(\frac{\nu}{\sigma} + \rho d \sigma^2)$.

*Proof.* By triangle inequality:

$$\|\nabla \tilde{f}_\sigma(\mathbf{x}) - \nabla F(\mathbf{x})\| \leq \|\nabla \tilde{f}_\sigma(\mathbf{x}) - \nabla \tilde{F}_\sigma(\mathbf{x})\| + \|\nabla \tilde{F}_\sigma(\mathbf{x}) - \nabla F(\mathbf{x})\|.$$

Then the result follows from Lemma 13 and 14 $\qquad\square$

### A.1.4 Hessian Difference

We bound the difference between the Hessian of smoothed function $\tilde{f}_\sigma(\mathbf{x})$ and that of the true objective $F$.

**Lemma 6** (Hessian Difference). $\|\nabla^2 \tilde{f}_\sigma(\mathbf{x}) - \nabla^2 F(\mathbf{x})\| \leq O(\rho\sqrt{d}\sigma + \frac{\nu}{\sigma^2})$.

*Proof.* By triangle inequality:

$$
\begin{aligned}
\|\nabla^2 \tilde{f}_\sigma(\mathbf{x}) - \nabla^2 F(\mathbf{x})\| &\leq \|\nabla^2 \tilde{F}_\sigma(\mathbf{x}) - \nabla^2 F(\mathbf{x})\| + \|\nabla^2 \tilde{f}_\sigma(\mathbf{x}) - \nabla^2 \tilde{F}_\sigma(\mathbf{x})\| \\
&\leq \mathbb{E}_{\mathbf{z}}\|\nabla^2 F(\mathbf{x} + \mathbf{z}) - \nabla^2 F(\mathbf{x})\| + \frac{2\nu}{\sigma^2} \\
&\leq \mathbb{E}_{\mathbf{z}}\|\rho\mathbf{z}\| + \frac{2\nu}{\sigma^2} \leq \rho\sqrt{d}\sigma + \frac{2\nu}{\sigma^2}
\end{aligned}
$$

The first inequality follows exactly from the proof of lemma 3. The second equality follows from the definition of Hessian Lipschitz. The third inequality follows from $\mathbb{E}_{\mathbf{z}}\|\rho\mathbf{z}\| \leq \rho\sqrt{\mathbb{E}[\|\mathbf{z}\|^2]}$. $\qquad\square$

## A.2 Properties of the stochastic gradient

We prove the properties of the stochastic gradient, $\mathbf{g}(\mathbf{x}; \mathbf{z})$, as stated in Lemma 7, restated as follows. Intuitively this lemma shows that the stochastic gradient is well-behaved and can be used in the standard algorithms.

**Lemma 7** (Property of stochastic gradient). *Let* $\mathbf{g}(\mathbf{x}; \mathbf{z}) = \mathbf{z}[f(\mathbf{x} + \mathbf{z}) - f(\mathbf{x})]/\sigma^2$, *where* $\mathbf{z} \sim \mathcal{N}(0, \sigma^2\mathbf{I})$. *Then* $\mathbb{E}_{\mathbf{z}}\mathbf{g}(\mathbf{x}; \mathbf{z}) = \nabla \tilde{f}_\sigma(\mathbf{x})$, *and* $\mathbf{g}(\mathbf{x}; \mathbf{z})$ *is sub-Gaussian with parameter* $\frac{B}{\sigma}$.

*Proof.* The first part follows from Lemma 2. Given any $\mathbf{u} \in \mathbb{R}^d$, by assumption A1 ($f$ is $B$-bounded),

$$|\langle \mathbf{u}, \mathbf{g}(\mathbf{x}; \mathbf{z})\rangle| = |(f(\mathbf{x} + \mathbf{z}) - f(\mathbf{x}))||\langle \mathbf{u}, \frac{\mathbf{z}}{\sigma^2}\rangle| \leq \frac{B\|u\|}{\sigma}|\langle \frac{\mathbf{u}}{\|u\|}, \frac{\mathbf{z}}{\sigma}\rangle|.$$

Note that $\langle \frac{\mathbf{u}}{\|\mathbf{u}\|}, \frac{\mathbf{z}}{\sigma}\rangle \sim \mathcal{N}(0, 1)$. Thus, for $X \sim \mathcal{N}(0, \frac{B\|u\|}{\sigma})$,

$$\mathbb{P}(|\langle \mathbf{u}, \mathbf{g}(\mathbf{x}; \mathbf{z})\rangle| > s) \leq \mathbb{P}(|X| > s).$$

This shows that $\mathbf{g}$ is sub-Gaussian with parameter $\frac{B}{\sigma}$.

$\qquad\square$

## A.3 Proof of Theorem 7: SOSP of $\tilde{f}_\sigma$ are also SOSP of $F$

Using the properties proven in Lemma 7, we can apply Theorem 15 to find an $\tilde{\epsilon}$-SOSP for $\tilde{f}_\sigma$ for any $\tilde{\epsilon}$. The running time of the algorithm is polynomial as long as $\tilde{\epsilon}$ depends polynomially on the relevant parameters. Now we will show that every $\tilde{\epsilon}$-SOSP of $\tilde{f}_\sigma$ is an $O(\epsilon)$-SOSP of $F$ when $\epsilon'$ is small enough.

More precisely, we use Lemma 13 to show that any $\frac{\epsilon}{\sqrt{d}}$-SOSP of $\tilde{f}_\sigma(\mathbf{x})$ is also an $O(\epsilon)$-SOSP of $F$.

**Lemma 8** (SOSP of $\tilde{f}_\sigma(\mathbf{x})$ and SOSP of $F(\mathbf{x})$)**.** *Suppose $\mathbf{x}^*$ satisfies*
$$\|\nabla \tilde{f}_\sigma(\mathbf{x}^*)\| \leq \tilde{\epsilon} \text{ and } \lambda_{min}(\nabla^2 \tilde{f}_\sigma(\mathbf{x}^*)) \geq -\sqrt{\tilde{\rho}\tilde{\epsilon}},$$
*where $\tilde{\rho} = \rho + \frac{\nu}{\sigma^3}$ and $\tilde{\epsilon} = \epsilon/\sqrt{d}$. Then there exists constants $c_1, c_2$ such that*
$$\sigma \leq c_1 \sqrt{\frac{\epsilon}{\rho d}}, \ \nu \leq c_2 \sqrt{\frac{\epsilon^3}{\rho d^2}}.$$
*implies $\mathbf{x}^*$ is an $O(\epsilon)$-SOSP of $F$.*

*Proof.* By applying Lemma 13 and Weyl's inequality, we have that the following inequalities hold up to a constant factor:
$$\|\nabla F(\mathbf{x}^*)\| \leq \rho d \sigma^2 + \frac{\nu}{\sigma} + \tilde{\epsilon}$$
$$\lambda_{\min}(\nabla^2 F(\mathbf{x}^*)) \geq \lambda_{\min}(\nabla^2 \tilde{f}_\sigma(\mathbf{x}^*)) + \lambda_{\min}(\nabla^2 F(\mathbf{x}^*) - \nabla^2 \tilde{f}_\sigma(\mathbf{x}^*))$$
$$\geq -\sqrt{(\rho + \frac{\nu}{\sigma^3})\tilde{\epsilon}} - \|\nabla^2 \tilde{f}_\sigma(\mathbf{x}) - \nabla^2 F(\mathbf{x})\|$$
$$= -\sqrt{\frac{(\rho + \frac{\nu}{\sigma^3})}{\sqrt{d}}\epsilon} - (\rho\sqrt{d}\sigma + \frac{\nu}{\sigma^2})$$

Suppose we want any $\tilde{\epsilon}$-SOSP of $\tilde{f}_\sigma(\mathbf{x})$ to be a $O(\epsilon)$-SOSP of $F$. Then satisfying the following inequalities is sufficient (up to a constant factor):
$$\rho\sqrt{d}\sigma + \frac{\nu}{\sigma^2} \leq \sqrt{\rho\epsilon} \tag{1}$$
$$\rho d\sigma^2 + \frac{\nu}{\sigma} \leq \epsilon \tag{2}$$
$$\rho + \frac{\nu}{\sigma^3} \leq \rho\sqrt{d} \tag{3}$$

We know Eq.(1), (2) $\implies \sigma \leq \frac{\sqrt{\rho\epsilon}}{\rho\sqrt{d}} = \sqrt{\frac{\epsilon}{\rho d}}$ and $\sigma \leq \sqrt{\frac{\epsilon}{\rho d}}$.

Also Eq. (1), (2) $\implies \nu \leq \sigma\epsilon \leq \sqrt{\frac{\epsilon^3}{\rho d}}$ and $\nu \leq \sqrt{\rho\epsilon}\sigma^2 \leq \sqrt{\rho\epsilon}\frac{\epsilon}{\rho d} = \sqrt{\frac{\epsilon^3}{\rho d^2}}$.

Finally Eq.(3) $\implies \nu \leq \rho\sqrt{d}\sigma^3 \leq \frac{\epsilon^{1.5}}{\rho^{0.5}d}$.

Thus the following choice of $\sigma$ and $\nu$ ensures that $\mathbf{x}^*$ is an $O(\epsilon)$-SOSP of $F$:
$$\sigma \leq c_1 \sqrt{\frac{\epsilon}{\rho d}}, \ \nu \leq c_2 \sqrt{\frac{\epsilon^3}{\rho d^2}},$$
where $c_1, c_2$ are universal constants. $\qquad\square$

*Proof of Theorem 7.* Applying Theorem 15 on $\tilde{f}_\sigma(\mathbf{x})$ guarantees finding an $c\frac{\epsilon}{\sqrt{d}}$-SOSP of $\tilde{f}_\sigma(\mathbf{x})$ in number of queries polynomial in all the problem parameters. By Lemma 8, for some universal constant $c$, this is also an $\epsilon$-SOSP of $F$. This proves Theorem 7. $\qquad\square$

### A.4 Technical Lemmas

In this section, we collect and prove the technical lemmas used in the proofs of the above.

**Lemma 9.** *Let $\lambda$ be a real-valued random variable and $A$ be a random PSD matrix that can depend on $\lambda$. Denote the matrix spectral norm as $\|\cdot\|$. Then $\|\mathbb{E}[A\lambda]\| \leq \|\mathbb{E}[A|\lambda|]\|$.*

*Proof.* For any $\mathbf{x} \in \mathbb{R}^d$,
$$\mathbf{x}^\top \mathbb{E}[A\lambda]\mathbf{x} = \mathbf{x}^\top \mathbb{E}[A\lambda|\lambda \geq 0]\mathbf{x} \cdot \mathbb{P}(\lambda \geq 0) + \mathbf{x}^\top \mathbb{E}[A\lambda|\lambda < 0]\mathbf{x} \cdot \mathbb{P}(\lambda < 0)$$
$$\leq \mathbf{x}^\top \mathbb{E}[A\lambda|\lambda \geq 0]\mathbf{x} \cdot \mathbb{P}(\lambda \geq 0) - \mathbf{x}^\top \mathbb{E}[A\lambda|\lambda < 0]\mathbf{x} \cdot \mathbb{P}(\lambda < 0)$$
$$= \mathbf{x}^\top \mathbb{E}[A|\lambda|]\mathbf{x}$$

$\qquad\square$

The following two technical lemmas bound the Hessian Lipschitz constants of $\tilde{F}_\sigma$ and $(\tilde{f}_\sigma - \tilde{F}_\sigma)$ respectively.

**Lemma 10.** $\|\nabla^2 \tilde{F}_\sigma(\mathbf{x}) - \nabla^2 \tilde{F}_\sigma(\mathbf{y})\| \leq \rho \|\mathbf{x} - \mathbf{y}\|.$

*Proof.* By the Hessian-Lipschitz property of $F$:

$$
\begin{aligned}
\|\nabla^2 \tilde{F}_\sigma(\mathbf{x}) - \nabla^2 \tilde{F}_\sigma(\mathbf{y})\| &= \|\mathbb{E}_{\mathbf{z}}[\nabla^2 F(\mathbf{x}+\mathbf{z}) - \nabla^2 F(\mathbf{y}+\mathbf{z})]\| \\
&\leq \mathbb{E}_{\mathbf{z}} \|\nabla^2 F(\mathbf{x}+\mathbf{z}) - \nabla^2 F(\mathbf{y}+\mathbf{z})\| \\
&\leq \rho \|\mathbf{x}-\mathbf{y}\|
\end{aligned}
$$

$\square$

**Lemma 11.** $\|\nabla^2 \tilde{f}_\sigma(\mathbf{x}) - \nabla^2 \tilde{F}_\sigma(\mathbf{x}) - (\nabla^2 \tilde{f}_\sigma(\mathbf{y}) - \nabla^2 \tilde{F}_\sigma(\mathbf{y}))\| \leq O(\frac{\nu}{\sigma^3})\|\mathbf{x}-\mathbf{y}\| + O(\|\mathbf{x}-\mathbf{y}\|^2).$

*Proof.* For brevity, denote $h = \frac{1}{(2\pi\sigma^2)^{\frac{d}{2}}}$.

$$
\begin{aligned}
&\nabla^2 \tilde{f}_\sigma(\mathbf{x}) - \nabla^2 \tilde{F}_\sigma(\mathbf{x}) - (\nabla^2 \tilde{f}_\sigma(\mathbf{y}) - \nabla^2 \tilde{F}_\sigma(\mathbf{y})) \\
&= \mathbb{E}_{\mathbf{z}}\left[\frac{\mathbf{z}\mathbf{z}^\top - \sigma^2 \mathbf{I}}{\sigma^4}((f-F)(\mathbf{x}+\mathbf{z}) - (f-F)(\mathbf{y}+\mathbf{z})]\right. \\
&= h\left(\int \frac{\mathbf{z}\mathbf{z}^\top - \sigma^2 \mathbf{I}}{\sigma^4}(f-F)(\mathbf{x}+\mathbf{z})e^{-\frac{\|\mathbf{z}\|^2}{2\sigma^2}}\,d\mathbf{z} - \int \frac{\mathbf{z}\mathbf{z}^\top - \sigma^2 \mathbf{I}}{\sigma^4}(f-F)(\mathbf{y}+\mathbf{z})e^{-\frac{\|\mathbf{z}\|^2}{2\sigma^2}}\,d\mathbf{z}\right) \\
&= h\int (f-F)(\mathbf{z}+\frac{\mathbf{x}+\mathbf{y}}{2})\,(\omega(\Delta) - \omega(-\Delta))\,d\mathbf{z}, \quad (4)
\end{aligned}
$$

where $\omega(\Delta) := \frac{(\mathbf{z}+\Delta)(\mathbf{z}+\Delta)^\top - \sigma^2 \mathbf{I}}{\sigma^4}e^{-\frac{\|\mathbf{z}+\Delta\|^2}{2\sigma^2}}$ and $\Delta = \frac{\mathbf{y}-\mathbf{x}}{2}$. Equality (4) follows from a change of variables. Now, denote $g(\mathbf{z}) := (f-F)(\mathbf{z}+\frac{\mathbf{x}+\mathbf{y}}{2})$.

Using $\omega(\Delta) = \frac{(\mathbf{z}+\Delta)(\mathbf{z}+\Delta)^\top - \sigma^2 \mathbf{I}}{\sigma^4}e^{-\frac{\|\Delta\|^2 + 2\langle\Delta, \mathbf{z}\rangle}{2\sigma^2}}e^{-\frac{\|\mathbf{z}\|^2}{2\sigma^2}}$, we have the following

$$
h\int g(\mathbf{z})\,(\omega(\Delta) - \omega(-\Delta))\,d\mathbf{z} = \mathbb{E}_{\mathbf{z}}\left[g(\mathbf{z})\left(\omega(\Delta)e^{\frac{\|\mathbf{z}\|^2}{2\sigma^2}} - \omega(-\Delta)e^{\frac{\|\mathbf{z}\|^2}{2\sigma^2}}\right)\right].
$$

By a Taylor expansion up to only the first order terms in $\Delta$,

$$
\omega(\Delta)e^{\frac{\|\mathbf{z}\|^2}{2\sigma^2}} = \frac{\mathbf{z}\mathbf{z}^\top + \Delta\mathbf{z}^\top + \mathbf{z}\Delta^\top - \sigma^2 \mathbf{I}}{\sigma^4}(1 + \frac{1}{\sigma^2}\langle\Delta, \mathbf{z}\rangle).
$$

We then write the Taylor expansion of $\mathbb{E}_{\mathbf{z}}\left[g(\mathbf{z})\left(\omega(\Delta)e^{\frac{\|\mathbf{z}\|^2}{2\sigma^2}} - \omega(-\Delta)e^{\frac{\|\mathbf{z}\|^2}{2\sigma^2}}\right)\right]$ as follows.

$$
\begin{aligned}
&\mathbb{E}_{\mathbf{z}}[g(\mathbf{z})\cdot \frac{\mathbf{z}\mathbf{z}^\top - \sigma^2 \mathbf{I}}{\sigma^4}\cdot \frac{2}{\sigma^2}\langle\Delta, \mathbf{z}\rangle + g(\mathbf{z})\cdot 2\cdot \frac{\Delta\mathbf{z}^\top + \mathbf{z}\Delta^\top}{\sigma^4}] \\
&= \mathbb{E}_{\mathbf{z}}[g(\mathbf{z})\cdot \frac{\mathbf{z}\mathbf{z}^\top}{\sigma^4}\cdot \frac{2}{\sigma^2}\langle\Delta, \mathbf{z}\rangle] - \mathbb{E}_{\mathbf{z}}[g(\mathbf{z})\cdot \frac{\sigma^2 \mathbf{I}}{\sigma^4}\cdot \frac{2}{\sigma^2}\langle\Delta, \mathbf{z}\rangle] + \mathbb{E}_{\mathbf{z}}[g(\mathbf{z})\cdot 2\cdot \frac{\Delta\mathbf{z}^\top + \mathbf{z}\Delta^\top}{\sigma^4}].
\end{aligned}
$$

Therefore,

$$
\begin{aligned}
&\|\nabla^2 \tilde{f}_\sigma(\mathbf{x}) - \nabla^2 \tilde{F}_\sigma(\mathbf{x}) - (\nabla^2 \tilde{f}_\sigma(\mathbf{y}) - \nabla^2 \tilde{F}_\sigma(\mathbf{y}))\| \\
\leq & \|\mathbb{E}_{\mathbf{z}}[g(\mathbf{z})\cdot \frac{\mathbf{z}\mathbf{z}^\top}{\sigma^4}\cdot \frac{2}{\sigma^2}\langle\Delta, \mathbf{z}\rangle]\| + \|\mathbb{E}_{\mathbf{z}}[g(\mathbf{z})\cdot \frac{\mathbf{I}}{\sigma^4}\cdot 2\langle\Delta, \mathbf{z}\rangle]\| + \|\mathbb{E}_{\mathbf{z}}[g(\mathbf{z})\cdot 2\cdot \frac{\Delta\mathbf{z}^\top + \mathbf{z}\Delta^\top}{\sigma^4}]\| + O(\|\Delta\|^2) \\
= & \frac{2}{\sigma^6}\|\mathbb{E}_{\mathbf{z}}[g(\mathbf{z})\cdot \mathbf{z}\mathbf{z}^\top\langle\Delta, \mathbf{z}\rangle]\| + \frac{2}{\sigma^4}\|\mathbb{E}_{\mathbf{z}}[g(\mathbf{z})\langle\Delta, \mathbf{z}\rangle\mathbf{I}]\| + \frac{2}{\sigma^4}\|\mathbb{E}_{\mathbf{z}}[g(\mathbf{z})(\Delta\mathbf{z}^\top + \mathbf{z}\Delta^\top)]\| + O(\|\Delta\|^2) \\
\leq & O(\frac{\nu}{\sigma^3})\|\Delta\| + O(\|\Delta\|^2).
\end{aligned}
$$

The last inequality follows from Lemma 12.

$\square$

**Lemma 12.** *Given* $\mathbf{z} \sim N(0, \sigma \mathbf{I}_{d \times d})$, *some* $\Delta \in \mathbb{R}^d$, $\|\Delta\| = 1$, *and* $f : \mathbb{R}^d \to [-1, 1]$,

    1. $\|\mathbb{E}_{\mathbf{z}}[f(\mathbf{z}) \cdot \mathbf{z}\mathbf{z}^\top \langle \Delta, \mathbf{z} \rangle]\| = O(\sigma^3)$;                 2. $\|\mathbb{E}_{\mathbf{z}}[f(\mathbf{z})\langle \Delta, \mathbf{z} \rangle \mathbf{I}]\| = O(\sigma)$;

    3. $\|\mathbb{E}_{\mathbf{z}}[f(\mathbf{z})(\Delta \mathbf{z}^\top)]\| = O(\sigma)$;                         4. $\|\mathbb{E}_{\mathbf{z}}[f(\mathbf{z})(\mathbf{z}\Delta^\top)]\| = O(\sigma)$.

*Proof.* For the first inequality:

$$\|\mathbb{E}_{\mathbf{z}}[f(\mathbf{z}) \cdot \mathbf{z}\mathbf{z}^\top \langle \Delta, \mathbf{z} \rangle]\| = \sup_{\mathbf{v} \in \mathbb{R}^d, \|\mathbf{v}\|=1} \mathbb{E}[\mathbf{v}^\top f(\mathbf{z}) \cdot \mathbf{z}\mathbf{z}^\top \langle \Delta, \mathbf{z} \rangle \mathbf{v}]$$

$$= \sup_{\mathbf{v} \in \mathbb{R}^d, \|\mathbf{v}\|=1} \mathbb{E}[f(\mathbf{z})(\mathbf{v}^\top \mathbf{z})^2(\Delta^\top \mathbf{z})] \leq \sup_{\mathbf{v}, \Delta \in \mathbb{R}^d, \|\mathbf{v}\|=\|\Delta\|=1} \mathbb{E}[f(\mathbf{z})(\mathbf{v}^\top \mathbf{z})^2(\Delta^\top \mathbf{z})]$$

$$\leq \sup_{\mathbf{v}, \Delta \in \mathbb{R}^d, \|\mathbf{v}\|=\|\Delta\|=1} \mathbb{E}[(\mathbf{v}^\top \mathbf{z})^2 |\Delta^\top \mathbf{z}|] \leq \sup_{\mathbf{v}, \Delta \in \mathbb{R}^d, \|\mathbf{v}\|=\|\Delta\|=1} \mathbb{E}[|\mathbf{v}^\top \mathbf{z}|^3 + |\Delta^\top \mathbf{z}|^3]$$

$$= 2\mathbb{E}[|\mathbf{v}^{*\top} \mathbf{z}|^3] = 4\sqrt{\frac{2}{\pi}}\sigma^3.$$

For the second inequality:

$$\|\mathbb{E}_{\mathbf{z}}[f(\mathbf{z})\langle \Delta, \mathbf{z} \rangle \mathbf{I}]\| = |\mathbb{E}[X\tilde{f}(X)]| \leq \mathbb{E}|X| = \sqrt{\frac{2}{\pi}}\sigma,$$

where $X = \langle \Delta, \mathbf{z} \rangle \sim N(0, \sigma^2)$ and $\tilde{f}(a) = \mathbb{E}[f(\mathbf{z})|X = a] \in [-1, 1]$.
For the third inequality:

$$\|\mathbb{E}_{\mathbf{z}}[f(\mathbf{z})(\Delta \mathbf{z}^\top)]\| = \sup_{\mathbf{v} \in \mathbb{R}^d, \|\mathbf{v}\|=1} \mathbb{E}_{\mathbf{z}}[f(\mathbf{z})\mathbf{v}^\top \Delta \mathbf{z}^\top \mathbf{v})]$$

$$= \mathbf{v}^{*\top}\Delta \mathbb{E}_{\mathbf{z}}[f(\mathbf{z})\mathbf{z}^\top \mathbf{v}^*)] \leq \sqrt{\frac{2}{\pi}}\sigma,$$

where the last step is correct due to the second inequality we proved. The proof of the fourth inequality directly follows from the third inequality. $\qquad\square$

**Lemma 13.** $\|\nabla \tilde{f}_\sigma(\mathbf{x}) - \nabla \tilde{F}_\sigma(\mathbf{x})\| \leq \sqrt{\frac{2}{\pi}}\frac{\nu}{\sigma}$.

*Proof.* By the Gaussian smoothing identity,

$$\|\nabla \tilde{f}_\sigma(\mathbf{x}) - \nabla \tilde{F}_\sigma(\mathbf{x})\| = \|\mathbb{E}_{\mathbf{z}}[\frac{\mathbf{z}}{\sigma^2}(f - F)(\mathbf{x} - \mathbf{z})]\| \leq \sqrt{\frac{2}{\pi}}\frac{\nu}{\sigma}.$$

The last inequality follows from Lemma 15. $\qquad\square$

**Lemma 14.** $\|\nabla \tilde{F}_\sigma(\mathbf{x}) - \nabla F(\mathbf{x})\| \leq \rho d\sigma^2$.

*Proof.* By definition of Gaussian smoothing,

$$\|\nabla \tilde{F}_\sigma(\mathbf{x}) - \nabla F(\mathbf{x})\|$$

$$= \|\nabla \mathbb{E}_{\mathbf{z}}[F(\mathbf{x} - \mathbf{z})] - \nabla F(\mathbf{x})\| \leq \|\mathbb{E}_{\mathbf{z}}[\left(\int_0^1 \nabla^2 f(\mathbf{x} + t\mathbf{z})dt\right)\mathbf{z}]\| \qquad (5)$$

$$= \|\mathbb{E}_{\mathbf{z}}[\left(\int_0^1 \nabla^2 f(\mathbf{x}) + \nabla^2 f(\mathbf{x} + t\mathbf{z}) - \nabla^2 f(\mathbf{x})dt\right)\mathbf{z}]\|$$

$$\leq \|\mathbb{E}_{\mathbf{z}}[\nabla^2 f(\mathbf{x})\mathbf{z}]\| + \|\mathbb{E}_{\mathbf{z}}[\left(\int_0^1 \nabla^2 f(\mathbf{x} + t\mathbf{z}) - \nabla^2 f(\mathbf{x})dt\right)\mathbf{z}]\|$$

$$\leq \mathbb{E}_{\mathbf{z}}[\left(\int_0^1 \|\nabla^2 f(\mathbf{x} + t\mathbf{z}) - \nabla^2 f(\mathbf{x})\|dt\right)\|\mathbf{z}\|]$$

$$\leq \mathbb{E}_{\mathbf{z}}[\left(\int_0^1 \rho\|t\mathbf{z}\|dt\right)\|\mathbf{z}\|] = \rho\|z\|^2 \leq \rho d\sigma^2.$$

Inequality (5) follows by applying a generalization of mean-value theorem to vector-valued functions. $\qquad\square$

**Lemma 15.** *Given* $\mathbf{z} \sim N(0, \sigma \mathbf{I}_{d \times d})$ *and* $f : \mathbb{R}^d \to [-1, 1]$,

$$\|\mathbb{E}\mathbf{z}f(\mathbf{z})\| \leq \sqrt{\frac{2}{\pi}}\sigma.$$

*Proof.* By definition of the 2-norm,

$$
\begin{aligned}
\|\mathbb{E}\mathbf{z}f(\mathbf{z})\| &= \sup_{\mathbf{v} \in \mathbb{R}^d, \|\mathbf{v}\|=1} \mathbb{E}[\mathbf{v}^\top \mathbf{z}f(\mathbf{z})] = \mathbb{E}[\mathbf{v}^{*\top}\mathbf{z}f(\mathbf{z})] \\
&= \mathbb{E}[\mathbb{E}[Xf(\mathbf{z})|X]] \qquad \text{where } X = \mathbf{v}^{*\top}\mathbf{z} \sim N(0, \sigma^2) \\
&= \mathbb{E}[X\tilde{f}(X)] \qquad \text{where } \tilde{f}(a) = \mathbb{E}[f(\mathbf{z})|X = a] \in [-1, 1] \\
&\leq \mathbb{E}|X| = \sqrt{\frac{2}{\pi}}\sigma.
\end{aligned}
$$

$\square$

# B    Overview for polynomial queries lower bound

In this section, we discuss the key ideas for proving Theorem 8. We illustrate the construction in two steps: (1) construct a hard instance $(F, f)$ contained in a $d$-dimensional ball; (2) extend this hard instance to $\mathbb{R}^n$. The second step is necessary as Problem 1 is an unconstrained problem; in non-convex optimization the hardness of optimizing unconstrained problems and optimizing constrained problems can be very different. For simplicity, in this section we assume $\rho, \epsilon$ are both 1 and focus on the $d$ dependencies, to highlight the difference between polynomial queries and the information-theoretic limit. The general result involving dependency on $\epsilon$ and $\rho$ follows from a simple scaling of the hard functions.

Figure 1: Key regions in lower bound

Figure 2: Landscape of $h$

**Constructing a lower-bound example within a ball**    The target function $F(\mathbf{x})$ we construct contains a special direction $\mathbf{v}$ in a $d$-dimensional ball $\mathbb{B}_r$ with radius $r$ centered at the origin. More concretely, let $F(\mathbf{x}) = h(\mathbf{x}) + \|\mathbf{x}\|^2$, where $h$ (see Figure 2) depends on a special direction $\mathbf{v}$, but is spherically symmetric in its orthogonal subspace. Let the direction $\mathbf{v}$ be sampled uniformly at random from the $d$-dimensional unit sphere. Define a region around the equator of $\mathbb{B}_r$, denoted $S_{\mathbf{v}} = \{\mathbf{x}|\mathbf{x} \in \mathbb{B}_r \text{ and } |\mathbf{v}^\top \mathbf{x}| \leq r \log d/\sqrt{d}\}$, as in Figure 1. The key ideas of this construction relying on the following three properties:

1. For any fixed point $\mathbf{x}$ in $\mathbb{B}_r$, we have $\Pr(\mathbf{x} \in S_{\mathbf{v}}) \geq 1 - O(1/d^{\log d})$.
2. The $\epsilon$-SOSP of $F$ is located in a very small set $\mathbb{B}_r - S_{\mathbf{v}}$.
3. $h(\mathbf{x})$ has very small function value inside $S_{\mathbf{v}}$, that is, $\sup_{\mathbf{x} \in S_{\mathbf{v}}} |h(\mathbf{x})| \leq \tilde{O}(1/d)$.

The first property is due to the concentration of measure in high dimensions. The latter two properties are intuitively shown in Figure 2. These properties suggest a natural construction for $f$:

$$f(\mathbf{x}) = \begin{cases} \|\mathbf{x}\|^2 & \text{if } \mathbf{x} \in S_{\mathbf{v}} \\ F(\mathbf{x}) & \text{otherwise} \end{cases}.$$

When $\mathbf{x} \in S_{\mathbf{v}}$, by property 3 above we know $|f(\mathbf{x}) - F(\mathbf{x})| \leq \nu = \tilde{O}(1/d)$.

To see why this construction gives a hard instance of Problem 1, recall that the direction $\mathbf{v}$ is uniformly random. Since the direction $\mathbf{v}$ is unknown to the algorithm at initialization, the algorithm's first query is independent of $\mathbf{v}$ and thus is likely to be in region $S_{\mathbf{v}}$, due to property 1. The queries inside $S_{\mathbf{v}}$ give no information about $\mathbf{v}$, so any polynomial-time algorithm is likely to continue to make queries in $S_{\mathbf{v}}$ and eventually fail to find $\mathbf{v}$. On the other hand, by property 2 above, finding an $\epsilon$-SOSP of $F$ requires approximately identifying the direction of $\mathbf{v}$, so any polynomial-time algorithm will fail with high probability.

**Extending to the entire space**   To extend this construction to the entire space $\mathbb{R}^d$, we put the ball (the previous construction) inside a hypercube (see Figure 1) and use the hypercube to tile the entire space $\mathbb{R}^d$. There are two challenges in this approach: (1) The function $F$ must be smooth even at the boundaries between hypercubes; (2) The padding region ($S_2$ in Figure 1) between the ball and the hypercube must be carefully constructed to not ruin the properties of the hard functions.

We deal with first problem by constructing a function $\bar{F}(\mathbf{y})$ on $[-1, 1]^d$, ignoring the boundary condition, and then composing it with a smooth periodic function. For the second problem, we carefully construct a smooth function $h$, as shown in Figure 2, to have zero function value, gradient and Hessian at the boundary of the ball and outside the ball, so that no algorithm can make use of the padding region to identify an SOSP of $F$. Details are deferred to section C in the appendix.

## C  Constructing Hard Functions

In this section, we prove Theorem 8, the lower bound for algorithms making a polynomial number of queries. We start by describing the hard function construction that is key to the lower bound.

### C.1  "Scale-free" hard instance

We will first present a "scale-free" version of the hard function, where we assume $\rho = 1$ and $\epsilon = 1$. In section C.2, we will show how to scale this hard function to prove Theorem 8.

Denote $\sin \mathbf{x} = (\sin(x_1), \cdots, \sin(x_d))$. Let $\mathbb{I}\{a\}$ denote the indicator function that takes value 1 when event $a$ happens and 0 otherwise. Let $\mu = 300$ . Let the function $F : \mathbb{R}^d \to \mathbb{R}$ be defined as follows.

$$F(\mathbf{x}) = h(\sin \mathbf{x}) + \|\sin \mathbf{x}\|^2, \tag{6}$$

where $h(\mathbf{y}) = h_1(\mathbf{v}^\top \mathbf{y}) \cdot h_2(\sqrt{\|\mathbf{y}\|^2 - (\mathbf{v}^\top \mathbf{y})^2})$, and

$$h_1(x) = g_1(\mu x), \quad g_1(x) = \left(-16|x|^5 + 48x^4 - 48|x|^3 + 16x^2\right) \mathbb{I}\{|x| < 1\},$$
$$h_2(x) = g_2(\mu x), \quad g_2(x) = \left(3x^4 - 8|x|^3 + 6x^2 - 1\right) \mathbb{I}\{|x| < 1\},$$

and the vector $\mathbf{v}$ is uniformly distributed on the $d$-dimensional unit sphere.

We will state the properties of the hard instance by breaking the space into different regions:

- "ball" $S = \{\mathbf{x} \in \mathbb{R}^d : \|\mathbf{x}\| \leq 3/\mu\}$ be the $d$-dimensional ball with radius $3/\mu$.
- "hypercube" $H = [-\frac{\pi}{2}, \frac{\pi}{2}]^d$ be the $d$-dimensional hypercube with side length $\pi$.
- "band" $S_{\mathbf{v}} = \{\mathbf{x} \in S : \langle \sin \mathbf{x}, \mathbf{v} \rangle \leq \frac{\log d}{\sqrt{d}}\}$
- "padding" $S_2 = H - S$

We also call the union of $S_2$ and $S_{\mathbf{v}}$ the "non-informative" region.

Figure 3: Polynomials $g_1, g_2$

Define the perturbed function, $f$:

$$f(\mathbf{x}) = \begin{cases} \|\sin \mathbf{x}\|^2, & \mathbf{x} \in S_{\mathbf{v}} \\ F(\mathbf{x}), & \mathbf{x} \notin S_{\mathbf{v}}. \end{cases} \tag{7}$$

Our construction happens within the ball. However it is hard to fill the space using balls, so we pad the ball into a hypercube. Our construction will guarantee that any queries to the non-informative region do not reveal any information about $\mathbf{v}$. Intuitively the non-informative region is very large so that it is hard for the algorithm to find any point outside of the non-informative region (and learn any information about $\mathbf{v}$).

**Lemma 16** (Properties of scale-free hard function pair $F, f$)**.** *Let $F, f, \mathbf{v}$ be as defined in equations* (6), (7)*. Then $F, f$ satisfies:*

1. *$f$ in the non-informative region $S_2 \cup S_{\mathbf{v}}$ is independent of $\mathbf{v}$.*

2. *$\sup_{x \in S_{\mathbf{v}}} |f - F| \leq \tilde{O}(\frac{1}{d})$.*

3. *$F$ has no SOSP in the non-informative region $S_2 \cup S_{\mathbf{v}}$.*

4. *$F$ is $O(d)$-bounded, $O(1)$-Hessian Lipschitz, and $O(1)$-gradient Lipschitz.*

These properties will be proved based on the properties of $h(\mathbf{y})$, which we defined in (6) to be the product of two functions.

*Proof.* **Property 1.** On $S_{\mathbf{v}}$, $f(\mathbf{x}) = F(\mathbf{x}) = \|\sin \mathbf{x}\|^2$, which is independent of $\mathbf{v}$. On $S_2$, we argue that $h(\sin \mathbf{x}) = 0$ and therefore $f(\mathbf{x}) = \|\sin \mathbf{x}\|^2 \ \forall \mathbf{x} \in S_2$. Note that on $S_2$, $\|\mathbf{x}\| > 3/\mu$ and $(\sin x)^2 > (\frac{2x}{\pi})^2 \ \forall |x| < \frac{\pi}{2}$, so

$$\|\sin \mathbf{x}\| > \|\frac{2\mathbf{x}}{\pi}\| > \frac{6}{\pi \mu} \implies \max \left\{ \mathbf{v}^\top \sin \mathbf{x}, \sqrt{\|\sin \mathbf{x}\|^2 - (\mathbf{v}^\top \sin \mathbf{x})^2} \right\} > \frac{6}{\sqrt{2} \cdot \pi \mu} > \frac{1}{\mu}.$$

Therefore, $h(\sin \mathbf{x}) = h_1(\mathbf{v}^\top \sin \mathbf{x}) \cdot h_2(\sqrt{\|\sin \mathbf{x}\|^2 - (\mathbf{v}^\top \sin \mathbf{x})^2}) = 0$.

**Property 2.** It suffices to show that for $\mathbf{x} \in S_{\mathbf{v}}$, $|h(\sin \mathbf{x})| = \tilde{O}(\frac{1}{d})$.

For $\mathbf{x} \in S_{\mathbf{v}}$, we have $\mathbf{v}^\top \sin \mathbf{x} \in [-\frac{\log d}{\sqrt{d}}, \frac{\log d}{\sqrt{d}}]$. By symmetry, we may just consider the case where $\mathbf{v}^\top \sin \mathbf{x} > 0$.

$$
\begin{aligned}
|h(\sin \mathbf{x})| &\leq |h_1(\mathbf{v}^\top \sin \mathbf{x})| \\
&= |\left(-16|x|^5 + 48x^4 - 48|x|^3 + 16x^2\right)| \text{ where } x = \mu \mathbf{v}^\top \sin \mathbf{x} \\
&\leq C(\frac{\log d}{\sqrt{d}})^2 \\
&\leq C \frac{\log^2 d}{d}
\end{aligned}
$$

Here $C > 0$ is a large enough universal constant.

**Property 3.** **(Part I.)** We show that there are no SOSP in $S_2$. For $\mathbf{x} : \|\mathbf{x}\| > 3/\mu$, we argue that either the gradient is large, due to contribution from $\|\sin \mathbf{x}\|^2$, or the Hessian has large negative eigenvalue (points close to the boundary of $H$). Denote $G(\mathbf{x}) = \|\sin \mathbf{x}\|^2$. We may compute the gradient of $G$ as follows:

$$
\frac{\partial}{\partial x_i} G(\mathbf{x}) = 2 \sin(x_i) \cos(x_i) = \sin(2x_i),
$$

$$
\|\nabla G(\mathbf{x})\| = \sqrt{\sum_i (\sin(2x_i))^2} \geq \sqrt{\sum_i x_i^2} = \|\mathbf{x}\| \text{ for all } \mathbf{x} \in [-\xi, \xi]^d,
$$

where $\xi \approx 0.95$ is the positive root of the equation $\sin 2x = x$. On $S_2$, $\nabla F(\mathbf{x}) = \nabla G(\mathbf{x})$, so $\|\nabla F(\mathbf{x})\| > \frac{3}{\mu} = 1 \times 10^{-2}$ for $\mathbf{x} \in S_2 \cap [-\xi, \xi]^d$. We may also compute the Hessian of $G$:

$$
\nabla^2 G(\mathbf{x}) = diag(2 \cos(2\mathbf{x})),
$$

$$
\lambda_{\min}(\nabla^2 G(\mathbf{x})) = \min_i \{2 \cos(2x_i)\} \leq 2 \cdot (\frac{\pi}{4} - x) < 2 \cdot (\frac{\pi}{4} - \xi) \ \forall \mathbf{x} \in S_2 \setminus [-\xi, \xi]^d.
$$

Since $(\frac{\pi}{4} - \xi) < -0.15$, $\lambda_{\min}(\nabla^2 F(\mathbf{x})) < -0.3$.

**(Part II.)** We argue that $F$ has no SOSP in $S_{\mathbf{v}}$. For $\mathbf{y} = \sin(\mathbf{x})$, we consider two cases: (i) $z = \sqrt{\|\mathbf{y}\|^2 - (\mathbf{v}^\top \mathbf{y})^2}$ large and (ii) $z$ small.

Write $g(\mathbf{x}) = h(\sin \mathbf{x})$, and denote $\nabla h(\mathbf{x})|_{\sin \mathbf{x}}$, $\nabla^2 h(\mathbf{x})|_{\sin \mathbf{x}}$ with $\nabla h(\mathbf{y}), \nabla^2 h(\mathbf{y})$. Let $\mathbf{u} \circ \mathbf{v}$ denote the Schur product of $\mathbf{u}$ and $\mathbf{v}$. We may compute the gradient and Hessian of $g$:

$$
\nabla g(\mathbf{x}) = \nabla h(\mathbf{y}) \circ \cos(\mathbf{x}),
$$
$$
\nabla^2 g(\mathbf{x}) = diag(\cos \mathbf{x}) \nabla^2 h(\mathbf{y}) diag(\cos \mathbf{x}) - \nabla h(\mathbf{y}) \circ \sin(\mathbf{x}).
$$

Now we change the coordinate system such that $\mathbf{v} = (1, 0, \cdots, 0)$. $\|\nabla h(\mathbf{y})\|$ and $\lambda_{\min}(\nabla^2 h(\mathbf{y}))$ are invariant to such a transform. Under this coordinate system, $h(\mathbf{y}) = h_1(y_1) \cdot h_2(\sqrt{\|\mathbf{y}\|^2 - (y_1)^2})$

(i): $z \geq \frac{1}{2\mu}$. We show that $\|\nabla F\|$ is large.

Let $\mathcal{P}_{-1}(\mathbf{u})$ denote the projection of $\mathbf{u}$ onto the orthogonal component of the first standard basis vector.

Since $\forall i \neq 1$, $\frac{\partial}{\partial y_i} h(\mathbf{y}) = h_1(y_1) h_2'(z) \frac{y_i}{z}$, we have

$$
\begin{aligned}
\mathcal{P}_{-1}(\nabla h(\mathbf{y})) &= \frac{h_1(y_1) h_2'(z)}{z} \mathcal{P}_{-1}(\mathbf{y}) \text{ where } \frac{h_1(y_1) h_2'(z)}{z} > 0 \\
\mathcal{P}_{-1}(\nabla F(\mathbf{x})) &= \mathcal{P}_{-1}(\nabla g(\mathbf{x}) + \nabla G(x)) = \mathcal{P}_{-1}(\nabla h(\mathbf{y}) \circ \cos(\mathbf{x}) + \mathbf{y} \circ \cos \mathbf{x}) \\
&= \left( \frac{h_1(y_1) h_2'(z)}{z} \mathcal{P}_{-1}(\mathbf{y}) + \mathcal{P}_{-1}(\mathbf{y}) \right) \circ \cos \mathbf{x} \\
\|\nabla F(\mathbf{x}))\| &\geq \|\mathcal{P}_{-1}(\nabla F(\mathbf{x}))\| \geq \|\mathcal{P}_{-1}(\mathbf{y}) \circ \cos \mathbf{x}\| \geq z \cdot \min_i |\cos(x_i)| \\
&\geq \frac{1}{2\mu} \cdot 0.999 \geq 1 \times 10^{-3} \text{ since } |x_i| \leq 3/\mu
\end{aligned}
$$

(ii): $z < \frac{1}{2\mu}$. We show that $\nabla^2 F(\mathbf{x})$ has large negative eigenvalue. First we compute the second derivative of $h$ in the direction of the first coordinate:

$$\frac{\partial^2 h}{\partial y_1^2} = h_2(z)h_1''(y_1) \leq 32\mu^2 g_2(1/2) = -10\mu^2.$$

Now we use this to upper bound the smallest eigenvalue of $\nabla^2 F(\mathbf{x})$.

$$\lambda_{\min}(\nabla^2 h(\mathbf{y})) \leq \min_i \frac{\partial^2}{\partial y_i^2} h(\mathbf{y}) \leq \frac{\partial^2 h}{\partial y_1^2}$$

$$\lambda_{\min}(\nabla^2 g(\mathbf{x})) \leq \left(\lambda_{\min}\left(diag(\cos\mathbf{x})\nabla^2 h(\mathbf{y})diag(\cos\mathbf{x})\right) + \lambda_{\max}\left(-diag(\nabla h(\mathbf{y}) \circ \sin(\mathbf{x}))\right)\right)$$

$$\leq -0.999 \cdot 10\mu^2 + 0.01 \cdot \max_{\mathbf{y}}(h_2(z)h_1'(y_1) + h_1(y_1)h_2'(z)) \quad \text{since } |x_i| \leq 3/\mu$$

$$\leq -0.999 \cdot 10\mu^2 + 0.01 \cdot 3\mu$$

Finally,

$$\nabla^2 G(\mathbf{x}) = diag(2\cos(2\mathbf{x})) \implies \|\nabla^2 G(\mathbf{x})\| \leq 2,$$

$$\lambda_{\min}(\nabla^2 F(\mathbf{x})) \leq -0.999 \cdot 10\mu^2 + 0.01 \cdot 3\mu + 2 \leq -8 \times 10^5.$$

**Property 4.** $O(1)$-bounded: Lemma 17 shows that $|h(\mathbf{y})| \leq 1$. $\|\sin\mathbf{x}\|^2 \leq d$. Therefore $|F| \leq 1 + d$.

$O(1)$-gradient Lipschitz: $\|\nabla^2 F(\mathbf{x})\| \leq \|\nabla^2 G(\mathbf{x})\| + \|\nabla^2 g(\mathbf{x})\|$. We know $\|\nabla^2 G(\mathbf{x})\| \leq 2$.

$$\|\nabla^2 g(\mathbf{x})\| = \|diag(\cos\mathbf{x})\nabla^2 h(\mathbf{y})diag(\cos\mathbf{x}) - diag(\nabla h(\mathbf{y}) \circ \sin(\mathbf{x}))\|$$

$$\leq \|diag(\cos\mathbf{x})\|^2 \cdot \|\nabla^2 h(\mathbf{y})\| + \|diag(\nabla h(\mathbf{y}))\| \cdot \|diag(\sin\mathbf{x})\|$$

$$\leq 1 \cdot 68\mu^2 + 3\mu \cdot 1 \leq 7 \times 10^6 \quad \text{using lemma 17}$$

$O(1)$-Hessian Lipschitz: First bound the Hessian Lipschitz constant of $G(\mathbf{x})$.

$$\|\nabla^2 G(\mathbf{x}) - \nabla^2 G(\mathbf{z})\| \leq 2\|\cos(2\mathbf{x}) - \cos(2\mathbf{z})\| \leq 4\|\mathbf{x} - \mathbf{z}\|.$$

Now we bound the Hessian Lipschitz constant of $g(\mathbf{x})$. Denote $\mathbf{A}(\mathbf{x}) = diag(\cos\mathbf{x})$ and $\mathbf{B}(\mathbf{x}) = diag(\sin\mathbf{x})$.

$$\|\mathbf{A}(\mathbf{x}_1)\nabla^2 h(\mathbf{y}_1)\mathbf{A}(\mathbf{x}_1) - \mathbf{A}(\mathbf{x}_2)\nabla^2 h(\mathbf{y}_2)\mathbf{A}(\mathbf{x}_2)\|$$

$$\leq \|\mathbf{A}(\mathbf{x}_1)\nabla^2 h(\mathbf{y}_1)\mathbf{A}(\mathbf{x}_1) - \mathbf{A}(\mathbf{x}_1)\nabla^2 h(\mathbf{y}_1)\mathbf{A}(\mathbf{x}_2)\| + \|\mathbf{A}(\mathbf{x}_1)\nabla^2 h(\mathbf{y}_1)\mathbf{A}(\mathbf{x}_2) - \mathbf{A}(\mathbf{x}_1)\nabla^2 h(\mathbf{y}_2)\mathbf{A}(\mathbf{x}_2)\|$$

$$+ \mathbf{A}(\mathbf{x}_1)\nabla^2 h(\mathbf{y}_2)\mathbf{A}(\mathbf{x}_2) - \mathbf{A}(\mathbf{x}_2)\nabla^2 h(\mathbf{y}_2)\mathbf{A}(\mathbf{x}_2)\|$$

$$\leq 68\mu^2\|\mathbf{x}_1 - \mathbf{x}_2\| + 1000\mu^3\|\mathbf{x}_1 - \mathbf{x}_2\| + 68\mu^2\|\mathbf{x}_1 - \mathbf{x}_2\| \quad \text{from lemma 17}.$$

$$\|diag(\nabla h(\mathbf{y}_1))\mathbf{B}(\mathbf{x}_1) - diag(\nabla h(\mathbf{y}_2))\mathbf{B}(\mathbf{x}_2)\|$$

$$\leq \|diag(\nabla h(\mathbf{y}_1))\mathbf{B}(\mathbf{x}_1) - diag(\nabla h(\mathbf{y}_1))\mathbf{B}(\mathbf{x}_2)\| + \|diag(\nabla h(\mathbf{y}_1))\mathbf{B}(\mathbf{x}_2) - diag(\nabla h(\mathbf{y}_2))\mathbf{B}(\mathbf{x}_2)\|$$

$$\leq (3\mu + 68\mu^2)\|\mathbf{x}_1 - \mathbf{x}_2\| \quad \text{from lemma 17}.$$

$$\|\nabla^2 g(\mathbf{x}_1) - \nabla^2 g(\mathbf{x}_2)\|$$

$$= (144\mu^3 + 204\mu^2 + 3\mu)\|\mathbf{x}_1 - \mathbf{x}_2\|.$$

Therefore $F(\mathbf{x})$ is $(2.8 \times 10^{10})$-Hessian Lipschitz.

$\square$

Now we need to prove smoothness properties of $h(\mathbf{y})$ that are used in the previous proof. In the following lemma, we prove that $h(\mathbf{y})$ as defined in equation (6) is bounded, Lipschitz, gradient-Lipschitz, and Hessian-Lipschitz.

**Lemma 17** (Properties of $h(\mathbf{y})$)**.** $h(\mathbf{y})$ *as given in Definition* 6 *is O(1)-bounded, O(1)-Lipschitz, O(1)-gradient Lipschitz, and O(1)-Hessian Lipschitz.*

*Proof.* WLOG assume $\mathbf{v} = (1, 0, \cdots, 0)^\top$. Denote $u = y_1, \mathbf{w} = (y_2, \cdots, y_d)^\top$. Let $\otimes$ denote tensor product.

Note that $|h_1'| \le 3\mu, |h_2'| \le 2\mu, |h_1''| \le 32\mu^2, |h_2''| \le 12\mu^2, |h_1'''| \le 300\mu^3, |h_2'''| \le 48\mu^3$. Assume $\mu > 2$.

1. O(1)-bounded: $|h| \le |h_1| \cdot |h_2| \le 1$.

2. O(1)-Lipschitz: $\|\nabla h(\mathbf{y})\| = \sqrt{h_2(\|\mathbf{w}\|)h_1'(u) + h_1(u)h_2'(\|\mathbf{w}\|)} \le 3\mu \le O(1)$.

3. O(1)-gradient Lipschitz:

   $$\nabla^2 h(\mathbf{y}) = h_1(u)\nabla^2 h_2(\|\mathbf{w}\|) + h_2(\|\mathbf{w}\|)\nabla^2 h_1(u) + \nabla h_1(u)\nabla h_2(\|\mathbf{w}\|)^\top + \nabla h_2(\|\mathbf{w}\|)\nabla h_1(u)^\top.$$

   $\|\nabla h_1(u)\| \le 3\mu$. Notice that the following are also O(1):

   $$\|\nabla h_2(\|\mathbf{w}\|)\| = \|h_2'(\|\mathbf{w}\|)\frac{\mathbf{w}}{\|\mathbf{w}\|}\| \le 2\mu;$$

   $$\|\nabla^2 h_2(\mathbf{w})\| \le \|h_2''(\|\mathbf{w}\|)\frac{\mathbf{w}\mathbf{w}^\top}{\|\mathbf{w}\|^2}\| + \|h_2'(\|\mathbf{w}\|)\frac{\|\mathbf{w}\|^2 \mathbf{I} - \mathbf{w}\mathbf{w}^\top}{\|\mathbf{w}\|^3}\| \le |h_2''(\|\mathbf{w}\|)| + |h_2'(\|\mathbf{w}\|)|/\|\mathbf{w}\| \le 12\mu^2 + 12\mu.$$

   Therefore, $\|\nabla^2 h(\mathbf{y})\| \le 24\mu^2 + 32\mu^2 + 2 \cdot 2\mu \cdot 3\mu \le 68\mu^2$.

4. O(1)-Hessian Lipschitz: We first argue that $\nabla^2 h_2(\mathbf{w})$ is Lipschitz. For $\|\mathbf{w}\| \ge 1/\mu$, $\nabla^2 h_2(\mathbf{w}) = 0$. So we consider $\|\mathbf{w}\| < \mu$. We obtain the following by direct computation.

   $$\nabla^2 h_2(\mathbf{w}) = h_2''(\|\mathbf{w}\|)\frac{\mathbf{w}\mathbf{w}^\top}{\|\mathbf{w}\|^2} + h_2'(\|\mathbf{w}\|)\frac{\|\mathbf{w}\|^2 \mathbf{I} - \mathbf{w}\mathbf{w}^\top}{\|\mathbf{w}\|^3}$$

   $$= 24\mu^4 \mathbf{w}\mathbf{w}^\top - 24\mu^3 \frac{\mathbf{w}\mathbf{w}^\top}{\|\mathbf{w}\|} + (12\mu^4\|\mathbf{w}\|^2 - 24\mu^3\|\mathbf{w}\| + 12\mu^2)\mathbf{I}$$

   $$\nabla^3 h_2(\mathbf{w}) = 24\mu^4(\mathbf{w} \otimes \mathbf{I} + \mathbf{I} \otimes \mathbf{w}) - 24\mu^3 \frac{\|\mathbf{w}\|^2(\mathbf{w} \otimes \mathbf{I} + \mathbf{I} \otimes \mathbf{w}) - \mathbf{w} \otimes \mathbf{w} \otimes \mathbf{w}}{\|\mathbf{w}\|^3}$$

   $$+ 24\mu^4 \mathbf{I} \otimes \mathbf{w} - 24\mu^3 \frac{\mathbf{I} \otimes \mathbf{w}}{\|\mathbf{w}\|}$$

   $$\|\nabla^3 h_2(\mathbf{w})\| \le 48\mu^3 + 72\mu^2 + 24\mu^3 + 24\mu^2 \le 144\mu^3$$

   We may easily check that indeed $\lim_{\|\mathbf{w}\| \to 1} \|\nabla^2 h_2(\mathbf{w})\| = 0$.

   Therefore $\nabla^2 h_2(\mathbf{w})$ is $144\mu^3$-Lipschitz.

   $$\|h_1(u_1)\nabla^2 h_2(\|\mathbf{w}_1\|) - h_1(u_2)\nabla^2 h_2(\|\mathbf{w}_1\|)\| \le (144\mu^3 + 3\mu \cdot 24\mu^2)\|\mathbf{y}_1 - \mathbf{y}_2\|$$
   $$\|h_2(\|\mathbf{w}_1\|)\nabla^2 h_1(u_1) - h_2(\|\mathbf{w}_2\|)\nabla^2 h_1(u_2)\| \le (32\mu^2 \cdot 2\mu + 300\mu^3)\|\mathbf{y}_1 - \mathbf{y}_2\|$$
   $$\|\nabla h_1(u_1)\nabla h_2(\|\mathbf{w}_1\|)^\top - \nabla h_1(u_2)\nabla h_2(\|\mathbf{w}_2\|)^\top\| \le (3\mu \cdot 24\mu^2 + 2\mu \cdot 32\mu^2)\|\mathbf{y}_1 - \mathbf{y}_2\|$$

   By triangle inequality, using the above, we obtain

   $$\|\nabla^2 h(\mathbf{y}_1) - \nabla^2 h(\mathbf{y}_2)\| \le 1000\mu^3\|\mathbf{y}_1 - \mathbf{y}_2\|.$$

   This proves that $h(\mathbf{y})$ is $1000\mu^3$-Hessian Lipschitz.

$\square$

## C.2 Scaling the Hard Instance

Now we show how to scale the function we described in order to achieve the final lower bound with correct dependencies on $\epsilon$ and $\rho$.

Given any $\epsilon, \rho > 0$, define

$$\tilde{F}(\mathbf{x}) = \epsilon r F(\frac{1}{r}\mathbf{x}), \tilde{f}(\mathbf{x}) = \epsilon r f(\frac{1}{r}\mathbf{x}), \tag{8}$$

where $r = \sqrt{\epsilon/\rho}$ and $F, f$ are defined as in Equation 6. Define the 'scaled' regions:

- $\tilde{S} = \{\mathbf{x} \in \mathbb{R}^d : \|\mathbf{x}\| \le 3r/\mu\}$ be the $d$-dimensional ball with radius $3r/\mu$.
- $\tilde{H} = [-\frac{\pi}{2}r, \frac{\pi}{2}r]^d$ be the $d$-dimensional hypercube with side length $\pi r$.
- $\tilde{S}_{\mathbf{v}} = \{\mathbf{x} \in \tilde{S} : \langle \sin\frac{1}{r}\mathbf{x}, \mathbf{v}\rangle \le \frac{\log d}{\sqrt{d}}\}$.
- $\tilde{S}_2 = \tilde{H} - \tilde{S}$.

Defined as above, $(\tilde{F}, \tilde{f})$ satisfies the properties stated in lemma 18, which makes it hard for any algorithm to optimize $\tilde{F}$ given only access to $\tilde{f}$.

**Lemma 18.** *Let $\tilde{F}, \tilde{f}, \mathbf{v}, \tilde{S}_2, \tilde{S}_{\mathbf{v}}$ be as defined in 8. Then for any $\epsilon, \rho > 0$, $F, f$ satisfies:*

1. *$\tilde{f}$ in the non-informative region $\tilde{S}_2 \cup \tilde{S}_{\mathbf{v}}$ is independent of $\mathbf{v}$.*

2. *$\sup_{x \in \tilde{S}_{\mathbf{v}}} |\tilde{f} - \tilde{F}| \le \frac{\epsilon^{1.5}}{\sqrt{\rho}d}$ up to poly-$\log d$ and constant factors.*

3. *$\tilde{F}$ has no $O(\epsilon)$-SOSP in the non-informative region $\tilde{S}_2 \cup \tilde{S}_{\mathbf{v}}$.*

4. *$\tilde{F}$ is $B$-bounded, $O(\rho)$-Hessian Lipschitz, and $O(\ell)$-gradient Lipschitz.*

*Proof.* This is implied by Lemma 16. To see this, notice

1. We have simply scaled each coordinate axis by $r$.

2. $|\tilde{F} - \tilde{f}| = \epsilon r |F - f| = \frac{\epsilon^{1.5}}{\sqrt{\rho}}|F - f|$.

3. $\|\nabla \tilde{F}\| = \epsilon \|\nabla F\|$ and $\|\nabla^2 \tilde{F}\| = \sqrt{\rho\epsilon}\|\nabla^2 F\|$. Since $F$ has no $1 \times 10^{-12}$-SOSP in $S_2 \cup S_{\mathbf{v}}$. Taking into account the Hessian Lipschitz constant of $F$, $\tilde{F}$ has no $\frac{\epsilon}{10^{12}}$-SOSP in $\tilde{S}_2 \cup \tilde{S}_{\mathbf{v}}$.

4. We must have $B > d + \frac{\epsilon^{1.5}}{\sqrt{\rho}}$. Then, $\tilde{F}$ is $B$-bounded, $(7 \times 10^6)\sqrt{\rho\epsilon}$- gradient Lipschitz, and $(2.8 \times 10^{10})\rho$-Hessian Lipschitz.

$\square$

## C.3 Proof of the Theorem

We are now ready to state the two main lemmas used to prove Theorem 8.

The following lemma uses the concentration of measure in higher dimensions to argue that the probability that any fixed point lies in the informative region $\tilde{S}_{\mathbf{v}}$ is very small.

**Lemma 19** (Probability of landing in informative region). *For any arbitrarily fixed point $\mathbf{x} \in \tilde{S}$, $\Pr(\mathbf{x} \notin \tilde{S}_{\mathbf{v}}) \le 2e^{-(\log d)^2/2}$.*

*Proof.* Recall the definition of $\tilde{S}_{\mathbf{v}}$: $\tilde{S}_{\mathbf{v}} = \{\mathbf{x} \in \tilde{S} : \langle \sin\frac{1}{r}\mathbf{x}, \mathbf{v}\rangle \le \frac{\log d}{\sqrt{d}}\}$. Since $\mathbf{x} \in \tilde{S}$, we have $\|\mathbf{x}\| \le 3r/\mu \le r$ (as $\mu \ge 3$). Therefore, by inequality $|\sin\theta| \le |\theta|$, we have:

$$\|\sin\frac{\mathbf{x}}{r}\|^2 = \sum_{i=1}^d |\sin\frac{x^{(i)}}{r}|^2 \le \sum_{i=1}^d |\frac{x^{(i)}}{r}|^2 \le 1.$$

Denote unit vector $\hat{\mathbf{y}} = \sin \frac{\mathbf{x}}{r}/\|\sin \frac{\mathbf{x}}{r}\|$. This gives:

$$\Pr(\mathbf{x} \notin \tilde{S}_{\mathbf{v}}) = \Pr(|\langle \sin \frac{\mathbf{x}}{r}, \mathbf{v}\rangle| \geq \frac{\log d}{\sqrt{d}}) = \Pr(|\langle \hat{\mathbf{y}}, \mathbf{v}\rangle| \geq \frac{\log d}{\sqrt{d}\|\sin \frac{\mathbf{x}}{r}\|})$$

$$\leq \Pr(|\langle \hat{\mathbf{y}}, \mathbf{v}\rangle| \geq \frac{\log d}{\sqrt{d}}) \quad \text{since } \|\sin \frac{\mathbf{x}}{r}\| \leq 1$$

$$= \frac{\text{Area}(\{\mathbf{u} : \|\mathbf{u}\| = 1, |\langle \hat{\mathbf{y}}, \mathbf{u}\rangle| > \frac{\log d}{\sqrt{d}}\})}{\text{Area}(\{\mathbf{u} : \|\mathbf{u}\| = 1\})}$$

$$\leq 2e^{-(\log d)^2/2} \text{ by lemma 22}$$

This finishes the proof. $\qquad\square$

Thus we know that for a single fixed point, the probability of landing in $\tilde{S}_{\mathbf{v}}$ is less than $2(1/d)^{\log d/2}$. We note that this is smaller than $1/\text{poly}(d)$. The following lemma argues that even for a possibly adaptive sequence of points (of polynomial size), the probability that any of them lands in $\tilde{S}_{\mathbf{v}}$ remains small, as long as the query at each point does not reveal information about $\tilde{S}_{\mathbf{v}}$.

**Lemma 20** (Probability of adaptive sequences landing in the informative region). *Consider a sequence of points and corresponding queries with size $T$: $\{(\mathbf{x}_i, q(\mathbf{x}_i))\}_{i=1}^{T}$, where the sequence can be adaptive, i.e. $\mathbf{x}_t$ can depend on all previous history $\{(\mathbf{x}_i, q(\mathbf{x}_i))\}_{i=1}^{t-1}$. Then as long as $q(\mathbf{x}_i) \perp \mathbf{v} \mid \mathbf{x}_i \in \tilde{S}_{\mathbf{v}}$, we have $\Pr(\exists t \leq T : \mathbf{x}_t \notin \tilde{S}_{\mathbf{v}}) \leq 2Te^{-(\log d)^2/2}$.*

*Proof.* Clearly $\Pr(\exists t \leq T : \mathbf{x}_t \notin \tilde{S}_{\mathbf{v}}) = 1 - \Pr(\forall t \leq T : \mathbf{x}_t \in \tilde{S}_{\mathbf{v}})$. By product rule, we have:

$$\Pr(\forall t \leq T : \mathbf{x}_t \in \tilde{S}_{\mathbf{v}}) = \prod_{t=1}^{T} \Pr(\mathbf{x}_t \in \tilde{S}_{\mathbf{v}} | \forall \tau < t : \mathbf{x}_\tau \in \tilde{S}_{\mathbf{v}}).$$

Denote $D_i = \{\mathbf{v} \in \mathbb{S}^{d-1} | \langle \sin \frac{1}{r}\mathbf{x}_i, \mathbf{v}\rangle > \frac{\log d}{\sqrt{d}}\}$, where $\mathbb{S}^{d-1}$ denotes the unit sphere in $\mathbb{R}^d$ centered at the origin. Clearly, $\mathbf{v} \notin D_i$ is equivalent to $\mathbf{x}_i \in \tilde{S}_{\mathbf{v}}$. Consider term $\Pr(\mathbf{x}_t \in \tilde{S}_{\mathbf{v}} | \forall \tau < t : \mathbf{x}_\tau \in \tilde{S}_{\mathbf{v}})$. Conditioned on the event that $E = \{\forall \tau < t : \mathbf{x}_\tau \in \tilde{S}_{\mathbf{v}}\}$, we know $\mathbf{v} \in \mathbb{S}^{d-1} - \cup_{i=1}^{t-1} D_i$. On the other hand, since $q(\mathbf{x}_\tau) \perp \mathbf{v}|E$ for all $\tau < t$, therefore, conditioned on event $E$, $\mathbf{v}$ is uniformly distributed over $\mathbb{S}^{d-1} - \cup_{i=1}^{t-1} D_i$, and:

$$\Pr(\mathbf{x}_t \in \tilde{S}_{\mathbf{v}} | \forall \tau < t : \mathbf{x}_\tau \in \tilde{S}_{\mathbf{v}}) = \frac{\text{Area}(\mathbb{S}^{d-1} - \cup_{i=1}^{t} D_i)}{\text{Area}(\mathbb{S}^{d-1} - \cup_{i=1}^{t-1} D_i)}.$$

Thus by telescoping:

$$\Pr(\forall t \leq T : \mathbf{x}_t \in \tilde{S}_{\mathbf{v}}) = \prod_{t=1}^{T} \frac{\text{Area}(\mathbb{S}^{d-1} - \cup_{i=1}^{t} D_i)}{\text{Area}(\mathbb{S}^{d-1} - \cup_{i=1}^{t-1} D_i)} = \frac{\text{Area}(\mathbb{S}^{d-1} - \cup_{i=1}^{T} D_i)}{\text{Area}(\mathbb{S}^{d-1})}.$$

This gives:

$$\Pr(\exists t \leq T : \mathbf{x}_t \notin \tilde{S}_{\mathbf{v}}) = 1 - \Pr(\forall t \leq T : \mathbf{x}_t \in \tilde{S}_{\mathbf{v}}) = \frac{\text{Area}(\cup_{i=1}^{T} D_i)}{\text{Area}(\mathbb{S}^{d-1})}$$

$$\leq \sum_{i=1}^{T} \frac{\text{Area}(D_i)}{\text{Area}(\mathbb{S}^{d-1})} \leq T \max_i \frac{\text{Area}(D_i)}{\text{Area}(\mathbb{S}^{d-1})} = T \max_{\mathbf{x}} \Pr(\mathbf{x} \notin \tilde{S}_{\mathbf{v}}) \leq 2Te^{-(\log d)^2/2}.$$

In last inequality, we used Lemma 19, which finishes the proof. $\qquad\square$

Now we have all the ingredients to prove Theorem 8, restated below more formally.

**Theorem 21** (Lower bound). *For any $B > 0, \ell > 0, \rho > 0$, there exists $\epsilon_0 = \Theta(\min\{\ell^2/\rho, (B^2\rho)^{1/3}\})$ so that for any $\epsilon \in (0, \epsilon_0]$, there exists a function pair $(F, f)$ satisfying Assumption A1 with $\nu = \tilde{\Theta}(\sqrt{\epsilon^3/\rho} \cdot (1/d))$, so that any algorithm will fail, with high probability, to find SOSP of $F$ given only $o(d^{\sqrt{\log d}})$ of zero-th order queries of $f$.*

*Proof.* Take $(\tilde{F}, \tilde{f})$ to be as defined in Definition 8. The proof proceeds by first showing that no SOSP can be found in a constrained set $\tilde{S}$, and then using a reduction argument. The key step of the proof involves the following two claims:

1. First we claim that any algorithm $\mathcal{A}$ making $o(d^{\sqrt{\log d}})$ function-value queries of $f$ to find $\epsilon$-SOSP of $F$ in $\tilde{S}$, only queries points in $\tilde{S}_{\mathbf{v}}$, w.h.p., and fails to output $\epsilon$-SOSP of $F$.

2. Next suppose if there exists $\mathcal{A}$ making $o(d^{\sqrt{\log d}})$ function-value queries of $f$ that finds $\epsilon$-SOSP of $F$ in $\mathbb{R}^d$ w.h.p. Then this algorithm also finds $\epsilon$-SOSP of $F$ on $\tilde{S}$ w.h.p., which is a contradiction.

Proof of claim 1:

Note that because $\|\mathbf{x}_t\| \le r/100$, $\|\sin \frac{1}{r}\mathbf{x}_t\| \le \|\mathbf{x}_t\|/r \le 1/100$.

Let $\mathbf{v}$ be an arbitrary unit vector. Suppose a possibly randomized algorithm $\mathcal{A}$ queries points in $\tilde{S}$, $\{X_t\}_{t=1}^T$. Let $\mathcal{F}_t$ denote $\sigma(f(X_1), \cdots, f(X_t))$. Let $X_t \sim \mathcal{A}(t|\mathcal{F}_{t-1})$.

For any $i$, on the event that $X_i \in \tilde{S}_{\mathbf{v}}$, we have that $f(X_i) = 0$, as established in Lemma 18. Therefore it is trivially true that $f(X_i)$ is independent of $\mathbf{v}$ conditioned on $\{X_i \in \tilde{S}_{\mathbf{v}}\}$.

By Lemma 20,

$$\Pr(X_t \in \tilde{S}_{\mathbf{v}} \forall t \le T) \ge 1 - 2Te^{-(\log d)^2/2}$$
$$\ge 1 - e^{-(\log d)^2/4} \text{ for all } d \text{ large enough since } T = o(d^{\sqrt{\log d}}).$$

Proof of claim 2:

Since $f, F$ are periodic over $d$-dimensional hypercubes of side length $\pi r$, finding $\epsilon$-SOSP of $F$ on $\mathbb{R}$ implies finding $\epsilon$-SOSP of $F$ in $S$. Given claim 1, any algorithm making only $o(d^{\sqrt{\log d}})$ queries will fail to find $\epsilon$-SOSP of $F$ in $\mathbb{R}^d$ w.h.p. $\qquad\square$

For completeness, we now state the classical result showing that most of the surface area of a sphere lies close to the equator; it was used in the proof of Lemma 19.

**Lemma 22** (Surface area concentration for sphere). *Let $S^{d-1} = \{x \in \mathbb{R}^d : \|x\|_2 = 1\}$ denote the Euclidean sphere in $\mathbb{R}^d$. For $\varepsilon > 0$, let $C(\varepsilon)$ denote the spherical cap of height $\varepsilon$ above the origin. Then*
$$\frac{Area(C(\varepsilon))}{Area(S^{d-1})} \le e^{-d\varepsilon^2/2}.$$

*Proof.* Let $D$ be the spherical cone subtended at one end by $C(\varepsilon)$ and let $B^d$ denote the unit Euclidean ball in $\mathbb{R}^d$. By Pythagoras' Theorem, we can enclose $D$ in a sphere of radius $\sqrt{1 - \varepsilon^2}$. By elementary calculus,

$$\frac{\text{Area}(C(\varepsilon))}{\text{Area}(S^{d-1})} = \frac{\text{Volume}(D)}{\text{Volume}(B^d)} \le \frac{\text{Volume}(\sqrt{1-\varepsilon^2}B^d)}{\text{Volume}(B^d)} \le (1 - \varepsilon^2)^{d/2} \le e^{-d\varepsilon^2/2}.$$

$\qquad\square$

# D  Information-theoretic Limits

In this section, we prove upper and lower bounds for algorithms that may run in exponential time. This establishes the information-theoretic limit for problem 1. Compared to the previous (polynomial time) setting, now the dependency on dimension $d$ is removed.

**Algorithm 1** Exponential Time Algorithm

---

**Input:** function value oracle for $f$, hyperparameter $\epsilon'$

Construct (1) $\{\mathbf{x}_t\}_{t=1}^N$, an $O(\epsilon/\ell)$-cover in the euclidean metric of the ball of radius $O(B/\epsilon)$ in $\mathbb{R}^d$ centered at the origin (lemma 27); (2) $\{\mathbf{v}_i\}_{i=1}^V$, an $O(\epsilon/\ell)$-cover in the euclidean metric of the ball of radius $O(\epsilon)$ in $\mathbb{R}^d$ centered at the origin (lemma 27); (3) $\{H_j\}_{j=1}^P$, an $O(\epsilon/\ell)$-cover in the $L_\infty$ metric of the ball of radius $O(\ell)$ in $\mathbb{R}^{d\times d}$ centered at the origin (lemma 28); (4) $\mathcal{Z}$, an $O(\epsilon')$ cover in the euclidean metric of the unit sphere $\mathbb{S}^{d-1}$ in $\mathbb{R}^d$ (lemma 27)

**for** $t = 0, 1, \ldots, N$ **do**
    **for** $i = 0, 1, \ldots, V$ **do**
        **for** $j = 0, 1, \ldots, P$ **do**
            **if** $|f(\mathbf{x})+\mathbf{v}_i^\top(\mathbf{y}-\mathbf{x})+\frac{1}{2}(\mathbf{y}-\mathbf{x})^\top H_j(\mathbf{y}-\mathbf{x})-f(\mathbf{y})| \leq O(\rho r^3+\nu)$    $\forall\mathbf{y}=\mathbf{x}+r\mathbf{z}$,    $\mathbf{z} \in \mathcal{Z}$
            **then**
                **if** $\|\mathbf{v}_i\| \leq O(\epsilon)$ and $\lambda_{\min}(H_j) \geq -O(\sqrt{\rho\epsilon})$ **then**
                    **return** $\mathbf{x}_t$

---

## D.1 Exponential Time Algorithm to Remove Dimension Dependency

We first restate our upper bound, first stated in Theorem 9, below.

**Theorem 23.** *There exists an algorithm so that if the function pair $(F, f)$ satisfies Assumption A1 with $\nu \leq O(\sqrt{\epsilon^3/\rho})$ and $\ell > \sqrt{\rho\epsilon}$, then the algorithm will find an $\epsilon$-second-order stationary point of $F$ with an exponential number of queries.*

The algorithm is based on a procedure to estimate the gradient and Hessian at point $x$. This procedure will be applied to a exponential-sized covering of a compact space to find an SOSP.

Let $\mathcal{Z}$ be a $\epsilon'$ covering for unit sphere $\mathbb{S}^{d-1}$, where $\mathcal{Z}$ is symmetric (i.e. if $\mathbf{z} \in \mathcal{Z}$ then $-\mathbf{z} \in \mathcal{Z}$). It is easy to verify that such covering can be efficiently constructed with $|\mathcal{Z}| \leq O((1/\epsilon')^d)$ (Lemma 27). Then, for each point in the cover, we solve following feasibility problem:

$$\text{find} \quad \mathbf{g}, \mathcal{H} \tag{9}$$
$$\text{s.t.} \quad |f(\mathbf{x}) + \mathbf{g}^\top(\mathbf{y} - \mathbf{x}) + \frac{1}{2}(\mathbf{y} - \mathbf{x})^\top \mathcal{H}(\mathbf{y} - \mathbf{x}) - f(\mathbf{y})| \leq O(\rho r^3 + \nu)$$
$$\forall\mathbf{y} = \mathbf{x} + r\mathbf{z}, \quad \mathbf{z} \in \mathcal{Z},$$

where $r$ is scalar in the order of $O(\sqrt{\epsilon/\rho})$.

We will first show that any solution of this problem will give good estimates of the gradient and Hessian of $F$.

**Lemma 24.** *Any solution $(\mathbf{g}, \mathcal{H})$ to the above feasibility problem, Eq.(9), gives*

$$\|\mathbf{g} - \nabla F(\mathbf{x})\| \leq O(\epsilon) \ \text{ and } \ \|\mathcal{H} - \nabla^2 F(\mathbf{x})\| \leq O(\sqrt{\rho\epsilon})$$

*Proof.* When we have $\|f - F\|_\infty \leq \nu$, above feasibility problem is equivalent to solve following:

$$\text{find} \quad \mathbf{g}, \mathcal{H}$$
$$\text{s.t.} \quad |F(\mathbf{x}) + \mathbf{g}^\top(\mathbf{y} - \mathbf{x}) + \frac{1}{2}(\mathbf{y} - \mathbf{x})^\top \mathcal{H}(\mathbf{y} - \mathbf{x}) - F(\mathbf{y})| \leq O(\rho r^3 + \nu)$$
$$\forall\mathbf{y} = \mathbf{x} + r\mathbf{z}, \quad \mathbf{z} \in \mathcal{Z}.$$

Due to the Hessian-Lipschitz property, we have $|F(\mathbf{y}) - F(\mathbf{x}) - \nabla F(\mathbf{x})^\top(\mathbf{y} - \mathbf{x}) - (\mathbf{y} - \mathbf{x})^\top\nabla^2 F(\mathbf{x})(\mathbf{y} - \mathbf{x})| \leq \frac{1}{6}\rho r^3$, this means above feasibility problem is also equivalent to:

$$\text{find} \quad \mathbf{g}, \mathcal{H}$$
$$\text{s.t.} \quad |(\mathbf{g} - \nabla F(\mathbf{x})^\top(\mathbf{y} - \mathbf{x}) + \frac{1}{2}(\mathbf{y} - \mathbf{x})^\top(\mathcal{H} - \nabla^2 F(\mathbf{x}))(\mathbf{y} - \mathbf{x})| \leq O(\rho r^3 + \nu)$$
$$\forall\mathbf{y} = \mathbf{x} + r\mathbf{z}, \quad \mathbf{z} \in \mathcal{Z}.$$

Picking $\mathbf{y} - \mathbf{x} = \pm r\mathbf{z}$, by triangular inequality and the fact that $\mathcal{Z}$ is an $\epsilon'$-covering of $\mathbb{S}^{d-1}$, it is not hard to verify:

$$\|\mathbf{g} - \nabla F(\mathbf{x})\| \leq O\left(\frac{1}{1-\epsilon'}(\rho r^2 + \frac{\nu}{r})\right)$$

$$\|\mathcal{H} - \nabla^2 F(\mathbf{x})\| \leq O\left(\frac{1}{1-2\epsilon'}(\rho r + \frac{\nu}{r^2})\right).$$

Given $\nu \leq \frac{1}{c}\sqrt{\frac{\epsilon^3}{\rho}}$ for large enough constant $c$, and picking $r = c'\sqrt{\frac{\epsilon}{\rho}}$ with proper constant $c'$, we prove the lemma. $\qquad\square$

We then argue that (9) always has a solution.

**Lemma 25.** *Consider the metric $\|\cdot\| : \mathbb{R}^d \times \mathbb{R}^{d \times d} \to \mathbb{R}$, where $\|(\mathbf{g}, H)\| = \sqrt{\|\mathbf{g}\|^2 + \|H\|^2}$. Then $(\nabla F(\mathbf{x}), \nabla^2 F(\mathbf{x}))$ and a $O(\epsilon/\ell)$-neighborhood around it with respect to the $\|\cdot\|$ metric are the solutions to above feasibility problem.*

*Proof.* $(\nabla F(\mathbf{x}), \nabla^2 F(\mathbf{x}))$ is clearly one solution to the feasibility problem Eq.(9). Then, this lemma is true due to Hessian Lipschitz and gradient Lipschitz properties of $F$. $\qquad\square$

Now, since the algorithm can do an exhaustive search over a compact space, we just need to prove that there is an $\epsilon$-SOSP within a bounded distance.

**Lemma 26.** *Suppose function $f$ is $B$-bounded, then inside any ball of radius $B/\epsilon$, there must exist a $O(\epsilon/\ell)$-ball full of $2\epsilon$-SOSP.*

*Proof.* We can define a search path $\{\mathbf{x}_t\}$ to find a $\epsilon$-SOSP. Starting from an arbitrary point $\mathbf{x}_0$. (1) If the current point $\mathbf{x}_t$ satisfies $\|\mathbf{g}\| \geq \epsilon$, then following gradient direction with step-size $\epsilon/\ell$ decreases the function value by at least $\Omega(\|\mathbf{g}\|\epsilon/\ell)$; (2) If the current point $\mathbf{x}_t$ has negative curvature $\gamma \leq -\sqrt{\rho\epsilon}$, moving along direction of negative curvature with step-size $\sqrt{\epsilon/\rho}$ decreases the function value by at least $\Omega(\gamma\epsilon/\rho)$.

In both cases, we decrease the function value on average by $\Omega(\epsilon)$ per step. That is in a ball of radius $B/\epsilon$ around $\mathbf{x}_0$, there must be a $\epsilon$-SOSP. and in a $O(\epsilon/\ell)$-ball around this $\epsilon$-SOSP are all $2\epsilon$-SOSP due to the gradient and Hessian Lipschitz properties of $F$. $\qquad\square$

Combining all these lemmas we are now ready to prove the main theorem of this section:

*Proof of Theorem 9.* We show that Algorithm 1 is guaranteed to succeed within a number of function value queries of $f$ that is exponential in all problem parameters. First, by Lemma 26, we know that at least one of $\{\mathbf{x}_t\}_{t=1}^N$ must be an $O(\epsilon)$-SOSP of $F$. It suffices to show that for any $\mathbf{x}$ that is an $O(\epsilon)$-SOSP, Algorithm 1's subroutine will successfully return $\mathbf{x}$, that is, it must find a solution $\mathbf{g}, \mathcal{H}$, to the feasibility problem 9 that satisfies $\|\mathbf{g}\| \leq O(\epsilon)$ and $\lambda_{\min}(\mathcal{H}) \geq -O(\sqrt{\rho\epsilon})$.

If $\mathbf{x}$ satisfies $\|\nabla F(\mathbf{x})\| \leq O(\epsilon)$, then by lemma 24, all solutions to the feasibility problem 9 at $\mathbf{x}$ must satisfy $\|\mathbf{g}\| \leq O(\epsilon)$ and we must have $\|\mathcal{H}\|_\infty \leq \ell$ (implied by $\ell$-gradient Lipschitz). Therefore, by Lemma 25, we can guarantee that at least one of $\{\mathbf{v}_i, H_j\}_{i=1,j=1}^{i=V,j=P}$ will be in a solution to the feasibility problem.

Next, notice that because all the covers in Algorithm 1 have size at most $O((d/\epsilon)^{d^2})$ must terminate in $O(e^{d^2 \log \frac{d}{\epsilon}})$ steps. $\qquad\square$

In the following two lemmas, we provide simple methods for constructing an $\epsilon$-cover for a ball (as well as a sphere), and for matrices with bounded spectral norm.

**Lemma 27** (Construction of $\epsilon$-cover for ball and sphere). *For a ball in $\mathbb{R}^d$ of radius $R$ centered at the origin, the set of points $C = \{\mathbf{x} \in \mathbb{R}^d : \forall i, \mathbf{x}_i = j \cdot \frac{\epsilon}{\sqrt{d}}, j \in \mathbb{Z}, -\frac{R\sqrt{d}}{\epsilon} - 1 \leq j \leq \frac{R\sqrt{d}}{\epsilon} + 1\}$ is an $\epsilon$-cover of the ball, of size $O((R\sqrt{d}/\epsilon)^d)$. Consequently, it is also an $\epsilon$-cover for the sphere of radius $R$ centered at the origin.*

*Proof.* For any point $\mathbf{y}$ in the ball, we can find $\mathbf{x} \in C$ such that $|\mathbf{y}_i - \mathbf{x}_i| \le \epsilon/\sqrt{d}$ for each $i \in [d]$. By the Pythagorean theorem, this implies $\|\mathbf{y} - \mathbf{x}\| \le \epsilon$. $\qquad\square$

**Lemma 28** (Construction of $\epsilon$-cover for matrices with $\ell$-bounded spectral norm). *Let $\mathcal{M} = \{A \in \mathbb{R}^{d \times d} : \|A\| \le \ell\}$ denote the set of $d$ by $d$ matrices with $\ell$-bounded spectral norm. Then the set of points $C = \{M \in \mathbb{R}^{d \times d} : \forall i, k, M_{i,k} = j \cdot \frac{\epsilon}{d}, j \in \mathbb{Z}, -\frac{\ell d}{\epsilon} - 1 \le j \le \frac{\ell d}{\epsilon} + 1\}$ is an $\epsilon$-cover for $\mathcal{M}$, of size $O((\ell d/\epsilon)^{d^2})$*

*Proof.* For any matrix $M$ in $\mathcal{M}$, we can find $N \in C$ such that $|N_{i,k} - M_{i,k}| \le \epsilon/d$ for each $i, k \in [d]$. Since the Frobenius norm dominates the spectral norm, we have $\|N - M\| \le \|N - M\|_F \le \epsilon$. $\qquad\square$

## D.2  Information-theoretic Lower bound

To prove the lower bound for an arbitrary number of queries, we base our hard function pair on our construction in definition 6, except $\tilde{f}$ now coincides with $\tilde{F}$ only outside the sphere $S$. With this construction, no algorithm can do better than random guessing within $S$, since $\tilde{f}$ is completely independent of $\mathbf{v}$.

**Theorem 29** (Information-theoretic lower bound). *For $\tilde{f}, \tilde{F}$ defined as follows:*

$$\tilde{F}(\mathbf{x}) = \epsilon r F(\frac{1}{r}\mathbf{x}), \tilde{f}(\mathbf{x}) = \begin{cases} \epsilon r \|\sin \frac{1}{r}\mathbf{x}\|^2 & , \mathbf{x} \in S \\ \tilde{F}(\mathbf{x}) & , \mathbf{x} \notin S \end{cases}$$

*where $F$ is as defined in definition 6. Then we have $\sup_{\mathbf{x}} |\tilde{F}(\mathbf{x}) - \tilde{f}(\mathbf{x})| \le O(\frac{\epsilon^{1.5}}{\sqrt{\rho}})$ and no algorithm can output SOSP of $F$ with probability more than a constant.*

*Proof.* $\sup_{\mathbf{x}} |\tilde{F}(\mathbf{x}) - \tilde{f}(\mathbf{x})| = \sup_{\mathbf{x} \in S} |\tilde{F}(\mathbf{x}) - \tilde{f}(\mathbf{x})| \le \epsilon r$. Any solution output by any algorithm must be independent of $\mathbf{v}$ with probability 1, since $h = 0$ outside of $S$. Suppose the algorithm $\mathcal{A}$ outputs $\mathbf{x}$. Then $\Pr(\mathbf{x} \text{ is } \epsilon\text{-SOSP of } \tilde{F}) \le \Pr(\mathbf{x} \notin S_{\mathbf{v}}) \le 2e^{-(\log d)^2/2}$. The upper bound on probability of success does not depend on the number of iterations. Therefore, no algorithm can output SOSP of $F$ with probability more than a constant. $\qquad\square$

# E  Extension: Gradients pointwise close

In this section, we present an extension of our results to the problem of optimizing an unknown smooth function $F$ (population risk) when given only a gradient vector field $\mathbf{g} : \mathbb{R}^d \to \mathbb{R}^d$ that is pointwise close to the gradient $\nabla F$. In other words, we now consider the analogous problem but for a first-order oracle. Indeed, in some applications including the optimization of deep neural networks, it might be possible to have a good estimate of the gradient of the population risk. A natural question is, what is the error in the gradient oracle that we can tolerate to obtain optimization guarantees for the true function $F$? More precisely, we work with the following assumption.

**Assumption A1.** Assume that the function pair $(F : \mathbb{R}^d \to \mathbb{R}, f : \mathbb{R}^d \to \mathbb{R})$ satisfies the following properties:

1. $F$ is $\ell$-gradient Lipschitz and $\rho$-Hessian Lipschitz.

2. $f$ is $L$-Lipschitz and differentiable, and $\nabla f, \nabla F$ are $\tilde{\nu}$-pointwise close; i.e., $\|\nabla f - \nabla F\|_\infty \le \tilde{\nu}$.

We henceforth refer to $\tilde{\nu}$ as the *gradient error*. As we explained in Section 2, our goal is to find second-order stationary points of $F$ given only function value access to $\mathbf{g}$. More precisely:

**Problem 2.** Given function pair $(F, f)$ that satisfies Assumption A1, find an $\epsilon$-second-order stationary point of $F$ with only access to function values of $\mathbf{g} = \nabla f$.

We provide an algorithm, Algorithm 2, that solves Problem 2 for gradient error $\tilde{\nu} \le O(\epsilon/\sqrt{d})$. Like Algorithm 1, Algorithm 2 is also a variant of SGD whose stochastic gradient oracle, $\mathbf{g}(\mathbf{x} + \mathbf{z})$ where $\mathbf{z} \sim \mathcal{N}(0, \sigma^2 \mathbf{I})$, is derived from Gaussian smoothing.

---

**Algorithm 2** First order Perturbed Stochastic Gradient Descent (FPSGD)

---

**Input:** $\mathbf{x}_0$, learning rate $\eta$, noise radius $r$, mini-batch size $m$.
  **for** $t = 0, 1, \ldots,$ **do**
    sample $(\mathbf{z}_t^{(1)}, \cdots, \mathbf{z}_t^{(m)}) \sim \mathcal{N}(0, \sigma^2 \mathbf{I})$
    $\mathbf{g}_t(\mathbf{x}_t) \leftarrow \sum_{i=1}^m \mathbf{g}(\mathbf{x}_t + \mathbf{z}_t^{(i)})$
    $\mathbf{x}_{t+1} \leftarrow \mathbf{x}_t - \eta(\mathbf{g}_t(\mathbf{x}_t) + \xi_t),$      $\xi_t$ uniformly $\sim \mathbb{B}_0(r)$
  **return** $\mathbf{x}_T$

---

**Theorem 30** (Rates for Algorithm 2). *Given that the function pair $(F, f)$ satisfies Assumption A1 with $\tilde{\nu} \leq O(\epsilon/\sqrt{d})$, then for any $\delta > 0$, with smoothing parameter $\sigma = \Theta(\sqrt{\epsilon/(\rho d)})$, learning rate $\eta = 1/\ell$, perturbation $r = \tilde{\Theta}(\epsilon)$ and large mini-batch size $m = poly(d, B, \ell, \rho, 1/\epsilon, \log(1/\delta))$, FPSGD will find an $\epsilon$-second-order stationary point of $F$ with probability $1 - \delta$, in $poly(d, B, \ell, \rho, 1/\epsilon, \log(1/\delta))$ number of queries.*

Note that Algorithm 2 doesn't require oracle access to $f$, only to $\mathbf{g}$. We also observe the tolerance on $\tilde{\nu}$ is much better compared to Theorem 7, as noisy gradient information is available here while only noisy function value is avaliable in Theorem 7. The proof of this theorem can be found in Appendix F.

# F   Proof of Extension: Gradients pointwise close

This section proceeds similarly as in section A with the exception that all the results are now in terms of the gradient error, $\tilde{\nu}$. First, we present the gradient and Hessian smoothing identities (10 and 11) that we use extensively in the proofs. In section F.1, we present and prove the key lemma on the properties of the smoothed function $\tilde{f}_\sigma(\mathbf{x})$. Next, in section F.2, we prove the properties of the stochastic gradient $\mathbf{g}(\mathbf{x} + \mathbf{z})$. Then, using these lemmas, in section F.3 we prove a main theorem about the guarantees of FPSGD (Theorem 30). For clarity, we defer all technical lemmas and their proofs to section F.4.

Recall the definition of the gradient smoothing of a function given in Definition 11. In this section we will consider a smoothed version of the (possibly erroneous) gradient oracle, defined as follows.

$$\nabla \tilde{f}_\sigma(\mathbf{x}) = \mathbb{E}_\mathbf{z} \nabla f(\mathbf{x} + \mathbf{z}). \tag{10}$$

Note that indeed $\nabla \tilde{f}_\sigma(\mathbf{x}) = \nabla \mathbb{E}_\mathbf{z} f(\mathbf{x} + \mathbf{z})$. We can also write down following identity for the Hessian of the smoothed function.

$$\nabla^2 \tilde{f}_\sigma(\mathbf{x}) = \mathbb{E}_\mathbf{z}[\frac{\mathbf{z}}{\sigma^2} \nabla f(\mathbf{x} + \mathbf{z})^\top] \tag{11}$$

The proof is a simple calculation.

*Proof of Equation 11.* We proceed by exchanging the order of differentiation. The last equality follows from applying lemma 2 to the function $\frac{\partial}{\partial x_j} f(\mathbf{x} + \mathbf{z})$

$$\frac{\partial}{\partial x_i \partial x_j} \tilde{f}_\sigma(\mathbf{x}) = \frac{\partial}{\partial x_i} \frac{\partial}{\partial x_j} \mathbb{E}_\mathbf{z}[f(\mathbf{x} + \mathbf{z})] = \frac{\partial}{\partial x_i} \mathbb{E}_\mathbf{z}[\frac{\partial}{\partial x_j} f(\mathbf{x} + \mathbf{z})] = \mathbb{E}_\mathbf{z}[\frac{z_i}{\sigma^2} \frac{\partial}{\partial x_j} f(\mathbf{x} + \mathbf{z})]$$

$\square$

## F.1   Properties of the Gaussian smoothing

In this section, we show the properties of smoothed function $\nabla \tilde{f}_\sigma(\mathbf{x})$.

**Lemma 31** (Property of smoothing). *Assume function pair $(F, f)$ satisfies Assumption A1, and let $\nabla \tilde{f}_\sigma(\mathbf{x})$ be as given in equation 10. Then, the following holds*

    *1. $\tilde{f}_\sigma(\mathbf{x})$ is $O(\ell + \frac{\tilde{\nu}}{\sigma})$-gradient Lipschitz and $O(\rho + \frac{\tilde{\nu}}{\sigma^2})$-Hessian Lipschitz.*

2. $\|\nabla \tilde{f}_\sigma(\mathbf{x}) - \nabla F(\mathbf{x})\| \leq O(\rho d \sigma^2 + \tilde{\nu})$ *and* $\|\nabla^2 \tilde{f}_\sigma(\mathbf{x}) - \nabla^2 F(\mathbf{x})\| \leq O(\rho \sqrt{d} \sigma + \frac{\tilde{\nu}}{\sigma})$.

We will prove the 4 claims of the lemma one by one, in the following 4 sub-subsections.

### F.1.1 Gradient Lipschitz

We bound the gradient Lipschitz constant of $\tilde{f}_\sigma$ in the following lemma.

**Lemma 32** (Gradient Lipschitz of $\tilde{f}_\sigma$ under gradient closeness). $\|\nabla^2 \tilde{f}_\sigma(\mathbf{x})\| \leq O(\ell + \frac{\tilde{\nu}}{\sigma})$.

*Proof.* By triangle inequality,

$$
\begin{aligned}
\|\nabla^2 \tilde{f}_\sigma(\mathbf{x})\| &= \|\nabla^2 \tilde{F}_\sigma(\mathbf{x}) + \nabla^2 \tilde{f}_\sigma(\mathbf{x}) - \nabla^2 \tilde{F}_\sigma(\mathbf{x})\| \\
&\leq \|\nabla^2 \tilde{F}_\sigma(\mathbf{x})\| + \|\nabla^2 \tilde{f}_\sigma(\mathbf{x}) - \nabla^2 \tilde{F}_\sigma(\mathbf{x})\| \\
&\leq \ell + \|\mathbb{E}_\mathbf{z}[\frac{\mathbf{z}}{\sigma^2}(\nabla f - \nabla F)(\mathbf{x} + \mathbf{z})^\top]\| \\
&\leq O(\ell + \frac{\tilde{\nu}}{\sigma})
\end{aligned}
$$

The last inequality follows from Lemma 38. $\qquad\square$

### F.1.2 Hessian Lipschitz

We bound the Hessian Lipschitz constant of $\tilde{f}_\sigma$ in the following lemma.

**Lemma 33** (Hessian Lipschitz of $\tilde{f}_\sigma$ under gradient closeness).

$$
\|\nabla^2 \tilde{f}_\sigma(\mathbf{x}) - \nabla^2 \tilde{f}_\sigma(\mathbf{y})\| \leq O(\rho + \frac{\tilde{\nu}}{\sigma^2})\|\mathbf{x} - \mathbf{y}\|.
$$

*Proof.* By triangle inequality:

$$
\begin{aligned}
&\|\nabla^2 \tilde{f}_\sigma(\mathbf{x}) - \nabla^2 \tilde{f}_\sigma(\mathbf{y})\| \\
=&\|\nabla^2 \tilde{f}_\sigma(\mathbf{x}) - \nabla^2 \tilde{F}_\sigma(\mathbf{x}) - \nabla^2 \tilde{f}_\sigma(\mathbf{y}) + \nabla^2 \tilde{F}_\sigma(\mathbf{y}) + \nabla^2 \tilde{F}_\sigma(\mathbf{x}) - \nabla^2 \tilde{F}_\sigma(\mathbf{y})\| \\
\leq&\|\nabla^2 \tilde{f}_\sigma(\mathbf{x}) - \nabla^2 \tilde{F}_\sigma(\mathbf{x}) - (\nabla^2 \tilde{f}_\sigma(\mathbf{y}) - \nabla^2 \tilde{F}_\sigma(\mathbf{y}))\| + \|\nabla^2 \tilde{F}_\sigma(\mathbf{x}) - \nabla^2 \tilde{F}_\sigma(\mathbf{y})\| \\
=&O(\frac{\tilde{\nu}}{\sigma^2})\|\mathbf{x} - \mathbf{y}\| + O(\rho)\|\mathbf{x} - \mathbf{y}\| + O(\|\mathbf{x} - \mathbf{y}\|^2)
\end{aligned}
$$

The last inequality follows from Lemmas 10 and 39. $\qquad\square$

### F.1.3 Gradient Difference

We bound the difference between the gradients of smoothed function $\tilde{f}_\sigma(\mathbf{x})$ and those of the true objective $F$.

**Lemma 34** (Gradient Difference under gradient closeness). $\|\nabla \tilde{f}_\sigma(\mathbf{x}) - \nabla F(\mathbf{x})\| \leq O(\rho d \sigma^2 + \tilde{\nu})$.

*Proof.* By triangle inequality:

$$
\begin{aligned}
\|\nabla \tilde{f}_\sigma(\mathbf{x}) - \nabla F(\mathbf{x})\| &\leq \|\nabla \tilde{f}_\sigma(\mathbf{x}) - \nabla \tilde{F}_\sigma(\mathbf{x})\| + \|\nabla \tilde{F}_\sigma(\mathbf{x}) - \nabla F(\mathbf{x})\| \\
&\leq \|\mathbb{E}_\mathbf{z}[(\nabla f - \nabla F)(\mathbf{x} + \mathbf{z})]\| + O(\rho d \sigma^2) \quad (12) \\
&\leq O(\tilde{\nu} + \rho d \sigma^2)
\end{aligned}
$$

The inequality at (12) follows from Lemma 14. $\qquad\square$

### F.1.4 Hessian Difference

We bound the difference between the Hessian of smoothed function $\tilde{f}_\sigma(\mathbf{x})$ and that of the true objective $F$.

**Lemma 35** (Hessian Difference under gradient closeness). $\|\nabla^2 \tilde{f}_\sigma(\mathbf{x}) - \nabla^2 F(\mathbf{x})\| \leq O(\rho\sqrt{d}\sigma + \frac{\tilde{\nu}}{\sigma})$

*Proof.* By triangle inequality:

$$\|\nabla^2 \tilde{f}_\sigma(\mathbf{x}) - \nabla^2 F(\mathbf{x})\| \leq \|\nabla^2 \tilde{F}_\sigma(\mathbf{x}) - \nabla^2 F(\mathbf{x})\| + \|\nabla^2 \tilde{f}_\sigma(\mathbf{x}) - \nabla^2 \tilde{F}_\sigma(\mathbf{x})\|$$
$$\leq O(\rho\sqrt{d}\sigma + \frac{\tilde{\nu}}{\sigma})$$

The last inequality follows from Lemma 6 and 32. $\qquad\square$

### F.2 Properties of the stochastic gradient

**Lemma 36** (Stochastic gradient $\mathbf{g}(\mathbf{x};\mathbf{z})$). *Let* $\mathbf{g}(\mathbf{x};\mathbf{z}) = \nabla f(\mathbf{x} + \mathbf{z})$, $\mathbf{z} \sim N(0, \sigma\mathbf{I})$. *Then* $\mathbb{E}_{\mathbf{z}}\mathbf{g}(\mathbf{x};\mathbf{z}) = \nabla\tilde{f}_\sigma(\mathbf{x})$ *and* $\mathbf{g}(\mathbf{x};\mathbf{z})$ *is sub-Gaussian with parameter L.*

*Proof.* For the first claim we simply compute:

$$\mathbb{E}_{\mathbf{z}}\mathbf{g}(\mathbf{x};\mathbf{z}) = \mathbb{E}_{\mathbf{z}}\nabla f(\mathbf{x} + \mathbf{z}) = \nabla\mathbb{E}_{\mathbf{z}}[f(\mathbf{x} + \mathbf{z})] = \nabla\tilde{f}_\sigma(\mathbf{x}).$$

For the second claim, since function $f$ is L-Lipschitz, we know $\|\mathbf{g}(\mathbf{x},\mathbf{z})\| = \|\nabla f(\mathbf{x} + \mathbf{z})\| \leq L$. This implies that $\mathbf{g}(\mathbf{x};\mathbf{z})$ is sub-Gaussian with parameter $L$. $\qquad\square$

### F.3 Proof of Theorem 30

Using the properties proved in Lemma 36, we can apply Theorem 15 to find an $\epsilon$-SOSP for $\tilde{f}_\sigma$.

We now use lemma 31 to prove that any $\frac{\epsilon}{\sqrt{d}}$-SOSP of $\tilde{f}_\sigma(\mathbf{x})$ is also an $O(\epsilon)$-SOSP of $F$.

**Lemma 37** (SOSP of $\tilde{f}_\sigma(\mathbf{x})$ and SOSP of $F(\mathbf{x})$). *Suppose* $\mathbf{x}^*$ *satisfies*

$$\|\nabla\tilde{f}_\sigma(\mathbf{x}^*)\| \leq \tilde{\epsilon} \text{ and } \lambda_{min}(\nabla^2\tilde{f}_\sigma(\mathbf{x}^*)) \geq -\sqrt{\tilde{\rho}\tilde{\epsilon}},$$

*where* $\tilde{\rho} = \rho + \frac{\tilde{\nu}}{\sigma^2}$ *and* $\tilde{\epsilon} = \epsilon/\sqrt{d}$. *Then there exists constants* $c_1, c_2$ *such that*

$$\sigma \leq c_1\sqrt{\frac{\epsilon}{\rho d}}, \ \tilde{\nu} \leq c_2\frac{\epsilon}{\sqrt{d}}$$

*implies* $\mathbf{x}^*$ *is an* $O(\epsilon)$*-SOSP of* $F$.

*Proof.* By Lemma 31 and Weyl's inequality, we have that the following inequalities hold up to a constant factor:

$$\|\nabla F(\mathbf{x}^*)\| \leq \rho d\sigma^2 + \tilde{\nu} + \tilde{\epsilon}$$
$$\lambda_{\min}(\nabla^2 F(\mathbf{x}^*)) \geq \lambda_{\min}(\nabla^2\tilde{f}_\sigma(\mathbf{x}^*)) + \lambda_{\min}(\nabla^2 F(\mathbf{x}^*) - \nabla^2\tilde{f}_\sigma(\mathbf{x}^*)) \text{ (Weyl's theorem)}$$
$$\geq -\sqrt{(\rho + \frac{\tilde{\nu}}{\sigma^2})\tilde{\epsilon}} - \|\nabla^2\tilde{f}_\sigma(\mathbf{x}) - \nabla^2 F(\mathbf{x})\|$$
$$= -\sqrt{\frac{(\rho + \frac{\tilde{\nu}}{\sigma^2})}{\sqrt{d}}\epsilon} - (\rho\sqrt{d}\sigma + \frac{\tilde{\nu}}{\sigma})$$

Suppose we want any $\tilde{\epsilon}$-SOSP of $\tilde{f}_\sigma(\mathbf{x})$ to be a $O(\epsilon)$-SOSP of $F$. Then the following is sufficient (up to a constant factor):

$$\rho\sqrt{d}\sigma + \frac{\tilde{\nu}}{\sigma} \leq \sqrt{\rho\epsilon} \tag{13}$$
$$\rho d\sigma^2 + \tilde{\nu} \leq \epsilon \tag{14}$$
$$\rho + \frac{\tilde{\nu}}{\sigma^2} \leq \rho\sqrt{d} \tag{15}$$

We know Eq.(13), (14) $\implies \sigma \le \frac{\sqrt{\rho\epsilon}}{\rho\sqrt{d}} = \sqrt{\frac{\epsilon}{\rho d}}$ and $\sigma \le \sqrt{\frac{\epsilon}{\rho d}}$.

Also Eq. (13), (14) $\implies \tilde{\nu} \le \epsilon$ and $\tilde{\nu} \le \sqrt{\rho\epsilon}\sigma \le \sqrt{\rho\epsilon}\sqrt{\frac{\epsilon}{\rho d}} = \frac{\epsilon}{\sqrt{d}}$.

Finally Eq.(15) $\implies \tilde{\nu} \le \rho\sqrt{d}\sigma^2 \le \frac{\epsilon}{\sqrt{d}}$.

Thus the following choices ensures $\mathbf{x}^*$ is an $O(\epsilon)$-SOSP of $F$:

$$\sigma \le \sqrt{\frac{\epsilon}{\rho d}}, \ \tilde{\nu} \le \frac{\epsilon}{\sqrt{d}}.$$

<div align="right">□</div>

*Proof of Theorem 30.* Applying Theorem 15 on $\tilde{f}_\sigma(\mathbf{x})$ guarantees finding an $c\frac{\epsilon}{\sqrt{d}}$-SOSP of $\tilde{f}_\sigma(\mathbf{x})$ in number of queries polynomial in all the problem parameters. By Lemma 37, for some universal constant $c$, this is also an $\epsilon$-SOSP of $F$. This proves Theorem 30. <div align="right">□</div>

## F.4 Technical lemmas

In this section, we collect and prove the technical lemmas used in section F.

**Lemma 38.** *Let* $\mathbf{z} \sim N(0, \sigma^2\mathbf{I})$, $g : \mathbb{R}^d \to \mathbb{R}^d$, *and* $\exists a \in \mathbb{R}^+$ *s.t.* $\|g(\mathbf{x})\| \le a \forall \mathbf{x} \in \mathbb{R}^d$. *Let* $\Delta \in \mathbb{R}^d$ *be fixed. Then,*

$$\|\mathbb{E}_{\mathbf{z}}[\mathbf{z}g(\mathbf{z})^\top]\| \le \sigma a; \tag{16}$$

$$\|\mathbb{E}_{\mathbf{z}}[\mathbf{z}\langle \mathbf{z}, \Delta \rangle g(\mathbf{z})^\top]\| \le \frac{a}{\sigma^2}. \tag{17}$$

*Proof.*

$$
\begin{aligned}
(16) : \|\mathbb{E}_z[\mathbf{z}g(\mathbf{z})^\top]\| &= \sup_{\mathbf{v}\in\mathbb{R}^d, \|\mathbf{v}\|=1} \mathbf{v}^\top (\mathbb{E}_z[\mathbf{z}g(\mathbf{z})^\top])\mathbf{v} \\
&= \mathbb{E}_{\mathbf{z}}[\mathbf{v}^{*\top}\mathbf{z}g(\mathbf{z})^\top\mathbf{v}] \\
&\le \sqrt{\mathbb{E}_{\mathbf{z}}[(\mathbf{v}^{*\top}\mathbf{z})^2]\mathbb{E}[(g(\mathbf{z})^\top\mathbf{v}^*)^2]} \\
&\le \sqrt{\sigma^2 a^4} \text{ since } \mathbf{v}^{*\top}\mathbf{z} \sim N(0, 1) \\
(17) : \|\mathbb{E}_{\mathbf{z}}[\mathbf{z}\langle \mathbf{z}, \Delta\rangle g(\mathbf{z})^\top]\| &= \sup_{\mathbf{v}\in\mathbb{R}^d, \|\mathbf{v}\|=1} \mathbf{v}^\top \mathbb{E}_{\mathbf{z}}[\mathbf{z}\langle \mathbf{z}, \Delta\rangle g(\mathbf{z})^\top]\mathbf{v} \\
&= \mathbb{E}_{\mathbf{z}}[\langle \mathbf{v}^*, \mathbf{z}\rangle\langle \mathbf{z}, \Delta\rangle\langle g(\mathbf{z}), \mathbf{v}^*\rangle] \\
&\le a\mathbb{E}_{\mathbf{z}}[|\langle \mathbf{v}^*, \mathbf{z}\rangle\langle \mathbf{z}, \Delta\rangle|] \\
&\le a\sqrt{\mathbb{E}_{\mathbf{z}}[\langle \mathbf{v}^*, \mathbf{z}\rangle^2]\mathbb{E}_{\mathbf{z}}[\langle \mathbf{z}, \Delta\rangle^2]} \\
&\le a\|\Delta\|\sigma^2.
\end{aligned}
$$

<div align="right">□</div>

**Lemma 39.** $\|\nabla^2\tilde{f}_\sigma(\mathbf{x}) - \nabla^2\tilde{F}_\sigma(\mathbf{x}) - (\nabla^2\tilde{f}_\sigma(\mathbf{y}) - \nabla^2\tilde{F}_\sigma(\mathbf{y}))\| \le O(\frac{\tilde{\nu}}{\sigma^2})\|\mathbf{x} - \mathbf{y}\| + O(\|\mathbf{x} - \mathbf{y}\|^2)$

*Proof.* For brevity, denote $h = \frac{1}{(2\pi\sigma^2)^{\frac{d}{2}}}$. We have:

$$
\begin{aligned}
&\nabla^2\tilde{f}_\sigma(\mathbf{x}) - \nabla^2\tilde{F}_\sigma(\mathbf{x}) - (\nabla^2\tilde{f}_\sigma(\mathbf{y}) - \nabla^2\tilde{F}_\sigma(\mathbf{y})) \\
&= \mathbb{E}_{\mathbf{z}}[\frac{\mathbf{z}}{\sigma^2}((\nabla f - \nabla F)(\mathbf{x} + \mathbf{z}) - (\nabla f - \nabla F)(\mathbf{y} + \mathbf{z}))^\top] \\
&= h\left(\int \frac{\mathbf{z}}{\sigma^2}(\nabla f - \nabla F)(\mathbf{x} + \mathbf{z})^\top e^{-\frac{\|\mathbf{z}\|^2}{2\sigma^2}}d\mathbf{z} - \int \frac{\mathbf{z}}{\sigma^2}(\nabla f - \nabla F)(\mathbf{y} + \mathbf{z})^\top e^{-\frac{\|\mathbf{z}\|^2}{2\sigma^2}}d\mathbf{z}\right) \\
&= h\left(\int \left((\mathbf{z} + \Delta)e^{-\frac{\|\mathbf{z}+\Delta\|^2}{2\sigma^2}} - (\mathbf{z} - \Delta)e^{-\frac{\|\mathbf{z}-\Delta\|^2}{2\sigma^2}}\right)(\nabla f - \nabla F)(\mathbf{z} + \frac{\mathbf{x}+\mathbf{y}}{2})^\top d\mathbf{z}\right), \quad (18)
\end{aligned}
$$

<div align="center">22</div>

where $\Delta = \frac{\mathbf{y}-\mathbf{x}}{2}$. The last equality follows from a change of variables. Now denote $\mathbf{g}(\mathbf{z}) := (\nabla f - \nabla F)(\mathbf{z} + \frac{\mathbf{x}+\mathbf{y}}{2})$. By a Taylor expansion up to only the first order terms in $\Delta$, we have

$$(18) - O(\|\Delta\|)^2 = h(\int ((\mathbf{z}+\Delta)(1 - \frac{\langle \mathbf{z}, \Delta \rangle}{\sigma^2}) - (\mathbf{z} - \Delta)(1 + \frac{\langle \mathbf{z}, \Delta \rangle}{\sigma^2}))g(\mathbf{z})^\top e^{-\frac{\|\mathbf{z}\|^2}{2\sigma^2}} d\mathbf{z}$$

$$= 2h(\int (\Delta - \mathbf{z}\frac{\langle \mathbf{z}, \Delta \rangle}{\sigma^2})g(\mathbf{z})^\top e^{-\frac{\|\mathbf{z}\|^2}{2\sigma^2}} d\mathbf{z}$$

$$= 2\mathbb{E}_{\mathbf{z}}[(\Delta - \mathbf{z}\frac{\langle \mathbf{z}, \Delta \rangle}{\sigma^2})g(\mathbf{z})^\top].$$

Therefore,

$$\|\nabla^2 \tilde{f}_\sigma(\mathbf{x}) - \nabla^2 \tilde{F}_\sigma(\mathbf{x}) - (\nabla^2 \tilde{f}_\sigma(\mathbf{y}) - \nabla^2 \tilde{F}_\sigma(\mathbf{y}))\|$$

$$\leq \frac{2}{\sigma^2}\|\mathbb{E}_{\mathbf{z}}[(\Delta - \frac{\langle \mathbf{z}, \Delta \rangle}{\sigma^2})g(\mathbf{z})^\top\| + O(\|\Delta\|^2)$$

$$\leq \frac{2}{\sigma^2}\|\mathbb{E}_{\mathbf{z}}[\Delta g(\mathbf{z})^\top]\| + \frac{2}{\sigma^4}\|\mathbb{E}_{\mathbf{z}}[\langle \mathbf{z}, \Delta \rangle g(\mathbf{z})^\top]\| + O(\|\Delta\|^2)$$

$$\leq \frac{2}{\sigma^2}\tilde{\nu}\|\Delta\| + \frac{2}{\sigma^2}\tilde{\nu}\|\Delta\| + O(\|\Delta\|^2).$$

The last inequality follows from Lemma 38.

$\square$

# G  Proof of Learning ReLU Unit

In this section we analyze the population loss of the simple example of a single ReLU unit.

Recall our assumption that $\|\mathbf{w}^\star\| = 1$ and that the data distribution is $\mathbf{x} \sim \mathcal{N}(0, \mathbf{I})$; thus,

$$y_i = \text{ReLU}(\mathbf{x}_i^\top \mathbf{w}^\star) + \zeta_i, \qquad \zeta_i \sim \mathcal{N}(0, 1).$$

We use the squared loss as the loss function, hence writing the empirical loss as:

$$\hat{R}_n(\mathbf{w}) = \frac{1}{2n}\sum_{i=1}^{n}(y_i - \text{ReLU}(\mathbf{x}_i^\top \mathbf{w}))^2.$$

The main tool we use is a closed-form formula for the kernel function defined by ReLU gates.

**Lemma 40.** *[Cho and Saul, 2009] For fixed* $\mathbf{u}, \mathbf{v}$, *if* $\mathbf{x} \sim \mathcal{N}(0, \mathbf{I})$, *then*

$$\mathbb{E}\, ReLU(\mathbf{x}^\top \mathbf{u}) \cdot ReLU(\mathbf{x}^\top \mathbf{v}) = \frac{1}{2\pi}\|\mathbf{u}\|\|\mathbf{v}\|[\sin\theta + (\pi - \theta)\cos\theta],$$

*where* $\theta$ *is the angel between* $\mathbf{u}$ *and* $\mathbf{v}$ *satisfying* $\cos\theta = \mathbf{u}^\top \mathbf{v}/(\|\mathbf{u}\|\|\mathbf{v}\|)$.

Then, the population loss has the following analytical form:

$$R(\mathbf{w}) = \frac{1}{4}\|\mathbf{w}\|^2 + \frac{5}{4} - \frac{1}{2\pi}\|\mathbf{w}\|[\sin\theta + (\pi - \theta)\cos\theta],$$

and so does the gradient ($\hat{\mathbf{w}}$ is the unit vector along $\mathbf{w}$ direction):

$$\nabla R(\mathbf{w}) = \frac{1}{2}(\mathbf{w} - \mathbf{w}^\star) + \frac{1}{2\pi}(\theta\mathbf{w}^\star - \hat{\mathbf{w}}\sin\theta).$$

## G.1  Properties of Population Loss

We first prove the properties of the population loss, which were stated in Lemma 16 and we also restate the lemma below. Let $\mathfrak{B} = \{\mathbf{w}|\mathbf{w}^\top\mathbf{w}^\star \geq \frac{1}{\sqrt{d}}\} \cap \{\mathbf{w}|\|\mathbf{w}\| \leq 2\}$.

**Lemma 41.** *The population and empirical risk* $R, \hat{R}_n$ *of learning a ReLU unit problem satisfies:*

1. *If $\mathbf{w}_0 \in \mathfrak{B}$, then runing ZPSGD (Algorithm 1) gives $\mathbf{w}_t \in \mathfrak{B}$ for all $t$ with high probability.*

2. *Inside $\mathfrak{B}$, $R$ is $O(1)$-bounded, $O(\sqrt{d})$-gradient Lipschitz, and $O(d)$-Hessian Lipschitz.*

3. $\sup_{\mathbf{w}\in\mathfrak{B}} |\hat{R}_n(\mathbf{w}) - R(\mathbf{w})| \leq \tilde{O}(\sqrt{d/n})$ *w.h.p.*

4. *Inside $\mathfrak{B}$, $R$ is nonconvex function, $\mathbf{w}^\star$ is the only SOSP of $R(\mathbf{w})$.*

To prove these four claims, we require following lemmas.

The first important property we use is that the gradient of population loss $l$ has the one-point convex property inside $\mathfrak{B}$, stated as follows:

**Lemma 42.** *Inside $\mathfrak{B}$, we have:*

$$\langle -\nabla R(\mathbf{w}), \mathbf{w}^\star - \mathbf{w}\rangle \geq \frac{1}{10}\|\mathbf{w} - \mathbf{w}^\star\|^2.$$

*Proof.* Note that inside $\mathfrak{B}$, we have the angle $\theta \in [0, \pi/2)$. Also, let $\mathfrak{W}_\theta = \{\mathbf{w}|\angle(\mathbf{w}, \mathbf{w}^\star) = \theta\}$, then for $\theta \in [0, \pi/2)$:

$$\min_{\mathbf{w}\in\mathfrak{W}_\theta} \|\mathbf{w} - \mathbf{w}^\star\| = \sin\theta.$$

On the other hand, note that $\theta \leq 2\sin\theta$ holds true for $\theta \in [0, \pi/2)$; thus we have:

$$\begin{aligned}
\langle -\nabla R(\mathbf{w}), \mathbf{w} - \mathbf{w}^\star\rangle =& \langle \frac{1}{2}(\mathbf{w} - \mathbf{w}^\star) + \frac{1}{2\pi}(\theta\mathbf{w}^\star - \hat{\mathbf{w}}\sin\theta), \mathbf{w} - \mathbf{w}^\star\rangle \\
=& \frac{1}{2}\|\mathbf{w} - \mathbf{w}^\star\|^2 + \frac{1}{2\pi}\langle[\mathbf{w}^\star(\theta - \sin\theta) + (\mathbf{w}^\star - \hat{\mathbf{w}})\sin\theta], \mathbf{w} - \mathbf{w}^\star\rangle \\
\geq& \frac{1}{2}\|\mathbf{w} - \mathbf{w}^\star\|^2 - \frac{1}{2\pi}(\sin\theta + \sqrt{2}\sin\theta)\|\mathbf{w} - \mathbf{w}^\star\| \\
\geq& (\frac{1}{2} - \frac{1+\sqrt{2}}{2\pi})\|\mathbf{w} - \mathbf{w}^\star\|^2 \geq \frac{1}{10}\|\mathbf{w} - \mathbf{w}^\star\|^2,
\end{aligned}$$

where the second last inequality used the fact that $\sin\theta \leq \|\mathbf{w} - \mathbf{w}^\star\|$ for all $\mathbf{w} \in \mathfrak{B}$. $\qquad\square$

One-point convexity guarantees that ZPSGD stays in the region $\mathfrak{B}$ with high probability.

**Lemma 43.** *ZPSGD (Algorithm 1) with proper hyperparameters will stay in $\mathfrak{B}$ with high probability.*

*Proof.* We prove this by two steps:

1. The algorithm always moves towards $\mathbf{x}^\star$ in the region $\mathfrak{B} - \{\|\mathbf{w} - \mathbf{w}^\star\| \leq 1/10\}$.

2. The algorithm will not jump from $\{\|\mathbf{w}\| \leq 1/10\}$ to $\mathfrak{B}^c$ in one step.

The second step is rather straightforward since the function $\ell(\mathbf{w})$ is Lipschitz, and the learning rate is small. The first step is due to the large minibatch size and the concentration properties of sub-Gaussian random variables:

$$\begin{aligned}
\|\mathbf{w}_{t+1} - \mathbf{w}^\star\|^2 =& \|\mathbf{w}_t - \eta(\mathbf{g}_t(\mathbf{x}_t) + \xi_t) - \mathbf{w}^\star\|^2 \\
\leq& \|\mathbf{w}_t - \mathbf{w}^\star\|^2 - \eta\langle\nabla f_\sigma(\mathbf{x}_t), \mathbf{w}_t - \mathbf{w}^\star\rangle + \eta\|\zeta_t\|\|\mathbf{w}_t - \mathbf{w}^\star\| + \eta^2\mathbb{E}\|\mathbf{g}_t(\mathbf{x}_t) + \xi_t\|^2 \\
\leq& \|\mathbf{w}_t - \mathbf{w}^\star\|^2 - \frac{\eta}{10}\|\mathbf{w}_t - \mathbf{w}^\star\|^2 + \eta\epsilon\|\mathbf{w}_t - \mathbf{w}^\star\| + \eta^2\mathbb{E}\|\mathbf{g}_t(\mathbf{x}_t) + \xi_t\|^2 \\
\leq& \|\mathbf{w}_t - \mathbf{w}^\star\|^2 - (\frac{\eta}{100} - \eta\epsilon - O(\eta^2))\|\mathbf{w}_t - \mathbf{w}^\star\| \leq 0
\end{aligned}$$

The last step is true when we pick a learning rate that is small enough (although we pick $\eta = 1/\ell$, this is still fine because a $\ell$-gradient Lipschitz function is clearly also a $10\ell$-gradient Lipschitz function) and $\epsilon$ is small. $\qquad\square$

**Lemma 44.** *Let* $\mathbf{w}(t) = \frac{1}{5}(\mathbf{w}^\star + t\mathbf{e})$ *where* $\mathbf{e}$ *is any direction so that* $\mathbf{e}^\top \mathbf{w}^\star = 0$

$$R(\mathbf{w}(t)) = \frac{t^2}{100} - \frac{t}{10\pi} + \frac{1}{10\pi}\tan^{-1}(t) + const,$$

*which is nonconvex in domain* $t \in [0,1]$. *Therefore* $f(\mathbf{w}(t))$ *is nonconvex along this line segment inside* $\mathfrak{B}$.

*Proof.* Note that in above setup, $\tan\theta = t$, so the population loss can be calculated as:

$$R(\mathbf{w}(t)) = \frac{1}{100}\|\mathbf{w}\|^2 - \frac{1}{2\pi}[\frac{t}{5} - \tan^{-1}(t)\cdot\frac{1}{5}] + const$$

$$= \frac{t^2}{100} - \frac{t}{10\pi} + \frac{1}{10\pi}\tan^{-1}(t) + const$$

It's easy to show $\mathbf{w}(t) \in \mathfrak{B}$ for all $t \in [0,1]$ and if $g(t) = R(\mathbf{w}(t))$, then $g''(0.6) < 0$ and thus the function is nonconvex. $\qquad\square$

Next, we show that the empirical risk and the population risk are close by a covering argument.

**Lemma 45.** *For sample size* $n \geq d$, *with high probability, we have:*

$$\sup_{\mathbf{w}\in\mathfrak{B}} |\hat{R}_n(\mathbf{w}) - R(\mathbf{w})| \leq \tilde{O}\left(\sqrt{\frac{d}{n}}\right).$$

*Proof.* Let $\{\mathbf{w}^j\}_{j=1}^J$ be a $\epsilon$-covering of $\mathfrak{B}$. By triangular inequality:

$$\sup_{\mathbf{w}\in\mathfrak{B}} |\hat{R}_n(\mathbf{w}) - R(\mathbf{w})| \leq \underbrace{\sup_{\mathbf{w}\in\mathfrak{B}} |\hat{R}_n(\mathbf{w}) - \hat{R}_n(\mathbf{w}^j)|}_{T_1} + \underbrace{\sup_{j\in J} |\hat{R}_n(\mathbf{w}^j) - R(\mathbf{w}^j)|}_{T_2} + \underbrace{\sup_{\mathbf{w}\in\mathfrak{B}} |R(\mathbf{w}^j) - R(\mathbf{w})|}_{T_3},$$

where $\mathbf{w}^j$ is the closest point in the cover to $\mathbf{w}$. Clearly, the $\epsilon$-net of $\mathfrak{B}$ requires fewer points than the $\epsilon$-net of $\{\mathbf{w}|\|\mathbf{w}\| \leq 2\}$. By the standard covering number argument, we have $\log N_\epsilon = O(d\log\frac{1}{\epsilon})$. We proceed to bound each term individually.

**Term $T_2$:** For a fixed $j$, we know $\hat{R}_n(\mathbf{w}^j) = \frac{1}{n}\sum_{i=1}^n(y_i - \text{ReLU}(\mathbf{x}_i^\top\mathbf{w}^j))^2$, where $y_i - \text{ReLU}(\mathbf{x}_i^\top\mathbf{w}^j)$ is sub-Gaussian with parameter $O(1)$, thus $(y_i - \text{ReLU}(\mathbf{x}_i^\top\mathbf{w}^j))^2$ is sub-Exponential with parameter $O(1)$. We have the concentration inequality:

$$\mathbb{P}(|\hat{R}_n(\mathbf{w}^j) - R(\mathbf{w}^j)| \geq t) \leq e^{O(\frac{nt^2}{1+t})}.$$

By union bound, we have:

$$\mathbb{P}(\sup_{j\in J} |\hat{R}_n(\mathbf{w}^j) - R(\mathbf{w}^j)| \geq t) \leq N_\epsilon e^{O(\frac{nt^2}{1+t})}.$$

That is, with $n \geq d$, and probability $1 - \delta$, we have:

$$\sup_{j\in J} |\hat{R}_n(\mathbf{w}^j) - R(\mathbf{w}^j)| \leq \sqrt{\frac{1}{n}(\log\frac{1}{\delta} + d\log\frac{1}{\epsilon})}.$$

**Term $T_3$:** Since the population loss is $O(1)$-Lipschitz in $\mathfrak{B}$, we have:

$$\sup_{\mathbf{w}\in\mathfrak{B}} |R(\mathbf{w}^j) - R(\mathbf{w})| \leq L|\mathbf{w}^j - \mathbf{w}| \leq O(\epsilon).$$

**Term $T_1$:** Note that for a fixed pair $(\mathbf{x}_i, \mathbf{y}_i)$, the function $g_i(\mathbf{w}) = (y_i - \text{ReLU}(\mathbf{x}_i^\top\mathbf{w}))^2$ is $O(\|\zeta_i\|\|\mathbf{x}_i\| + \|\mathbf{x}_i\|^2)$-Lipschitz. Therefore,

$$\sup_{\mathbf{w}\in\mathfrak{B}} |\hat{R}_n(\mathbf{w}^j) - \hat{R}_n(\mathbf{w})| \leq O(1)\cdot\frac{1}{n}\sum_i \left[\|\zeta_i\|\|\mathbf{x}_i\| + \|\mathbf{x}_i\|^2\right]|\mathbf{w}^j - \mathbf{w}|$$

$$\leq O(\epsilon)\cdot\frac{1}{n}\sum_i \left[\|\zeta_i\|\|\mathbf{x}_i\| + \|\mathbf{x}_i\|^2\right].$$

With high probability, $\frac{1}{n}\sum_i \left[\|\zeta_i\|\|\mathbf{x}_i\| + \|\mathbf{x}_i\|^2\right]$ concentrates around its mean, $O(d)$.

In summary, we have:

$$\sup_{\mathbf{w}\in\mathfrak{B}} |\hat{R}_n(\mathbf{w}) - R(\mathbf{w})| \leq \sqrt{\frac{1}{n}(\log\frac{1}{\delta} + d\log\frac{1}{\epsilon})} + O(\epsilon) + O(\epsilon d).$$

By picking $\epsilon$ (for the $\epsilon$-covering) small enough, we finish the proof. $\qquad\square$

Finally we prove the smoothness of population risk in $\mathfrak{B}$, we have $1/\sqrt{d} \leq \|\mathbf{w}\| \leq 2$.

**Lemma 46.** *For population loss* $R(\mathbf{w}) = \frac{1}{4}\|\mathbf{w}\|^2 + \frac{5}{4} - \frac{1}{2\pi}\|\mathbf{w}\|[\sin\theta + (\pi - \theta)\cos\theta]$, *its gradient and Hessian are equal to:*

$$\nabla R(\mathbf{w}) = \frac{1}{2}(\mathbf{w} - \mathbf{w}^\star) + \frac{1}{2\pi}(\theta\mathbf{w}^\star - \hat{\mathbf{w}}\sin\theta),$$

$$\nabla^2 R(\mathbf{w}) = \begin{cases} \frac{1}{2}\mathbf{I} & \text{if } \theta = 0 \\ \frac{1}{2}\mathbf{I} - \frac{\sin\theta}{2\pi\|\mathbf{w}\|}(\mathbf{I} + \hat{\mathbf{u}}\hat{\mathbf{u}}^\top - \hat{\mathbf{w}}\hat{\mathbf{w}}^\top) & \text{otherwise} \end{cases},$$

*where* $\hat{\mathbf{w}}$ *is the unit vector along the* $\mathbf{w}$ *direction, and* $\hat{\mathbf{u}}$ *is the unit vector along the* $\mathbf{w}^\star - \hat{\mathbf{w}}\cos\theta$ *direction.*

*Proof.* Note $\|\mathbf{w}^\star\| = 1$. Let $z(\mathbf{w}, \mathbf{w}^\star) = \frac{\mathbf{w}^\top\mathbf{w}^\star}{\|\mathbf{w}\|}$, we have:

$$\nabla_{\mathbf{w}}z(\mathbf{w}, \mathbf{w}^\star) = \frac{\mathbf{w}^\star\|\mathbf{w}\| - (\mathbf{w}^\top\mathbf{w}^\star)\hat{\mathbf{w}}}{\|\mathbf{w}\|^2} = \frac{(\mathbf{w}^\star - \hat{\mathbf{w}}\cos\theta)}{\|\mathbf{w}\|}.$$

Since $\cos\theta = z(\mathbf{w}, \mathbf{w}^\star)$, we obtain:

$$-\sin\theta \cdot \nabla\theta = \frac{(\mathbf{w}^\star - \hat{\mathbf{w}}\cos\theta)}{\|\mathbf{w}\|}.$$

This gives:

$$\begin{aligned}
\nabla R(\mathbf{w}) =& \frac{1}{2}\mathbf{w} - \frac{1}{2\pi}\hat{\mathbf{w}}[\sin\theta + (\pi - \theta)\cos\theta] - \frac{1}{2\pi}\|\mathbf{w}\|[\cos\theta - \cos\theta - (\pi - \theta)\sin\theta]\nabla\theta \\
=& \frac{1}{2}\mathbf{w} - \frac{1}{2\pi}\hat{\mathbf{w}}[\sin\theta + (\pi - \theta)\cos\theta] + \frac{1}{2\pi}\|\mathbf{w}\|(\pi - \theta)\sin\theta \cdot \nabla\theta \\
=& \frac{1}{2}\mathbf{w} - \frac{1}{2\pi}\hat{\mathbf{w}}[\sin\theta + (\pi - \theta)\cos\theta] - \frac{1}{2\pi}(\pi - \theta)(\mathbf{w}^\star - \hat{\mathbf{w}}\cos\theta) \\
=& \frac{1}{2}(\mathbf{w} - \mathbf{w}^\star) + \frac{1}{2\pi}(\theta\mathbf{w}^\star - \hat{\mathbf{w}}\sin\theta)
\end{aligned}$$

Therefore, the Hessian (when $\theta \neq 0$):

$$\begin{aligned}
\nabla^2 R(\mathbf{w}) =& \nabla[\frac{1}{2}(\mathbf{w} - \mathbf{w}^\star) + \frac{1}{2\pi}(\theta\mathbf{w}^\star - \hat{\mathbf{w}}\sin\theta)] \\
=& \frac{1}{2}\mathbf{I} + \frac{1}{2\pi}[\nabla\theta \cdot (\mathbf{w}^\star - \hat{\mathbf{w}}\cos\theta)^\top] - \frac{\sin\theta}{2\pi\|\mathbf{w}\|}(\mathbf{I} - \hat{\mathbf{w}}\hat{\mathbf{w}}^\top) \\
=& \frac{1}{2}\mathbf{I} - \frac{\sin\theta}{2\pi\|\mathbf{w}\|}(\mathbf{I} + \hat{\mathbf{u}}\hat{\mathbf{u}}^\top - \hat{\mathbf{w}}\hat{\mathbf{w}}^\top),
\end{aligned}$$

where $\hat{\mathbf{u}}$ is the unit vector along $\mathbf{w}^\star - \hat{\mathbf{w}}\cos\theta$ direction.

And for $\theta = 0$, Hessian $\nabla^2 R(\mathbf{w}) = \frac{1}{2}\mathbf{I}$. We prove this by taking the limit. For $\hat{\mathbf{v}} = \mathbf{w}^\star$

$$\nabla^2 R(\mathbf{w}) \cdot \hat{\mathbf{v}} = \lim_{\epsilon\to 0}\frac{\nabla R(\mathbf{w} + \epsilon\hat{\mathbf{v}}) - \nabla R(\mathbf{w})}{\epsilon} = \frac{1}{2}\hat{\mathbf{v}}.$$

For any $\hat{\mathbf{v}} \perp \mathbf{w}^{\star}$, the angle $\theta$ between $\mathbf{w} + \epsilon \hat{v}$ and $\mathbf{w}^{\star}$ is $\Theta(\frac{\epsilon}{\|\mathbf{w}\|})$ up to first order in $\epsilon$, we have:

$$\nabla^2 R(\mathbf{w}) \cdot \hat{\mathbf{v}} = \lim_{\epsilon \to 0} \frac{\nabla R(\mathbf{w} + \epsilon \hat{\mathbf{v}}) - \nabla R(\mathbf{w})}{\epsilon}$$
$$= \frac{1}{2}\hat{\mathbf{v}} + \frac{1}{2\pi} \lim_{\epsilon \to 0} \frac{\epsilon \mathbf{w}^{\star} - (\mathbf{w}^{\star} + \Theta(\frac{\epsilon}{\|\mathbf{w}\|})\hat{\mathbf{v}}) \cdot \Theta(\frac{\epsilon}{\|\mathbf{w}\|}) + o(\epsilon)}{\epsilon} = \frac{1}{2}\hat{\mathbf{v}}.$$

This finishes the proof. $\qquad\square$

**Lemma 47.** *The population loss function $R$ is $O(1)$-bounded, $O(1)$-Lipschitz, $O(\sqrt{d})$-gradient Lipschitz, and $O(d)$-Hessian Lipschitz.*

*Proof.* The bounded, Lipschitz, and gradient Lipschitz are all very straightforward given the formula of gradient and Hessian. We will focus on proving Hessian Lipschitz. Equivalently, we show upper bounds on following quantity:

$$\lim_{\epsilon \to 0} \frac{\|\nabla^2 R(\mathbf{w} + \epsilon \hat{\mathbf{v}}) - \nabla^2 R(\mathbf{w})\|}{\epsilon}.$$

Note that the change in $\theta$ is at most $O(\frac{\epsilon}{\|\mathbf{w}\|})$, we have:

$$\|\nabla^2 R(\mathbf{w} + \epsilon \hat{\mathbf{v}}) - \nabla^2 R(\mathbf{w})\| \le O(\frac{\epsilon}{\|\mathbf{w}\|^2}) + o(\epsilon).$$

This gives:

$$\lim_{\epsilon \to 0} \frac{\|\nabla^2 R(\mathbf{w} + \epsilon \hat{\mathbf{v}}) - \nabla^2 R(\mathbf{w})\|}{\epsilon} \le O(\frac{1}{\|\mathbf{w}\|^2}) \le O(d),$$

which finishes the proof.

$\qquad\square$

*Proof of Lemma 16.* For four claims in Lemma 16, claim 1 follows from Lemma 43; claim 2 follows from Lemma 47; claim 3 follows from Lemma 45; claim 4 follows from Lemma 44 and Lemma 42.
$\qquad\square$

## G.2 Proof of Theorem 17

*Proof.* The sample complexity $\tilde{O}(d^4/\epsilon^3)$ can be directly computed from Lemma 16 and Theorem 7.
$\qquad\square$

## H Proof of Stochastic gradient descent

Here for completeness we give the result for perturbed stochastic gradient descent, which is a adaptation of results in Jin et al. [2017] and will be formally presented in Jin et al. [2018].

Given stochastic gradient oracle $\mathbf{g}$, where $\mathbb{E}\mathbf{g}(\mathbf{x}; \theta) = \nabla f(\mathbf{x})$, and

**Assumption A2.** function $f$ satisfies following property:

- $f(\cdot)$ is $\ell$-gradient Lipschitz and $\rho$-Hessian Lipschitz.

- For any $\mathbf{x} \in \mathbb{R}^d$, $\mathbf{g}(\mathbf{x}; \theta)$ has sub-Gaussian tail with parameter $\sigma/\sqrt{d}$.

**Theorem 48.** *If function $f(\cdot)$ satisfies Assumption A2, then for any $\delta > 0$, with learning rate $\eta = 1/\ell$, perturbation $r = \tilde{\Theta}(\epsilon)$ and large mini-batch size $m = poly(d, B, \ell, \rho, \sigma, 1/\epsilon, \log(1/\delta))$, PSGD (Algorithm 3) will find $\epsilon$-second-order stationary point of $F$ with $1-\delta$ probability in following number of stochastic gradient queries:*

$$\tilde{O}\left(\frac{\ell \Delta_f}{\epsilon^2} \cdot m\right).$$

---

**Algorithm 3** Perturbed Stochastic Gradient Descent with Minibatch

---
**Input:** $\mathbf{x}_0$, learning rate $\eta$, noise radius $r$.
   **for** $t = 0, 1, \ldots,$ **do**
      sample $\{\theta_t^{(1)}, \cdots \theta_t^{(m)}\} \sim \mathcal{D}$
      $\mathbf{g}_t(\mathbf{x}_t) \leftarrow \sum_{i=1}^m \mathbf{g}(\mathbf{x}_t; \theta_t^{(i)})/m$
      $\mathbf{x}_{t+1} \leftarrow \mathbf{x}_t - \eta(\mathbf{g}_t(\mathbf{x}_t) + \xi_t),$       $\xi_t$ uniformly $\sim B_0(r)$
   **return** $\mathbf{x}_T$

---

In order to prove this theorem, let

$$\eta = \frac{1}{\ell}, \qquad \mathscr{T} = \frac{\chi c}{\eta \sqrt{\rho \epsilon}}, \quad \mathscr{F} = \sqrt{\frac{\epsilon^3}{\rho}} \chi^{-3} c^{-5}, \quad r = \epsilon \chi^{-3} c^{-6}, \quad m = \text{poly}(d, B, \ell, \rho, \sigma, 1/\epsilon, \log(1/\delta)),$$

$$(19)$$

where $c$ is some large constant and $\chi = \max\{1, \log \frac{d\ell \Delta_f}{\rho \epsilon \delta}\}$

**Lemma 49.** *for any $\lambda > 0, \delta > 0$, if minibatch size $m \geq \frac{2\lambda^2 \sigma^2}{\epsilon^2} \log \frac{d}{\delta}$, then for a fixed $\mathbf{x}$, with probability $1 - \delta$, we have:*

$$\|\nabla f(\mathbf{x}) - \frac{1}{m} \sum_{i=1}^m \mathbf{g}(\mathbf{x}; \theta^{(i)})\| \leq \frac{\epsilon}{\lambda}.$$

This lemma means, when mini-batch size is large enough, we can make noise in the stochastic gradient descent polynomially small.

**Lemma 50.** *Consider the setting of Theorem 48, if $\|\nabla f(\mathbf{x}_t)\| \geq \epsilon$, then by running Algorithm 1, with probability $1 - \delta$, we have $f(\mathbf{x}_{t+1}) - f(\mathbf{x}_t) \leq -\eta \epsilon^2 / 4$.*

*Proof.* By gradient Lipschitz, and the fact $\|\xi_t\| \leq \epsilon/20$ and with minibatch size $m$ large enough, with high probability we have $\|\nabla f(\mathbf{x}_t) - \mathbf{g}_t\| \leq \epsilon/20$. Let $\zeta_t = \mathbf{g}_t - \nabla f(\mathbf{x}_t) + \xi_t$, by triangle inequality, we have $\|\zeta_t\| \leq \epsilon/10$ and update equation $\mathbf{x}_{t+1} = \mathbf{x}_t - \eta(\nabla f(\mathbf{x}_t) + \zeta_t)$:

$$\begin{aligned} f(\mathbf{x}_{t+1}) \leq & f(\mathbf{x}_t) + \langle \nabla f(\mathbf{x}_t), \mathbf{x}_{t+1} - \mathbf{x}_t \rangle + \frac{\ell}{2} \|\mathbf{x}_{t+1} - \mathbf{x}_t\|^2 \\ \leq & f(\mathbf{x}_t) - \eta \|\nabla f(\mathbf{x}_t)\|^2 + \eta \|\nabla f(\mathbf{x}_t)\| \|\zeta_t\| + \frac{\eta^2 \ell}{2} \left[ \|\nabla f(\mathbf{x}_t)\|^2 + 2\|\nabla f(\mathbf{x}_t)\| \|\zeta_t\| + \|\zeta_t\|^2 \right] \\ \leq & f(\mathbf{x}_t) - \eta \|\nabla f(\mathbf{x}_t)\| \left[ \frac{1}{2} \|\nabla f(\mathbf{x}_t)\| - 2\|\zeta_t\| \right] + \frac{\eta}{2} \|\zeta_t\|^2 \leq f(\mathbf{x}_t) - \eta \epsilon^2 / 4 \end{aligned}$$

$\square$

**Lemma 51.** *Consider the setting of Theorem 48, if $\|\nabla f(\mathbf{x}_t)\| \leq \epsilon$ and $\lambda_{\min}(\nabla^2 f(\mathbf{x}_t)) \leq -\sqrt{\rho \epsilon}$, then by running Algorithm 1, with probability $1 - \delta$, we have $f(\mathbf{x}_{t+\mathscr{T}}) - f(\mathbf{x}_t) \leq -\mathscr{F}$.*

*Proof.* See next section. $\square$

*Proof of Theorem 48.* Combining lemma 50 and 51, we know with probability $1 - \frac{\Delta_f}{\mathscr{F}} \delta$, algorithm will find $\epsilon$-second order stationary point in following iterations:

$$\frac{\Delta_f}{\eta \epsilon^2} + \frac{\Delta_f \mathscr{T}}{\mathscr{F}} \leq O(\frac{2\Delta_f}{\eta \epsilon^2} \chi^4)$$

Let $\delta' = \frac{\Delta_f}{\mathscr{F}} \delta$ and substitute $\delta$ in $\chi$ with $\delta'$, since $\chi = \max\{1, \log \frac{d\ell \Delta_f}{\rho \epsilon \delta}\}$, this substitution only affects constants. Finally note in each iteration, we use $m$ queries, which finishes the proof. $\square$

## H.1 Proof of Lemma 51

**Lemma 52.** *Let $\eta \leq \frac{1}{\ell}$, then we have SGD satisfies:*

$$f(\mathbf{x}_{t+1}) - f(\mathbf{x}_t) \leq -\frac{\eta}{4}\|\nabla f(\mathbf{x}_t)\|^2 + 5\eta\|\zeta_t\|^2,$$

*where $\zeta_t = \mathbf{g}_t - \nabla f(\mathbf{x}_t) + \xi_t$.*

*Proof.* By assumption, function $f$ is $\ell$-gradient Lipschitz, we have:

$$\begin{aligned}
f(\mathbf{x}_{t+1}) \leq & f(\mathbf{x}_t) + \langle \nabla f(\mathbf{x}_t), \mathbf{x}_{t+1} - \mathbf{x}_t \rangle + \frac{\ell}{2}\|\mathbf{x}_{t+1} - \mathbf{x}_t\|^2 \\
\leq & f(\mathbf{x}_t) - \eta\langle \nabla f(\mathbf{x}_t), \nabla f(\mathbf{x}_t) + \zeta_t \rangle + \frac{\eta^2\ell}{2}(\|\nabla f(\mathbf{x}_t)\|^2 + 2\|\nabla f(\mathbf{x}_t)\|\|\zeta_t\| + \|\zeta_t\|^2) \\
\leq & f(\mathbf{x}_t) - \frac{\eta}{2}\|\nabla f(\mathbf{x}_t)\|^2 + 2\eta\|\nabla f(\mathbf{x}_t)\|\|\zeta_t\| + \frac{\eta}{2}\|\zeta_t\|^2 \\
\leq & f(\mathbf{x}_t) - \frac{\eta}{4}\|\nabla f(\mathbf{x}_t)\|^2 + \frac{9\eta}{2}\|\zeta_t\|^2
\end{aligned}$$

which finishes the proof. $\qquad\square$

**Lemma 53.** *(Improve or Localize) Suppose $\{\mathbf{x}_t\}_{t=0}^T$ is a SGD sequence, then for all $t \leq T$:*

$$\|\mathbf{x}_t - \mathbf{x}_0\|^2 \leq 8\eta T(f(\mathbf{x}_0) - f(\mathbf{x}_T)) + 50\eta^2 T \sum_{t=0}^{T-1} \|\zeta_t\|^2,$$

*where $\zeta_t = \mathbf{g}_t - \nabla f(\mathbf{x}_t) + \xi_t$*

*Proof.* For any $t \leq T - 1$, by Lemma 52, we have:

$$\begin{aligned}
\|\mathbf{x}_{t+1} - \mathbf{x}_t\|^2 \leq & \eta^2\|\nabla f(\mathbf{x}_t) + \zeta_t\|^2 \leq 2\eta^2\|\nabla f(\mathbf{x}_t)\|^2 + 2\eta^2\|\zeta_t\|^2 \\
\leq & 8\eta(f(\mathbf{x}_{t+1} - \mathbf{x}_t)) + 50\eta^2\|\zeta_t\|^2
\end{aligned}$$

By Telescoping argument, we have:

$$\sum_{t=0}^{T-1} \|\mathbf{x}_{t+1} - \mathbf{x}_t\|^2 \leq 8\eta(f(\mathbf{x}_T) - f(\mathbf{x}_0)) + 50\eta^2 \sum_{t=0}^{T-1} \|\zeta_t\|^2$$

Finally, by Cauchy-Schwarz, we have for all $t \leq T$:

$$\|\mathbf{x}_t - \mathbf{x}_0\|^2 \leq (\sum_{\tau=1}^{t} \|\mathbf{x}_\tau - \mathbf{x}_{\tau-1}\|)^2 \leq t \sum_{\tau=0}^{t-1} \|\mathbf{x}_{\tau+1} - \mathbf{x}_\tau\|^2 \leq T \sum_{\tau=0}^{T-1} \|\mathbf{x}_{\tau+1} - \mathbf{x}_\tau\|^2$$

which finishes the proof. $\qquad\square$

To study escaping saddle points, we need a notion of coupling. Recall the PSGD update has two source of randomness: $\mathbf{g}_t - \nabla f(\mathbf{x}_t)$ which is the stochasticity inside the gradient oracle and $\xi_t$ which is the perturbation we deliberately added into the algorithm to help escape saddle points. Let $\text{SGD}_\xi^{(t)}(\cdot)$ denote the update via SGD $t$ times with perturbation $\xi = \{\xi_2, \cdots\}$ fixed. Define Stuck region:

$$\mathcal{X}_{\text{stuck}}^\xi(\tilde{\mathbf{x}}) = \{\mathbf{x}|\mathbf{x} \in \mathbb{B}_{\tilde{\mathbf{x}}}(\eta r), \text{ and } \Pr(f(\text{SGD}_\xi^{(\mathscr{T})}(\mathbf{x})) - f(\tilde{\mathbf{x}}) \geq -\mathscr{F}) \geq \sqrt{\delta}\} \qquad (20)$$

Intuitively, the later perturbations of coupling sequence are the same, while the very first perturbation is used to escape saddle points.

**Lemma 54.** *There exists large enough constant $c$, so that if $\|\nabla f(\tilde{\mathbf{x}})\| \leq \epsilon$ and $\lambda_{\min}(\nabla^2 f(\tilde{\mathbf{x}})) \leq -\sqrt{\rho\epsilon}$, then the width of $\mathcal{X}_{\text{stuck}}^\xi(\tilde{\mathbf{x}})$ along the minimum eigenvector direction of $\tilde{\mathbf{x}}$ is at most $\delta\eta r\sqrt{2\pi/d}$.*

*Proof.* To prove this, let $\mathbf{e}_{\min}$ be the minimum eigenvector direction of $\nabla^2 f(\tilde{\mathbf{x}})$, it suffices to show for any $\mathbf{x}_0, \mathbf{x}_0' \in \mathbb{B}_{\tilde{\mathbf{x}}}(\eta r)$ so that $\mathbf{x}_0 - \mathbf{x}_0' = \lambda \mathbf{e}_{\min}$ where $|\lambda| \geq \delta \eta r \sqrt{2\pi/d}$, then either $\mathbf{x}_0 \notin \mathcal{X}_{\text{stuck}}^{\xi}(\tilde{\mathbf{x}})$ or $\mathbf{x}_0' \notin \mathcal{X}_{\text{stuck}}^{\xi}(\tilde{\mathbf{x}})$. Let $\mathbf{x}_{\mathscr{T}} = \text{SGD}^{(\mathscr{T})}(\mathbf{x}_0)$ and $\mathbf{x}_{\mathscr{T}}' = \text{SGD}^{(\mathscr{T})}(\mathbf{x}_0')$ where two sequence are independent. To show $\mathbf{x}_0 \notin \mathcal{X}_{\text{stuck}}^{\xi}(\tilde{\mathbf{x}})$ or $\mathbf{x}_0' \notin \mathcal{X}_{\text{stuck}}^{\xi}(\tilde{\mathbf{x}})$. We first argue showing following with probability $1 - \delta$ suffices:

$$\min\{f(\mathbf{x}_{\mathscr{T}}) - f(\tilde{\mathbf{x}}), f(\mathbf{x}_{\mathscr{T}}') - f(\tilde{\mathbf{x}})\} \leq -\mathscr{F} \tag{21}$$

Since $\mathbf{x}_{\mathscr{T}}$ and $\mathbf{x}_0'$ are independent, we have

$$\Pr(\mathbf{x}_1 \in \mathcal{X}_{\text{stuck}}^{\xi}(\tilde{\mathbf{x}})) \cdot \Pr(\mathbf{x}_2 \in \mathcal{X}_{\text{stuck}}^{\xi}(\tilde{\mathbf{x}})) = \Pr(\mathbf{x}_1 \in \mathcal{X}_{\text{stuck}}^{\xi}(\tilde{\mathbf{x}}) \text{ and } \mathbf{x}_2 \in \mathcal{X}_{\text{stuck}}^{\xi}(\tilde{\mathbf{x}})) \leq \delta$$

This gives $\min\{\Pr(\mathbf{x}_1 \in \mathcal{X}_{\text{stuck}}^{\xi}(\tilde{\mathbf{x}}), \Pr(\mathbf{x}_2 \in \mathcal{X}_{\text{stuck}}^{\xi}(\tilde{\mathbf{x}})\} \leq \sqrt{\delta}$ i.e. $\mathbf{x}_0 \notin \mathcal{X}_{\text{stuck}}^{\xi}(\tilde{\mathbf{x}})$ or $\mathbf{x}_0' \notin \mathcal{X}_{\text{stuck}}^{\xi}(\tilde{\mathbf{x}})$ by definition.

In the remaining proof, we will proceed proving Eq.(21) by showing two steps:

1. $\max\{f(\mathbf{x}_0) - f(\tilde{\mathbf{x}}), f(\mathbf{x}_0') - f(\tilde{\mathbf{x}})\} \leq \mathscr{F}$

2. $\min\{f(\mathbf{x}_{\mathscr{T}}) - f(\mathbf{x}_0), f(\mathbf{x}_{\mathscr{T}}') - f(\mathbf{x}_0')\} \leq -2\mathscr{F}$ with probability $1 - \delta$

The final result immediately follow from triangle inequality.

**Part 1.** Since $\mathbf{x}_0 \in \mathbb{B}_{\tilde{\mathbf{x}}}(\eta r)$ and $\|\nabla f(\mathbf{x})\| \leq \epsilon$, by smoothness, we have:

$$f(\mathbf{x}_0) - f(\tilde{\mathbf{x}}) \leq \epsilon \eta r + \frac{\ell}{2}(\eta r)^2 \leq O(\frac{\epsilon^2}{\ell}\chi^{-3}c^{-6}) \leq \mathscr{F}$$

The last inequality is due to $\ell/\sqrt{\rho\epsilon} \geq 1$, and constant $c$ large enough. By symmetry, we can also prove same upper bound for $f(\mathbf{x}_0') - f(\tilde{\mathbf{x}})$.

**Part 2.** Assume the contradiction $\min\{f(\mathbf{x}_{\mathscr{T}}) - f(\mathbf{x}_0), f(\mathbf{x}_{\mathscr{T}}') - f(\mathbf{x}_0')\} \geq -2\mathscr{F}$, by Lemma 53 (note $\|\zeta_t\| \leq \|\mathbf{g}_t - \nabla f(\mathbf{x}_t)\| + \|\xi_t\| \leq 2r$ with high probability when $m$ is large enough), with $1 - \delta/2$ probability, this implies localization:

$$\forall t \leq \mathscr{T}, \quad \max\{\|\mathbf{x}_t - \tilde{\mathbf{x}}\|, \|\mathbf{x}_t' - \tilde{\mathbf{x}}\|\}$$
$$\leq \max\{\|\mathbf{x}_t - \mathbf{x}_0\| + \|\mathbf{x}_0 - \tilde{\mathbf{x}}\|, \|\mathbf{x}_t' - \mathbf{x}_0'\| + \|\mathbf{x}_0' - \tilde{\mathbf{x}}\|\}$$
$$\leq \sqrt{8\eta\mathscr{T}\mathscr{F} + 50\eta^2\mathscr{T}\epsilon^2\chi^{-4}c^{-4}} + \eta r := \mathscr{S} = O(\sqrt{\frac{\epsilon}{\rho}}\chi^{-1}c^{-2})$$

That is, both SGD sequence $\{\mathbf{x}_t\}_{t=0}^{\mathscr{T}}$ and $\{\mathbf{x}_t'\}_{t=0}^{\mathscr{T}}$ will not leave a local ball with radius $\mathscr{S}$ around $\tilde{\mathbf{x}}$. Denote $\mathcal{H} = \nabla^2 f(\tilde{\mathbf{x}})$. By stochastic gradient update $\mathbf{x}_{t+1} = \mathbf{x}_t - \eta(\mathbf{g}_t(\mathbf{x}_t) + \xi_t)$, we can track the difference sequence $\mathbf{w}_t := \mathbf{x}_t - \mathbf{x}_t'$ as:

$$\mathbf{w}_{t+1} = \mathbf{w}_t - \eta[\nabla f(\mathbf{x}_t) - \nabla f(\mathbf{x}_t')] - \eta\boldsymbol{h}_t = (\mathbf{I} - \eta\mathcal{H})\mathbf{w}_t - \eta(\Delta_t\mathbf{w}_t + \boldsymbol{h}_t)$$
$$= (\mathbf{I} - \eta\mathcal{H})^{t+1}\mathbf{w}_0 - \eta\sum_{\tau=0}^{t}(\mathbf{I} - \eta\mathcal{H})^{t-\tau}(\Delta_\tau\mathbf{w}_\tau + \boldsymbol{h}_\tau),$$

where $\mathcal{H} = \nabla^2 f(\tilde{\mathbf{x}})$ and $\Delta_t = \int_0^1[\nabla^2 f(\mathbf{x}_t' + \theta(\mathbf{x}_t - \mathbf{x}_t')) - \mathcal{H}]d\theta$ and $\boldsymbol{h}_t = \mathbf{g}_t(\mathbf{x}_t) - \mathbf{g}_t(\mathbf{x}_t') - [\nabla f(\mathbf{x}_t) - \nabla f(\mathbf{x}_t')]$. By Hessian Lipschitz, we have $\|\Delta_t\| \leq \rho\max\{\|\mathbf{x}_t - \tilde{\mathbf{x}}\|, \|\mathbf{x}_t' - \tilde{\mathbf{x}}\|\} \leq \rho\mathscr{S}$. We use induction to prove following:

$$\|\eta\sum_{\tau=0}^{t-1}(\mathbf{I} - \eta\mathcal{H})^{t-1-\tau}(\Delta_\tau\mathbf{w}_\tau + \boldsymbol{h}_\tau)\mathbf{w}_\tau\| \leq \frac{1}{2}\|(\mathbf{I} - \eta\mathcal{H})^t\mathbf{w}_0\|$$

That is, the first term is always the dominating term. It is easy to check for base case $t = 0$; we have $0 \leq \|\mathbf{w}_0\|/2$. Suppose for all $t' \leq t$ the induction holds, this gives:

$$\|\mathbf{w}_{t'}\| \leq \|(\mathbf{I} - \eta\mathcal{H})^{t'}\mathbf{w}_0\| + \|\eta\sum_{\tau=0}^{t'-1}(\mathbf{I} - \eta\mathcal{H})^{t'-1-\tau}(\Delta_\tau\mathbf{w}_\tau + \boldsymbol{h}_\tau)\| \leq 2\|(\mathbf{I} - \eta\mathcal{H})^{t'}\mathbf{w}_0\|$$

Denote $\gamma = \lambda_{\min}(\nabla^2 f(\tilde{\mathbf{x}}))$, for case $t + 1 \le \mathcal{T}$, we have:

$$\left\|\eta \sum_{\tau=0}^{t} (\mathbf{I} - \eta\mathcal{H})^{t-\tau}\Delta_\tau \mathbf{w}_\tau\right\| \le \eta\rho\mathscr{S}\sum_{\tau=0}^{t}\|(\mathbf{I}-\eta\mathcal{H})^{t-\tau}\|\|\mathbf{w}_\tau\| \le \eta\rho\mathscr{S}\sum_{\tau=0}^{t}(1+\eta\gamma)^t\|\mathbf{w}_0\|$$

$$\le \eta\rho\mathscr{S}(t+1)\|(\mathbf{I}-\eta\mathcal{H})^{t+1}\mathbf{w}_0\| \le \eta\rho\mathscr{S}\mathcal{T}\|(\mathbf{I}-\eta\mathcal{H})^{t+1}\mathbf{w}_0\|$$

$$\le \frac{1}{4}\|(\mathbf{I}-\eta\mathcal{H})^{t+1}\mathbf{w}_0\|,$$

where the third last inequality use the fact $\mathbf{w}_0$ is along minimum eigenvector direction of $\mathcal{H}$, the last inequality uses the fact $\eta\rho\mathscr{S}T = c^{-1} \le 1/4$ for $c$ large enough.

On the other hand, with $1 - \delta/2$ probability, we also have:

$$\left\|\eta\sum_{\tau=0}^{t}(\mathbf{I}-\eta\mathcal{H})^{t-\tau}\boldsymbol{h}_\tau\right\| \le \eta\sum_{\tau=0}^{t}(1+\eta\gamma)^{t-\tau}\|\boldsymbol{h}_\tau\| \le (1+\eta\gamma)^{t+1}\frac{\max_\tau\|\boldsymbol{h}_\tau\|}{\gamma} \le \frac{1}{4}\|(\mathbf{I}-\eta\mathcal{H})^{t+1}\mathbf{w}_0\|,$$

where the last inequality requires $\max_\tau\|\boldsymbol{h}_\tau\| \le \gamma\|\mathbf{w}_0\|$ which can be achieved by making minibatch size $m$ large enough. Now, by triangular inequality, we finishes the induction.

Finally, we have:

$$\|\mathbf{w}_\mathcal{T}\| \ge \|(\mathbf{I}-\eta\mathcal{H})^\mathcal{T}\mathbf{w}_0\| - \left\|\eta\sum_{\tau=0}^{\mathcal{T}-1}(\mathbf{I}-\eta\mathcal{H})^{\mathcal{T}-1-\tau}(\Delta_\tau\mathbf{w}_\tau + \boldsymbol{h}_\tau)\right\|$$

$$\ge \frac{1}{2}\|(\mathbf{I}-\eta\mathcal{H})^\mathcal{T}\mathbf{w}_0\| \ge \frac{(1+\eta\sqrt{\rho\epsilon})^\mathcal{T}\|\mathbf{w}_0\|}{2}$$

$$= 2^{\chi c} \cdot \frac{\delta\epsilon\chi^{-3}c^{-6}}{2\ell}\sqrt{\frac{2\pi}{d}} \ge 8\sqrt{\frac{\epsilon}{\rho}}\chi^{-1}c^{-2} = 2\mathscr{S},$$

where the last inequality requires

$$2^{\chi c} \ge \frac{16}{\sqrt{2\pi}} \cdot \frac{\ell\sqrt{d}}{\delta\sqrt{\rho\epsilon}}\chi^2 c^4$$

Since $\chi = \max\{1, \log\frac{d\ell\Delta_f}{\rho\epsilon\delta}\}$, it is easy to verify when $c$ large enough, above inequality holds. This gives $\|\mathbf{w}_\mathcal{T}\| \ge 2\mathscr{S}$, which contradicts with the localization fact $\max\{\|\mathbf{x}_\mathcal{T} - \tilde{\mathbf{x}}\|, \|\mathbf{x}'_\mathcal{T} - \tilde{\mathbf{x}}\|\} \le \mathscr{S}$.

$\square$

*Proof of Lemma 51.* Let $r_0 = \delta r\sqrt{\frac{2\pi}{d}}$ and applying Lemma 54, we know $\mathcal{X}^\xi_{\text{stuck}}(\mathbf{x}_t)$ has at most width $\eta r_0$ in the minimum eigenvector direction of $\nabla^2 f(\mathbf{x}_t)$ and thus,

$$\text{Vol}(\mathcal{X}^\xi_{\text{stuck}}) \le \text{Vol}(\mathbb{B}^{(d-1)}_0(\eta r)) \cdot \eta r_0$$

which gives:

$$\frac{\text{Vol}(\mathcal{X}^\xi_{\text{stuck}})}{\text{Vol}(\mathbb{B}^{(d)}_{\mathbf{x}_t}(\eta r))} \le \frac{\eta r_0 \times \text{Vol}(\mathbb{B}^{(d-1)}_0(\eta r))}{\text{Vol}(\mathbb{B}^{(d)}_0(\eta r))} = \frac{r_0}{r\sqrt{\pi}}\frac{\Gamma(\frac{d}{2}+1)}{\Gamma(\frac{d}{2}+\frac{1}{2})} \le \frac{r_0}{r\sqrt{\pi}} \cdot \sqrt{\frac{d}{2}+\frac{1}{2}} \le \delta$$

Therefore with $1 - \delta$ probability, the perturbation lands in $\mathbb{B}^{(d)}_{\mathbf{x}_t}(\eta r) - \mathcal{X}^\xi_{\text{stuck}}$, where by definition we have with probability at least $1 - \sqrt{\delta}$

$$f(\text{SGD}^{(\mathcal{T})}_\xi(\mathbf{x})) - f(\tilde{\mathbf{x}}) \le -\mathscr{F}$$

Therefore the probabilty of escaping saddle point is $(1-\delta)(1-\sqrt{\delta}) \ge 1 - 2\sqrt{\delta}$. Reparametrizing $\delta' = 2\sqrt{\delta}$ only affects constant factors in $\chi$, hence we finish the proof. $\square$