[Reviews · NeurIPS 2018]

Reviewer 1



This paper considers the problem of minimizing a non-convex smooth population risk function, where one has access to a 0-th order oracle that can evaluate the empirical risk. With this goal, the authors propose an algorithm (ZPSGD) which is essentially an instance of PSGD [Ge COLT 2015] on a Gaussian smoothing of the empirical loss. Under the standard regularity conditions of the population risk (boundedness, gradient, and Hessian Lipschitzness), and additional pointwise closeness of the empirical risk to the population risk, this paper shows the following results: 1. if empirical and population risk are (eps^1.5)/d pointwise close, then ZPSGD returns an eps-approximate second order stationary point of the population risk with a number of 0-th order queries polynomial in the problem parameters. 2. the (eps^1.5)/d pointwise closeness in optimal up to polylogarithmic factors for all algorithms that make a polynomial number of 0-th order queries. 3. if we let an exponential number of 0-th order queries, there is an algorithm for finding eps-approximate second order stationary point if the empirical and population risk are eps^1.5 pointwise, and this dependence is optimal. General Evaluation: The paper considers an important problem at the intersection of machine learning and non-convex optimization. The paper is well-written, and the contributions are clear. The mathematical derivations and the proofs seem correct (I have not checked all the proofs). My main concern is about the novelty of the paper. ZPSGD is essentially SPGD on a Gaussian smoothing of the empirical loss. The fact that unbiased estimates of the gradient of the Gaussian smoothing of a function can be obtained by function evaluations on a batch is not novel [Duchi ToIT 2015]. Convergence of PSGD when unbiased stochastic gradients have sub-Gaussian tails is also established in [Jin PC 2018]. So the main novelty of the paper is in setting up the smoothing radius such that together with the other parameters, guarantees of PSGD can be carried over to the Gaussian smoothing of the empirical loss, and that the population risk and its gradients and Hessians are close to those of the Gaussian smoothing of the empirical risk. Despite this, I still believe this paper makes important contributions that makes it a good fit for NIPS. ============================= I have carefully read the author response. It is not clear to me why a zeroth order oracle is interesting. Authors fail to motivate it from a computational perspective, and zeroth order oracles are not typical in machine learning. More importantly, if one does have access to a first-order oracle, then the results presented here are either well known or follow as simple corollaries to the existing results. So what is the significance of the work here? Secondly, somewhat equally important is the fact that the authors analyze the ZPSGD as an inexact gradient method. The proofs are rather straightforward, and again it is not clear what are the contributions from a technical perspective (no new tools/techniques being contributed). I still think it is a good paper and agree that the lower bound results are novel.

Reviewer 2



Population risk is ultimately of interest in machine learning and statistics, while a direct optimization acting at the population risk is impossible due to some unknown measurements, or it is unreasonable due to the purpose of privacy preservation. This paper is to find an approximate Local minima of the underling function $F$ by implementing a simple zero-order SGD for an approximate function of $F$. Moreover, the authors proved that the proposed algorithm is optimal under some framework. Finally, they provided a popular example to show their theoretical results clearly. In summary, I believe that the current paper is well written and has significant contribution in the literature of nonvex optimization, and particularly established some deep theoretical concussions for their proposed algorithm.

Reviewer 3



This paper studies the problem of minimizing the population risk by the proposed ZPSGD algorithm. The main contribution of this work is the new algorithm that can find an \epsilon approximate local minima of this nonconvex optimization problem. The paper is well-written but lacks the numerical results which can show the superiority of the proposed methods compared with the existing works.